# Optimal Conservative Offline RL with General Function Approximation via Augmented Lagrangian

**Paria Rashidinejad**[†]   **Hanlin Zhu**[†]   **Kunhe Yang**[†]   **Stuart Russell**[†]   **Jiantao Jiao**[†,‡]

[†]Department of Electrical Engineering and Computer Sciences
[‡]Department of Statistics
University of California, Berkeley
`{paria.rashidinejad,hanlinzhu,kunheyang,russell,jiantao}@berkeley.edu`

## Abstract

Offline reinforcement learning (RL), which aims at learning good policies from historical data, has received significant attention over the past years. Much effort has focused on improving offline RL practicality by addressing the prevalent issue of partial data coverage through various forms of conservative policy learning. While the majority of algorithms do not have finite-sample guarantees, several provable conservative offline RL algorithms are designed and analyzed within the single-policy concentrability framework that handles partial coverage. Yet, in the nonlinear function approximation setting where confidence intervals are difficult to obtain, existing provable algorithms suffer from computational intractability, prohibitively strong assumptions, and suboptimal statistical rates. In this paper, we leverage the marginalized importance sampling (MIS) formulation of RL and present the first set of offline RL algorithms that are statistically optimal and practical under general function approximation and single-policy concentrability, by-passing the need for uncertainty quantification. We identify that the key to successfully solving the sample-based approximation of the MIS problem is ensuring that certain *occupancy validity constraints* are nearly satisfied. We enforce these constraints by a novel application of the augmented Lagrangian method and prove the following result: with the MIS formulation, augmented Lagrangian is enough for statistically optimal offline RL. In stark contrast to prior algorithms that induce additional conservatism through methods such as behavior regularization, our approach provably eliminates this need and reinterprets regularizers as "enforcers of occupancy validity" than "promoters of conservatism."

## 1 Introduction

The goal of offline RL is to design agents that learn to achieve competence in a task using only a previously-collected dataset of interactions (Lange et al., 2012). Offline RL is a promising tool for many critical applications, from healthcare to autonomous driving to scientific discovery, where the online mode of learning by interacting with the environment is dangerous, impractical, costly, or even impossible (Levine et al., 2020). Despite this, offline RL has not yet been truly successful in practice (Fujimoto et al., 2019) and impressive RL performance has been limited to settings with known environments (Silver et al., 2017; Moravčík et al., 2017), access to accurate simulators (Mnih et al., 2015; Degrave et al., 2022; Fawzi et al., 2022), or expert demonstrations (Vinyals et al., 2017).

One of the central challenges in offline RL is the lack of uniform coverage in real datasets and the *distribution shift* between the occupancy of candidate policies and offline data distribution, which pose difficulties in accurately evaluating the candidate policies. Over the past years, a body of literature has focused on addressing this challenge through developing conservative algorithms, which aim at picking a policy among those well-covered in the data. On the practical front, various forms of conservatism are proposed such as behavior regularization through policy constraints (Kumar et al., 2019; Fujimoto et al., 2019; Nachum & Dai, 2020), learning conservative values (Kumar et al., 2020; Liu et al., 2020; Agarwal et al., 2020), or learning pessimistic models (Kidambi et al., 2020; Yu et al., 2020; 2021); see Appendix B for further discussion on related work.

From a theoretical standpoint, partial data coverage has recently been studied within variants of the single-policy concentrability framework (Rashidinejad et al., 2021; Xie et al., 2021; Uehara & Sun, 2021), which characterizes the distribution shift between offline data and occupancy of a target (often optimal) policy, in contrast to all-policy concentrability commonly used in earlier works (Scherrer, 2014; Chen & Jiang, 2019; Liao et al., 2020; Zhang et al., 2020a; Xie & Jiang, 2021). Within this framework and in the tabular and linear function approximation settings, pessimistic algorithms that leverage uncertainty quantifiers to construct lower confidence bounds (Jin et al., 2021; Rashidinejad et al., 2021; Yin et al., 2021; Shi et al., 2022; Li et al., 2022) enjoy optimal statistical rate. In the general function approximation setting, pessimistic algorithms largely assume oracle access to uncertainty quantification, either for constructing penalties that are subtracted from rewards (Jin et al., 2021; Jiang & Huang, 2020) or selecting the most pessimistic option among those that fall within the confidence region implied by the offline data (Uehara & Sun, 2021; Xie et al., 2021; Chen & Jiang, 2022). However, uncertainty quantifiers are difficult to obtain in non-linear function approximation and existing heuristics are empirically observed to be unreliable (Rashid et al., 2019; Tennenholtz et al., 2021; Yu et al., 2021). Recent works by Cheng et al. (2022) and Zhan et al. (2022) propose provable alternatives to uncertainty-based methods, but leave achieving optimal statistical rate of $1/\sqrt{N}$, where $N$ is the dataset size, as an open problem.

Among all, the marginal importance sampling (MIS) methods, which aim at learning weights $w$ that estimate the distribution shift between induced policy occupancy $d_w$ and data distribution $\mu$, lend themselves well to the single-policy concentrability framework. Though more popular in off-policy evaluation (Liu et al., 2018; Xie et al., 2019; Uehara et al., 2020; Zhang et al., 2020b), MIS has also been used for conservative offline RL such as AlgaeDICE (Nachum et al., 2019b) and OptiDICE (Lee et al., 2021), both of which incorporate behavior regularization. Recently, Zhan et al. (2022) theoretically studied a variant of OptiDICE, showing that MIS with behavior regularization enjoys finite-sample guarantees (though achieving a suboptimal $1/N^{1/6}$ rate) and circumvents certain fundamental difficulties observed in value-based offline RL with function approximation (Du et al., 2019; Wang et al., 2020; 2021; Weisz et al., 2021; Zanette, 2021; Foster et al., 2021).

## 1.1 Contributions and results

Motivated by the benefits offered by MIS, we study designing statistically optimal offline learning algorithms under the MIS formulation with general function approximation and single-policy concentrability. We conduct theoretical investigations and design algorithms starting from multi-armed bandits (MABs), going forward to contextual bandits (CBs), and finally Markov decision processes (MDPs). In the rest of this section, we present a preview of our contributions and results.

**Multi-armed bandits.** Empirical MIS algorithms often incorporate behavior regularization, whose role is justified as promoting conservatism by keeping the occupancies of learned and behavior policies close (Nachum et al., 2019b; Lee et al., 2021). Yet, whether and why these regularizers are necessary from a theoretical perspective remain unclear. Zhan et al. (2022) motivates behavior regularization as a way of introducing curvature in an otherwise linear optimization problem. We extensively investigate the effect of regularization, starting from the simplest setting of MABs with function approximation, as existing algorithms when specialized to offline MABs, are either intractable, have suboptimal finite-sample guarantees, or require access to uncertainty quantifiers.

We state our results on offline MABs with general function approximation and single-policy concentrability in the informal theorem below.

**Theorem (informal)** *(I) There exists an offline MAB instance where the unregularized MIS fails to achieve a suboptimality that decays with $N$. (II) MIS with behavior regularization (PRO-MAB Algorithm 1) achieves $O(1/\sqrt{N})$ suboptimality. (III) If one searches only over the space of weights that induce valid occupancies ($d_w = 1$), then unregularized MIS achieves $O(1/\sqrt{N})$ suboptimality.*

Here, we prove that unregularized MIS fails even in bandits and provide a tight analysis of PRO-MAB, a special case of PRO-RL algorithm, improving over the original $1/N^{1/6}$ rate shown by Zhan et al. (2022). In our analysis, we find that the key to the success of PRO-MAB is *near-validity of the learned occupancy $d_w$*. In MABs, the validity constraint simply requires the learned occupancy to be a probability distribution: $d_w = \sum_a w(a)\mu(a) = 1$, where $a$ is an arm. With a proper choice of hyperparameter, we show that behavior regularization enforces learned occupancy to be nearly valid: $d_w = \Omega(1)$. We further prove that regularization is not required if validity is otherwise satisfied.

Given that occupancy validity is the constraint of the optimization problem solved by MIS (see e.g. (1)), we ask whether there are any methods for solving empirical optimization problems that find more constraint-adhering solutions compared to those yielded by Lagrange multipliers adopted in prior works (Lee et al., 2021; Zhan et al., 2022). The augmented Lagrangian method (ALM), which adds a quadratic loss on the constraints $(d_w - 1)^2$, is a natural choice for our purpose. The ALM term can be easily estimated from offline data and forms Algorithm 1. We show that ALM results in $d_w = \Omega(1)$, ensuring near-validity of learned occupancy and leading to the following guarantee.

**Theorem (informal)** *The policy returned by an algorithm that combines ALM with MIS (ALMIS) for offline MABs (Algorithm 1) achieves $O(1/\sqrt{N})$ suboptimality.*

ALMIS offers several benefits over PRO-MAB such as eliminating the need for picking the regularizer and only requiring single-policy concentrability instead of the two-policy requirement of PRO-MAB, which can be strong (Section 5.3). Additionally, behavior regularization introduces bias in the solution even with infinite data (Chen & Jiang, 2022) and the bias-variance tradeoff must be carefully handled. However, ALM merely enforces the optimization constraints and leads to provably unbiased solutions (Lemma 14). More importantly, as we see shortly, going beyond the single-state MABs, behavior regularization becomes suboptimal while ALMIS maintains optimality.

**Contextual bandits.** In offline CBs, we analyze two approaches: MIS with behavior regularization, and an extension of ALMIS. We state our results in the following informal theorem.

**Theorem (informal)** *(I) There exists a CB instance where MIS with behavior regularization (PRO-CB Algorithm 6) suffers from suboptimality $\Omega(N^\beta)$ with $\beta > -1/2$. (II) Policy returned by ALMIS for offline CBs (Algorithm 2) achieves suboptimality of $O(1/\sqrt{N})$.*

Informally, the failure of PRO-CB to achieve the optimal rate is because the regularization parameter has to be small to control bias, but such small regularization is not strong enough to ensure the validity of learned occupancy in most states. Therefore, one must choose larger regularization, leading to an overall suboptimal rate. Prior works Chen & Jiang (2022); Cheng et al. (2022) also allude to this phenomenon, explaining that regularizers appear to be the culprit behind suboptimal rates. In CBs, the occupancy validity constraints require conditional occupancy to be a valid probability distribution in every state. In Algorithm 2, we incorporate ALM in offline CBs by adding a weighted sum of quadratic losses describing the validity constraint in each state, where the weights are set to the state occupancies to capture their relative importance. Enforcement of the constraints by ALM without introducing any bias is the key to the optimality of our algorithm.

**MDPs.** Validity constraints in MDPs ensure that the learned state occupancy $d_w(s) = \sum_a w(s, a)\mu(s, a)$ is close to the actual state occupancy $d^{\pi_w}(s)$, where $\pi_w$ is the policy computed from weights $w$.[1] Directly enforcing this constraint results in an ALM term that cannot easily be estimated from offline data. We address this difficulty by expressing the ALM term in the variational form. From there, we derive two variants, one model-based and one model-free, of the ALMIS algorithm for offline RL, that enjoy the following guarantee.

**Theorem (informal)** *Both variants of ALMIS for offline RL achieve $O(1/\sqrt{N})$ suboptimality.*

This marks ALMIS as the first practical and statistically optimal offline RL algorithm that operates in the general function approximation and partial data coverage setting, while avoiding uncertainty quantification and additional regularizers. Conservatism of ALMIS is baked into the MIS formulation and supported by the ALM: bounded MIS weights prevent learned occupancy to deviate significantly from data distribution, and ALM ensures closeness of the learned and actual occupancies. When combined, ALMIS learns a policy whose actual occupancy is close to the data distribution.

We thus proved that ALM improves sample complexity compared to alternatives such as behavior regularization. This is in addition to the benefits on optimization stability that are likely to be offered by the ALM, as the ALM improves over the ill-posed Lagrange multiplier objective (Ben-Tal & Nemirovski, 2022). Our theoretical findings can explain the empirical observations of Yang et al. (2020), who find MIS with behavior regularization to be unstable and propose regularizers in "the spirit of ALM" that gain superior performance and attribute performance gain to improved optimization. In this work, we present a theoretically-founded way of introducing ALM in offline RL and our analysis shows that ALM also leads to optimal sample complexity.

---

[1]Notice that the validity constraints in MAB and CB are special cases of this constraint.

## 2    BACKGROUND

**Markov decision process.**   An infinite-horizon discounted MDP is described by a tuple $M = (\mathcal{S}, \mathcal{A}, P, R, \rho, \gamma)$, where $\mathcal{S}$ is the state space, $\mathcal{A}$ is the action space, $P : \mathcal{S} \times \mathcal{A} \mapsto \Delta(\mathcal{S})$ is the transition kernel, $R : \mathcal{S} \times \mathcal{A} \mapsto \Delta([0,1])$ encodes a family of reward distributions with $r : \mathcal{S} \times \mathcal{A} \mapsto [0,1]$ as the expected reward function, $\rho : \mathcal{S} \mapsto \Delta(\mathcal{S})$ is the initial state distribution, and $\gamma \in [0,1)$ is the discount factor. We assume $\mathcal{S}$ and $\mathcal{A}$ are finite however, our results do not depend on their cardinalities and can be naturally extended to infinite sets. A stationary (stochastic) policy $\pi : \mathcal{S} \mapsto \Delta(\mathcal{A})$ specifies a distribution over actions in each state. Each policy $\pi$ induces an occupancy density over state-action pairs $d^\pi : \mathcal{S} \times \mathcal{A} \mapsto [0,1]$ defined as $d^\pi(s,a) := (1-\gamma) \sum_{t=0}^{\infty} \gamma^t P_t(s_t = s, a_t = a; \pi)$, where $P_t(s_t = s, a_t = a; \pi)$ denotes $(s,a)$ visitation probability at step $t$, starting at $s_0 \sim \rho(\cdot)$ and following $\pi$. We abuse notation and also write $d^\pi(s) = \sum_{a \in \mathcal{A}} d^\pi(s,a)$ to denote the discounted state occupancy. Additionally, operator $\mathbb{P}^\pi$ is applied to any function $u : \mathcal{S} \times \mathcal{A} \to \mathbb{R}$ and is defined as $(\mathbb{P}^\pi u)(s,a) := \sum_{s',a'} P(s'|s,a)\pi(a'|s')u(s',a')$.

An important quantity is the value a policy $\pi$, which is the discounted sum of rewards $V^\pi(s) := \mathbb{E}[\sum_{t=0}^{\infty} \gamma^t r_t \mid s_0 = s, a_t \sim \pi(\cdot \mid s_t), \ \forall \ t \geq 0]$ starting at $s \in \mathcal{S}$. Q-function $Q^\pi(s,a)$ of a policy is similarly defined. We write $J(\pi) := (1-\gamma)\mathbb{E}_{s \sim \rho}[V^\pi(s)] = \mathbb{E}_{s,a \sim d^\pi}[r(s,a)]$ to represent a scalar summary of the performance of a policy $\pi$. We denote by $\pi^\star$ an optimal policy that maximizes the above objective and use the shorthand $V^\star := V^{\pi^\star}$ to denote the optimal value function.

**Offline reinforcement learning.**   We focus on the offline RL, where the agent is only provided with a previously-collected offline dataset $\mathcal{D} = \{(s_i, a_i, r_i, s_i')\}_{i=1}^{N}$. Here, $r_i \sim R(s_i, a_i)$, $s_i' \sim P(\cdot \mid s_i, a_i)$, and we assume $s_i, a_i$ pairs are generated i.i.d. according to a data distribution $\mu \in \Delta(\mathcal{S} \times \mathcal{A})$. To streamline the analysis, we assume that the conditional distribution $\mu(a|s)$ is known.[2] The goal of offline RL is to learn a policy $\hat{\pi}$ based on the offline dataset so as to minimize the sub-optimality with respect to an optimal policy $\pi^\star$, i.e. $J(\pi^\star) - J(\hat{\pi})$ with high probability.

**Marginalized importance sampling.**   In this paper, we consider marginalized importance sampling (MIS) formulation that aims at learning weights $w(s,a)$ to represent policy occupancy when multiplied by data distribution: $d_w(s,a) = w(s,a)\mu(s,a)$. Also denote $d_w(s) = \sum_{a \in \mathcal{A}} d_w(s,a)$. We define the policy induced by $w$ as $\pi_w(a|s) = d_w(s,a)/d_w(s)$ for $d_w(s) > 0$ and $\pi_w(a|s) = 1/|\mathcal{A}|$ for $d_w(s) = 0$.

**Offline data coverage assumption.**   We design and analyze our algorithms within the single-policy concentrability framework (Rashidinejad et al., 2021), stated below.

**Definition 1 (Single-policy concentrability)**  *Given a policy $\pi$, define $C^\pi$ to be the smallest constant that satisfies $\frac{d^\pi(s,a)}{\mu(s,a)} \leq C^\pi$ for all $s \in \mathcal{S}$ and $a \in \mathcal{A}$.*

$C^{\pi^\star} = C^\star$ captures coverage of $\pi^\star$ in the offline data and is much weaker than the widely used all-policy concentrability that assumes bounded $\max_\pi C^\pi$; see Appendix B for further discussion.

**Notation.**   We write $\Delta(\mathcal{S})$ to denote the probability simplex over a set $\mathcal{S}$. For a function class $\mathcal{F}$, we write $|\mathcal{F}|$ to denote its complexity (such as cardinality in the discrete case or covering number in the continuous case). We use the notation $x \lesssim y$ when there exists constant $c > 0$ such that $x \leq cy$ and $x \asymp y$ if constants $c_1, c_2 > 0$ exist such that $c_1|x| \leq |y| \leq c_2|x|$. We write $f(x) = O(g(x))$ if $M > 0, x_0$ exist such that $|f(x)| \leq Mg(x)$ for all $x \geq x_0$ and use $\widetilde{O}(\cdot)$ to be the big-$O$ notation ignoring logarithmic factors. Define $\mathrm{clip}(x,a,b) \triangleq \max\{a, \min\{x, b\}\}$ for $x, a, b \in \mathbb{R}$.

## 3    MULTI-ARMED BANDITS

We start by considering the offline learning problem in the multi-armed bandit (MAB) setting, which is a special case of MDP with $\gamma = 0$, $|\mathcal{S}| = 1$, and $\mathcal{D} = \{(a_i, r_i)\}_{i=1}^{N}$, where $a_i \sim \mu(\cdot), r_i \sim R(a_i)$. The goal of offline learning in MABs can be described as the following constrained optimization problem, where $d$ represents occupancy over actions (arms)

$$\max_{d \geq 0} \mathbb{E}_{a \sim d}[r(a)] \qquad \text{s.t.} \quad \sum_a d(a) = 1. \tag{1}$$

---

[2]When $\mu(a|s)$ is unknown, behavioral cloning can be used (Ross & Bagnell, 2014; Zhan et al., 2022).

## 3.1 PRIMAL-DUAL REGULARIZED OFFLINE BANDITS

To solve (1), the MIS approach with behavior regularization defines importance weights $w(a) = d(a)/\mu(a)$ and converts the problem (1) to its dual form by introducing the Lagrange multiplier $v$:

$$\max_{w \geq 0} \min_{v} L_\alpha^{\text{MAB}}(w, v) := \mathbb{E}_{a \sim \mu}[w(a)r(a)] - v(\mathbb{E}_{a \sim \mu}[w(a)] - 1) - \alpha\mathbb{E}_{a \sim \mu}[f(w(a))]. \quad (2)$$

The last term in (2) is the behavior regularizer that characterizes the $f$-divergence between the learned occupancy $d$ and data distribution $\mu$. This term was originally proposed to induce *conservatism* by keeping the learned policy close to behavior policy (Nachum et al., 2019b; Lee et al., 2021). Problem (2) satisfies strong duality and we denote optimal primal and dual variables by $w_\alpha^\star$ and $v_\alpha^\star$ (Appendix C.2). When $\alpha = 0$, weights $w^\star := w_0^\star$ (might not be unique) induce optimal policy and $v^\star := v_0^\star$ is the optimal reward. Approximating $w$ and $v$ to belong to classes $\mathcal{W} \subseteq \mathbb{R}^{|\mathcal{A}|}$ and $\mathcal{V} \in \mathbb{R}$ and solving the empirical version of (2) yields primal-dual regularized offline MAB (PRO-MAB Algorithm 5), a special case of PRO-RL algorithm of Zhan et al. (2022).

One might wonder whether the unregularized algorithm ($\alpha = 0$) is sufficient for solving the offline learning problem in MABs, particularly under the natural and common assumption that elements of the function class $\mathcal{W}$ are bounded: $w(a) = d(a)/\mu(a) \leq B_w$. In the following proposition, we show that the answer is negative and there exist a MAB instance in which the unregularized algorithm finds a policy that suffers from a constant suboptimality. The proof is provided in Appendix C.3.

**Proposition 1 (Unregularized MIS fails in MABs)** *Assume* $0 \leq w(a) \leq B_w$ *for* $w \in \mathcal{W}$ *and* $|v| \leq B_v$ *for* $v \in \mathcal{V}$. *Suppose realizability of any one of* $w^\star \in \mathcal{W}$ *and* $v^\star \in \mathcal{V}$ *and concentrability of* $\pi^\star := \pi_{w^\star}$. *For any* $N \geq 2$, *there exists a MAB instance where* $\hat{\pi}$ *returned by Algorithm 5 with* $\alpha = 0$ *satisfies* $J(\pi^\star) - J(\hat{\pi}) = 1/6$ *with a constant probability.*

We note that Zhan et al. (2022) also argues the failure of the unregularized algorithm by giving a counterexample in the MDP setting. We discuss this example in detail in Section 5.3. Proposition 1 reveals additional insights: the objective (16) with $\alpha = 0$ fails not just in MDPs but also in bandits, even when the optimal policy is unique and data are collected by running a behavior policy.

Given the failure of the unregularized algorithm, we conduct a tight analysis of PRO-MAB with $\alpha > 0$. In the next theorem, we prove that under similar assumptions as Zhan et al. (2022) and with a proper choice of $\alpha$, PRO-MAB returns a policy that enjoys optimal sample complexity.

**Theorem 1 (Suboptimality of PRO-MAB)** *Let* $f$ *be* $M_f$-*strongly convex and non-negative with bounded value* $|f(x)| \leq B_f$ *and derivative* $|f'(x)| \leq B_{f'}$. *Assume* $0 \leq w(a) \leq B_w$ *for* $w \in \mathcal{W}$ *and* $|v| \leq B_v$ *for* $v \in \mathcal{V}$. *Fix* $\delta \geq 0$ *and set* $\alpha \asymp B_w(B_v + 1) + B_f)/M_f\sqrt{\log(|\mathcal{V}||\mathcal{W}|/\delta)/N}$. *Suppose realizability of* $w_\alpha^\star \in \mathcal{W}$ *and* $v_\alpha^\star \in \mathcal{V}$ *and concentrability of* $\pi^\star := \pi_{w^\star}$ *and* $\pi_\alpha^\star := \pi_{w_\alpha^\star}$. *Then, with probability at least* $1 - \delta$, *policy* $\hat{\pi}$ *returned by Algorithm 5 achieves*

$$J(\pi^\star) - J(\hat{\pi}) \lesssim \frac{(B_f + B_w(B_v + 1))(B_f + B_{f'}B_w)}{M_f}\sqrt{\frac{\log(|\mathcal{V}||\mathcal{W}|/\delta)}{N}}.$$

To our knowledge, this is the first statistically optimal guarantee for a practical offline MAB algorithm with function approximation and partial coverage and improves over the $1/N^{1/6}$ guarantee given by Zhan et al. (2022). We now briefly explain the differences between the analysis methods; a complete proof is deferred to Appendix C.4. Zhan et al. (2022) bounds policy suboptimality by $\alpha + 1/(\alpha^{1/2}N^{1/4})$, where the first term stems from the bias caused by the regularizer and the second term emerges from bounding the difference of $\hat{w}$ and $w_\alpha^\star$ via strong convexity of $L_\alpha$. Optimizing the bound over $\alpha$ gives the final $1/N^{1/6}$ guarantee. In contrast, our analysis connects suboptimality to *occupancy validity*. We prove that suboptimality is bounded by $\alpha + 1/(d_{\hat{w}}\sqrt{N})$, where $d_{\hat{w}} = \sum_a \hat{w}(a)\mu(a)$. We then show that setting $\alpha \asymp 1/\sqrt{N}$ is sufficient to ensure near-validity of occupancy $d_{\hat{w}} = \Omega(1)$, yielding the optimal rate. We observe a similar phenomenon in Proposition 1 that small $d_w$ for certain $w \in \mathcal{W}$ can cause the unregularized MIS to fail. In the following section, we investigate this phenomenon further, leading to a new offline learning algorithm.

## 3.2 AUGMENTED LAGRANGIAN REPLACES BEHAVIOR REGULARIZATION

The next proposition further affirms the importance of policy validity and shows that if the occupancy is valid, such as by searching only over the weights that induce valid occupancies, then the unregularized algorithm enjoys an optimal rate. Proof of this result can be found in Appendix C.5.

---

**Algorithm 1** ALM with MIS (ALMIS) for offline MAB

---

1: **Inputs:** Dataset $\mathcal{D} = \{(a_i, r_i)\}_{i=1}^N$, classes $\mathcal{W}$ and $\mathcal{V}$.

2: Find a solution $\hat{w}, \hat{v}$ to the following problem

$$\max_{w \in \mathcal{W}} \min_{v \in \mathcal{V}} \hat{L}_{AL}^{\text{MAB}}(w, v) := \frac{1}{N} \sum_{i=1}^N w(a_i) r_i - v(w(a_i) - 1) - \left(\frac{1}{N} \sum_{i=1}^N w(a_i) - 1\right)^2. \quad (3)$$

3: **Return:** $\hat{\pi} = \pi_{\hat{w}}$.

---

**Proposition 2 (Constraint satisfaction is sufficient in MAB)** *Assume as in Theorem 1. Let $\hat{\pi}$ be the output of Algorithm 5 with $\alpha = 0$ and assume that $\sum_a \mu(a)\hat{w}(a) = 1$. Then, for any $\delta > 0$, the following holds with probability of at least $1 - \delta$*

$$J(\pi^\star) - J(\hat{\pi}) \lesssim (B_w(B_v + 1) + \alpha B_f) \sqrt{\frac{\log|\mathcal{V}||\mathcal{W}|/\delta}{N}}.$$

Motivated by the discussion above, we take a step back and ask: are there any other methods for solving constrained optimization problems that find *more constraint-satisfying* solutions when applied to the empirical approximation of the original problem? A promising candidate is the augmented Lagrangian method (ALM) which adds a quadratic loss on the constraints to the objective. Applied to (1), ALM forms the following objective, whose empirical version leads to Algorithm 1.

$$\max_{w \geq 0} \min_v L_{\text{AL}}^{\text{MAB}}(w, v) := \mathbb{E}_{a \sim \mu}[w(a)r(a)] - v(\mathbb{E}_{a \sim \mu}[w(a)] - 1) - (\mathbb{E}_{a \sim \mu}[w(a)] - 1)^2. \quad (4)$$

The following theorem establishes an upper bound on the suboptimality of the policy returned by Algorithm 1. This theorem is a special case of Theorem 3, whose proof is given in Appendix D.3.

**Theorem 2 (Suboptimality of Algorithm 1)** *Assume that $0 \leq w(a) \leq B_w$ for any $w \in \mathcal{W}$ and $|v| \leq B_v$ for any $v \in \mathcal{V}$. Further suppose realizability of any one of $w^\star \in \mathcal{W}$ and $v^\star \in \mathcal{V}$ and concentrability of $\pi^\star = \pi_{w^\star}$. For any fixed $\delta > 0$, policy $\hat{\pi}$ returned by Algorithm 1 achieves the following bound with probability of at least $1 - \delta$*

$$J(\pi^\star) - J(\hat{\pi}) \lesssim B_w^2(B_v + 1) \sqrt{\frac{\log(|\mathcal{W}||\mathcal{V}|/\delta)}{N}}. \quad (5)$$

In the proof, we show that ALM results in near-validity of $\hat{w}$ by ensuring that $d_{\hat{w}} = \Omega(1)$, leading to the optimal rate. Note that Algorithm 1 does not include any explicit form of conservatism through regularizers or uncertainty quantifiers. Colloquially, the MIS formulation and boundedness of $\mathcal{W}$ elements ensure that $d_{\hat{w}}(a)/\mu(a) = \hat{w}(a) \leq B_w$ and ALM ensures that $d_{\hat{w}}$ is close to the actual occupancy. Thus, Algorithm 1 seeks a policy whose *actual occupancy* is within data distribution. Algorithm 1 offers several benefits compared to PRO-MAB: it only requires $\pi^\star$-concentrability instead of the $\pi^\star, \pi_\alpha^\star$-concentrability requirement of PRO-MAB, removes the need to design regularization function $f$ and adjust $\alpha$, and does not introduce bias in the objective. The main advantage of ALM, however, becomes more evident as we move beyond bandits, where the behavior regularization provably fails to achieve the optimal statistical rate while ALM maintains optimality.

## 4 CONTEXTUAL BANDITS

The problem offline contextual bandits (CB) is a special case of offline RL with $\gamma = 0$ and offline dataset $\mathcal{D} = \{(s_i, a_i, r_i)\}_{i=1}^N$, where $s_i \sim \mu(\cdot) = \rho(\cdot)$, $a_i \sim \mu(\cdot \mid a_i)$, and $r_i \sim R(s_i, a_i)$. The linear programming constrained optimization problem for CB is given by

$$\max_{d \geq 0} \mathbb{E}_{s, a \sim d}[r(s, a)] \qquad \text{s.t.} \quad \sum_a d(s, a) = \rho(s) \quad \forall s \in \mathcal{S}. \quad (6)$$

### 4.1 ANALYSIS OF PRIMAL-DUAL REGULARIZED OFFLINE CONTEXTUAL BANDITS

In the following proposition, we prove a performance lower bound on the primal-dual regularized offline CB (PRO-CB) presented in Algorithm 6, which is MIS with behavior regularization.

**Proposition 3 (PRO-CB is suboptimal)** *Let $f$ be $M_f$-strongly convex, differentiable, and non-negative with bounded values $|f(x)| \leq B_f$ and derivative $|f'(x)| \leq B_{f'}$. Assume $0 \leq w(s, a) \leq B_w$ for $w \in \mathcal{W}$, $|v(s)| \leq B_v$, realizability $w^\star, w_\alpha^\star \in \mathcal{W}$, $v^\star, v_\alpha^\star \in \mathcal{V}$, and concentrability of $\pi^\star, \pi_\alpha^\star$. Let $\hat{\pi}$ be the output of Algorithm 6. Then, for $N \geq poly(\delta, B_w, B_v, B_f, B_{f'})$ and any $\alpha \geq 0$ there exists a CB instance such that $J(\pi^\star) - J(\hat{\pi}) \gtrsim N^\beta$ with a constant probability, where $\beta > -1/2$.*

---

**Algorithm 2** ALM with MIS (ALMIS) for offline CB

1: **Inputs:** Dataset $\mathcal{D} = \{(s_i, a_i, r_i)\}_{i=1}^N$, function classes $\mathcal{W}, \mathcal{V}$
2: Find a solution $\hat{w}, \hat{v}$ to the following problem

$$\max_{w \in \mathcal{W}} \min_{v \in \mathcal{V}} \hat{L}_{\text{AL}}^{\text{CB}}(w, v) \coloneqq \frac{1}{N} \sum_{i=1}^N w(s_i, a_i)(r_i - v(s_i)) + v(s_i) - \Big(\sum_{a \in \mathcal{A}} w(s_i, a)\mu(a|s_i) - 1\Big)^2 \quad (8)$$

3: **Return:** $\hat{\pi} = \pi_{\hat{w}}$.

---

Proposition 3 shows that behavior regularization is statistically suboptimal regardless of $\alpha$. The proof is presented in Appendix D.2. The main takeaway is that ensuring occupancy validity $\sum_a \hat{w}(s, a)\mu(a|s) = \Omega(1)$ for nearly all states appears to be critical in achieving the optimal rate. Yet, without introducing a large bias, behavior regularization is insufficient for such a guarantee.

### 4.2 OFFLINE CONTEXTUAL BANDITS WITH AUGMENTED LAGRANGIAN

To encourage occupancy validity, we extend ALM to CBs and propose the following objective:

$$\max_{w \geq 0} \min_{v} L_{\text{AL}}^{\text{CB}}(w, v)$$
$$\coloneqq \mathbb{E}_\mu\left[w(s, a)r(s, a)\right] - \mathbb{E}_\mu[v(s)(w(s, a) - 1)] - \mathbb{E}_{s \sim \mu}[(\mathbb{E}_{a \sim \mu(\cdot|s)}[w(s, a)] - 1)^2] \quad (7)$$

When $|\mathcal{S}| = 1$, (7) simplifies to the ALM objective (2) for MABs. The ALM term can be understood as follows. Each element encourages the validity of occupancy in each state $\sum_a w(s, a)\mu(s, a) \approx 1$ and the elements are weighted according to the true state distribution: validity is more important in states that are actually more likely to be visited. Denote by $w^\star$ an optimal solution to (7), which is equal to the optimal solution of (6), and define $v^\star(s) \coloneqq V^\star(s)$. The following theorem states that ALMIS achieves optimal rate in offline CBs, whose proof is in Appendix D.3.

**Theorem 3 (Suboptimality of Algorithm 2)** *Assume $0 \leq w(s, a) \leq B_w$ for $w \in \mathcal{W}$ and $v(s) \leq B_v$ for $v \in \mathcal{V}$. Moreover, suppose realizability of any one of $w^\star \in \mathcal{W}$ and $v^\star \in \mathcal{V}$ and concentrability of $\pi^\star = \pi_{w^\star}$. For any fixed $\delta \geq 0$, policy $\hat{\pi}$ returned by Algorithm 2 achieves the following suboptimality bound with probability of at least $1 - \delta$*

$$J(\pi^\star) - J(\hat{\pi}) \lesssim B_w^2(B_v + 1)\sqrt{\frac{\log(|\mathcal{W}||\mathcal{V}|/\delta)}{N}}.$$

## 5 MARKOV DECISION PROCESSES

We now turn to offline RL. In addition to the offline dataset, we assume access to a dataset $\mathcal{D}_0 = \{s_i\}_{i=1}^N$ with i.i.d. samples from the initial distribution $\rho$, similar to prior works (Lee et al., 2021; Zhan et al., 2022). The linear programming formulation of RL (Puterman, 2014) solves

$$\max_{d \geq 0} \mathbb{E}_{s, a \sim d}[r(s, a)] \quad \text{s.t.} \quad d(s) = (1 - \gamma)\rho(s) + \gamma \sum_{s', a'} P(s|s', a')d(s', a') \quad \forall s \in \mathcal{S}. \quad (9)$$

The constraints are known as Bellman flow equations and restrict the search to the space of valid occupancy distributions $d^\pi$ that can be induced in the MDP by running a policy $\pi$.

### 5.1 CONSERVATIVE OFFLINE RL WITH AUGMENTED LAGRANGIAN

Motivated by the success of ALM in bandits, we propose the following extension to offline RL:

$$\max_{w \geq 0} \min_{v} L_{\text{AL}}^{\text{MDP}}(w, v) \coloneqq (1 - \gamma)\mathbb{E}_\rho[v(s)] + \mathbb{E}_\mu\left[w(s, a)e_v(s, a)\right] - \mathbb{E}_{d^{\pi_w}}\left[\left[\frac{d_w(s)}{d^{\pi_w}(s)} - 1\right]^2\right] \quad (10)$$

where $e_v(s, a) \coloneqq r(s, a) + \gamma \sum_{s'} P(s'|s, a)v(s') - v(s)$. One can check that the first two terms are the Lagrange dual of (9) and the last term is a generalization of the ALM terms in bandits. The ALM elements encourage occupancy $d_w(s)$ to be close in ratio to the actual occupancy $d^{\pi_w}(s)$ in each state and as before, the ALM elements are weighted according to their actual visitation $d^{\pi_w}(s)$. Our particular ALM construction can be intuitively understood as follows. The MIS formulation learns bounded weights $\hat{w}(s, a) = d_{\hat{w}}(s, a)/\mu(s, a) \leq B_w$. The ALM term ensures that the ratio $d_{\hat{w}}(s)/d^{\pi_{\hat{w}}}(s) = \Omega(1)$ which roughly translates to $d^{\pi_{\hat{w}}}(s, a)/\mu(s, a) \lesssim B_w$.

The ALM term in (10) is difficult to estimate as it involves the expectation over unknown occupancy $d^{\pi_w}$ and the computation of the ratio $d_w(s)/d^{\pi_w}(s)$. We resolve this difficulty in the next sections.

---

**Algorithm 3** ALM with MIS (ALMIS) for offline RL — Model-based

---

1: **Inputs:** Datasets $\mathcal{D}, \mathcal{D}_0, \mathcal{D}_m$, function classes $\mathcal{W}, \mathcal{V}, \mathcal{U}, \mathcal{P}, f_*^{-1}(x) = 2\sqrt{x+1} - 2$.
2: Estimate transitions via maximum likelihood: $\hat{P} = \operatorname{argmax}_{P \in \mathcal{P}} \sum_{i=1}^{N_m} \ln P(s_i'|s_i, a_i)$.
3: Find a solution $\hat{w}, \hat{v}, \hat{u}$ to the following problem

$$\max_{w \in \mathcal{W}} \min_{v \in \mathcal{V}} \min_{u \in \mathcal{U}} \hat{L}_{AL}^{\text{model-based}}(w, v) := \frac{(1-\gamma)}{N_0} \sum_{i=1}^{N_0} \left( v(s_i) + \sum_a u(s_i, a)\pi_w(a|s_i) \right)$$

$$+ \frac{1}{N} \sum_{i=1}^{N} w(s_i, a_i) \left[ r_i + \gamma v(s_i') - v(s_i) - f_*^{-1}\left( u(s_i, a_i) - \gamma(\hat{\mathbb{P}}^{\pi_w} u)(s_i, a_i) \right) \right] \tag{14}$$

4: **Return:** $\hat{\pi} = \pi_{\hat{w}}$.

---

## 5.2 ESTIMATING THE ALM TERM AND ALMIS ALGORITHMS FOR OFFLINE RL

We view the ALM term as the negative $f$-divergence between $d_w$ and $d^{\pi_w}$ with $f(x) := (x-1)^2$ and express it in the variational form (Nguyen et al., 2010):

$$-\mathbb{E}_{d^{\pi_w}} \left( \frac{d_w(s)}{d^{\pi_w}(s)} - 1 \right)^2 = -D_f(d_w \| d^{\pi_w}) = \min_x \mathbb{E}_{d^{\pi_w}}[f_*(x(s,a))] - \mathbb{E}_{d_w}[x(s,a)]. \tag{11}$$

Here, $f_*$ is the convex conjugate of $f$ and we used the fact that $d_w(s,a)/d^{\pi_w}(s,a) = d_w(s)/d^{\pi_w}(s)$. Notice that $\mathbb{E}_{d^{\pi_w}}[f_*(x(s,a))]$ is the value of $\pi_w$ in the same MDP but with rewards $f_*(x(s,a))$. Define $u$ as the fixed point of the following Bellman equation

$$u(s,a) := f_*(x(s,a)) + \gamma(\mathbb{P}^{\pi_w} u)(s,a). \tag{12}$$

Since $u(s,a)$ is the Q-function of $\pi_w$ with rewards $f_*(x(s,a))$, we can rewrite (11) as

$$(11) = \min_u (1-\gamma)\mathbb{E}_{s \sim \rho, a \sim \pi_w}[u(s,a)] - \mathbb{E}_\mu \left[ w(s,a) f_*^{-1}\left( u(s,a) - \gamma(\mathbb{P}^{\pi_w} u)(s,a) \right) \right]. \tag{13}$$

Equation (13) involves expectations over $\rho$ and $\mu$, which can be estimated empirically. Below, we discuss model-free and model-based methods for estimating the term involving the transition operator $\mathbb{P}^{\pi_w}$. We include some details on practical implementations in Appendix E.1.

**Model-based ALMIS.** For the model-based route, we assume access to a class $\mathcal{P}$ that contains the true transitions and an additional dataset $\mathcal{D}_m = \{(s_i, a_i, s_i')\}_{i=1}^{N_m}$, where $s_i, a_i \sim \mu$ and $s_i' \sim P(.|s_i, a_i)$. Given $\mathcal{D}_m$, we obtain a maximum likelihood estimate of transitions and approximate the expectations using $\mathcal{D}_0$ and $\mathcal{D}$, which leads to model-based ALMIS for offline RL (Algorithm 3).

**Model-free ALMIS.** As an alternative, we consider developing a model-free that uses a single-sample estimate of $f_*^{-1}(u(s,a) - \gamma(P^{\pi_w} u)(s,a))$. This, however, roughly leads to the infamous double sampling problem (Baird, 1995). To circumvent this difficulty, in Appendix E.2 we use the dual embedding trick of Nachum et al. (2019a), to derive model-free ALMIS (Algorithm 4).

Theorem 4 shows that ALMIS for offline RL enjoys optimal rates; see Appendix E.3 for the proof.

**Theorem 4 (Suboptimality of ALMIS for offline RL)** *Assume* $0 \leq w(s,a) \leq B_w$ *for* $w \in \mathcal{W}$, $|v(s)| \leq B_v$ *for* $v \in \mathcal{V}$, *and* $|u(s,a)| \leq B_u$. *Suppose realizability of any one of* $w^\star \in \mathcal{W}$ *and* $v^\star(s) = V^\star(s) \in \mathcal{V}$ *and concentrability of* $\pi^\star = \pi_{w^\star}$. *Let* $\tilde{x}_w(s,a) = \operatorname{clip}(x_w^\star(s,a), -B_x, B_x)$, *where* $x_w^\star$ *is a solution to* (11) *and* $B_x = (1-\gamma)/4$, *and define* $u_w^\star$ *as the fixed-point solution to* (12) *when* $x = \tilde{x}_w$. *Assume* $u_w^\star \in \mathcal{U}$ *for any* $w \in \mathcal{W}$. *Then,* $B_u$ *satisfies* $(1-\gamma)^{-1}(B_x^2/4 + B_x) \leq B_u \leq \frac{1}{2}$. *Moreover, for any fixed* $\delta \geq 0$, *the following statements hold:*

*(I) Assume* $N = N_0 = N_m$ *for simplicity. If* $P^\star \in \mathcal{P}$, *then* $\hat{\pi}$ *returned by Algorithm 3 achieves*

$$J(\pi^\star) - J(\hat{\pi}) \lesssim \frac{B_v + B_u + (1 + B_v)B_w}{(1-\gamma)^3} \sqrt{\frac{B_u \log(|\mathcal{P}||\mathcal{U}||\mathcal{W}||\mathcal{V}|/\delta)}{N}}.$$

*(II) Assume* $N = N_0$ *for simplicity. Let* $\zeta_{w,u}^\star = \operatorname{argmax}_{\zeta < 0} L_{AL}^{\text{model-free}}(w, v, u, \zeta)$ *defined in* (54). *Assume* $\zeta_{w^\star,u}^\star \in \mathcal{Z}$ *for* $u \in \mathcal{U}$ *and* $B_{\zeta,L} \leq |\zeta(s,a)| \leq B_{\zeta,U}$ *for* $\zeta \in \mathcal{Z}$, *where* $B_{\zeta,L} \in (0, 2/(2 + B_x))$ *and* $B_{\zeta,U} \geq 2/(2 - B_x)$. *Let* $B_\zeta = \max\{B_{\zeta,U}, B_{\zeta,L}^{-1}\}$. *Then,* $\hat{\pi}$ *returned by Algorithm 4 achieves*

$$J(\pi^\star) - J(\hat{\pi}) \lesssim \frac{B_v + B_u + (1 + B_v + B_\zeta(B_u + 1))B_w}{(1-\gamma)^3} \sqrt{\frac{\log(|\mathcal{U}||\mathcal{W}||\mathcal{V}||\mathcal{Z}|/\delta)}{N}}.$$

---

**Algorithm 4** ALM with MIS (ALMIS) for offline RL — Model-free

---

1: **Inputs:** Datasets $\mathcal{D}, \mathcal{D}_0$, function classes $\mathcal{W}, \mathcal{V}, \mathcal{U}, \mathcal{Z}, g_*(x) = -x - 2 - \frac{1}{x}$.

2: Find a solution $\hat{w}, \hat{v}, \hat{u}, \hat{\zeta}$ to $\max_{w \in \mathcal{W}} \min_{v \in \mathcal{V}} \min_{u \in \mathcal{U}} \max_{\zeta \in Z} \hat{L}_{\text{AL}}^{\text{model-free}}(w, v, u, \zeta)$ defined as

$$
\frac{(1-\gamma)}{N_0} \sum_{i=1}^{N_0} v(s_i) + \sum_a u(s_i, a)\pi_w(a|s_i) + \frac{1}{N} \sum_{i=1}^{N} w(s_i, a_i)\Big[ r_i + \gamma v(s_i') - v(s_i)
$$
$$
+ \zeta(s_i, a_i)\Big( u(s_i, a_i) - \gamma \sum_{a' \in \mathcal{A}} u(s_i', a')\pi_w(a'|s_i') \Big) - g_*(\zeta(s_i, a_i)) \Big]
\tag{15}
$$

3: **Return:** $\hat{\pi} = \pi_{\hat{w}}$.

---

In Theorem 4, we make realizability assumptions on $u_w^\star$ for $w \in \mathcal{W}$ and $\zeta_{w^\star, u}^\star$ for $u \in \mathcal{U}$. Such assumptions are common in theory of RL with function approximation (Munos & Szepesvári, 2008; Xie et al., 2021; Jiang & Huang, 2020) and removing them can be difficult or impossible (Foster et al., 2021). Recently, Zhan et al. (2022); Chen & Jiang (2022) propose algorithms that only require optimal solution realizability, however, these algorithms are either intractable or suboptimal.

### 5.3 EXAMPLE: BEHAVIOR REGULARIZATION VS. AUGMENTED LAGRANGIAN

We examine the MDP example in Figure 1 presented by Zhan et al. (2022). Assume $\mathcal{V} = \{v^\star\}$ and $\mathcal{W} = \{w_1, w_2\}$, where $w_1$ always selects $L$ from $A$ and $w_2$ always selects $R$ from $A$. One can check $w_1(A, L) = 2, w_1(A, R) = 0$ and $w_2(A, L) = 0, w_2(A, R) = 1$.

**Unregularized algorithm.** As Zhan et al. (2022) state, the unregularized algorithm fails to distinguish between $w_1$ and $w_2$ even with infinite data as the objectives at $w_1$ and $w_2$ are exactly equal.

**Behavior regularization.** Consider an instantiation of PRO-RL with regularizer $-\alpha \mathbb{E}_\mu[w^2(s, a)]$. Since in this example $\mathbb{E}_\mu[w_1^2(s, a)] > \mathbb{E}_\mu[w_2^2(s, a)]$, PRO-RL picks the wrong weight $w_2$, suffering constant suboptimality. Note, however, that PRO-RL guarantees assume $\pi_\alpha^\star$-concentrability. Intuitively, behavior regularization causes $\pi_\alpha^\star$ to be more stochastic and thus requiring $\mu(s, a) > 0$ for more states and actions. Here, since $\mu$ covers both $(A, L)$ and $(A, R)$, behavior regularization causes $\pi_\alpha^\star(R|A) > 0$ and thus $d^{\pi_\alpha^\star}(C) > 0$. To handle the MDP in Figure 1, PRO-RL additionally requires $\mu(C) > 0$ to satisfy $\pi_\alpha^\star$-concentrability.

**ALM.** In this example, ALM successfully picks the optimal $w_1$, as it avoids a mismatch between the actual and learned occupancies. This is because in (10) the ALM term is zero at $w_1$ due to realizability whereas at $w_2$, it has a lower bound $\mathbb{E}_{s \sim d^{\pi_2}} \left( d_{w_2}(C)/d^{\pi_2}(C) - 1 \right)^2 \geq d^{\pi_2}(C) > 0$.

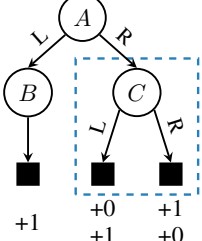

Figure 1: The agent starts from $A$. Action $L$ leads to $B$, from where the agent collects +1 reward. Action $R$ leads to $C$, from where only one action leads to a +1 reward. Nature decides which MDP is presented to the learner. Data distribution is $\mu(A, L) = 1/4, \mu(A, R) = 1/2, \mu(B) = 1/4, \mu(C) = 0$, which satisfies $\pi_{w_1}$-concentrability.

## 6 DISCUSSION

We present a set of practical and statistically optimal algorithms for offline MAB, CB, and RL, under general function approximation and single-policy concentrability. Our algorithms are designed within the MIS formulation combined with a novel application of augmented Lagrangian method. Importantly, our optimality guarantees hold under MIS combined with ALM alone, without any additional form of conservatism such as via regularization or uncertainty quantification. Furthermore, we investigate the role of regularizers in MIS algorithms. Although the empirical benefits of such regularizers are often attributed to conservatism, our analysis suggests that conservatism stems from the MIS formulation while the role of regularizers is to ensure the validity of learned occupancy. Interesting future directions include conducting empirical evaluations of ALM, examining the possibility of removing strong realizability assumptions, and investigating practical and optimal offline RL algorithms whose guarantees hold under milder variants of single-policy concentrability more suited to function approximation.

ACKNOWLEDGMENTS

The authors are grateful to Amy Zhang and Yuandong Tian. This work occurred under Meta AI-BAIR Commons at the University of California, Berkeley, and is partially supported by NSF Grants IIS-1901252 and CCF-1909499. PR is supported by the Open Philanthropy Foundation. Part of the work was done when HZ was a visiting researcher at Meta.

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

## A   RESULTS SUMMARY

Table 1: Summary of suboptimality bounds on the MIS objective with different added terms.

| Setting | Algorithm | Added term | Concentrability | Suboptimality |
|---|---|---|---|---|
| MAB | Unregularized MIS | — | single-policy $\pi^\star$ | $\Omega(1)$ (Proposition 1) |
| | MIS + behavior reg. (Algorithm 5) | $-\alpha\mathbb{E}_\mu[f(w(a))]$ $(\alpha \asymp 1/\sqrt{N})$ | two-policy $\pi^\star, \pi_\alpha^\star$ | $O\left(\frac{1}{\sqrt{N}}\right)$ (Theorem 1) |
| | MIS + ALM (Algorithm 1) | $(\mathbb{E}_\mu[w(a)]-1)^2$ | single-policy $\pi^\star$ | $O\left(\frac{1}{\sqrt{N}}\right)$ (Theorem 2) |
| CB | MIS + behavior reg. (Algorithm 6) | $-\alpha\mathbb{E}_\mu[f(w(s,a))]$ $(\alpha \geq 0)$ | two-policy $\pi^\star, \pi_\alpha^\star$ | $\Omega(N^{\beta\geq-\frac{1}{2}})$ (Proposition 3) |
| | MIS + ALM (Algorithm 2) | $\mathbb{E}_\mu(\sum_a w(s,a)\mu(a|s)-1)^2$ | single-policy $\pi^\star$ | $O\left(\frac{1}{\sqrt{N}}\right)$ (Theorem 3) |
| MDP | MIS + ALM (Algorithms 3, 4) | $\mathbb{E}_{d^{\pi_w}}\left[\left(\frac{d_w(s)}{d^{\pi_w}(s)}-1\right)^2\right]$ | single-policy $\pi^\star$ | $O\left(\frac{1}{\sqrt{N}}\right)$ (Theorem 4) |

## B   RELATED WORK

We covered a number of related works in the introduction and throughout the paper. In this section, we review more related literature.

### B.1   CONCENTRABILITY ASSUMPTIONS

The lack of sufficient coverage in the offline dataset is one of the main challenges in offline RL. In RL theory, dataset coverage has often been characterized by concentrability definitions (Munos, 2007; Scherrer, 2014). Earlier works on offline RL impose all-policy concentrability on the density ratio for all states and actions (Scherrer, 2014; Liu et al., 2019a; Chen & Jiang, 2019; Jiang, 2019; Wang et al., 2019; Liao et al., 2020; Zhang et al., 2020a), with some requiring this ratio to be bounded for every time step (Szepesvári & Munos, 2005; Munos, 2007; Antos et al., 2008; Farahmand et al., 2010; Antos et al., 2007). The works Xie & Jiang (2021); Feng et al. (2019); Uehara et al. (2020) use slightly milder definitions, such as requiring a bound on a weighted norm of density ratios. The work Xie & Jiang (2021) makes even stronger assumptions such as lower bounded conditionals $\mu(a|s)$ and exploratoriness of state marginals to circumvent the Bellman completeness requirement.

To handle partial coverage, recent algorithms are analyzed based on variants of single-policy concentrability (Rashidinejad et al., 2021). Some variants such as the ones presented in works Uehara & Sun (2021) (model-based) or Xie et al. (2021); Song et al. (2022) (model-free) are more suited to function approximation as they avoid bounded ratio assumption for all states and actions. However, existing offline RL algorithms based on these weaker definitions are either computationally intractable (Uehara & Sun, 2021; Xie et al., 2021) or their statistical rate is suboptimal (Cheng et al., 2022). The most related works are Zhan et al. (2022), which requires two-policy concentrability, and Chen & Jiang (2022), which requires single-policy concentrability on density ratio for all states and actions.

### B.2   CONSERVATIVE OFFLINE RL

A series of recent works on offline RL have focused on addressing partial coverage of offline dataset through conservative algorithm design. Broadly speaking, these methods can be broken down into several categories. The first category of methods applies policy constraints, enforcing the learned policy to be close to the behavior policy. Such constraints are applied either explicitly (Fujimoto et al., 2019; Ghasemipour et al., 2020; Jaques et al., 2019; Siegel et al., 2020; Kumar et al., 2019; Wu et al., 2019; Fujimoto & Gu, 2021), implicitly (Peng et al., 2019; Nair et al., 2020), or through importance sampling (Liu et al., 2019b; Swaminathan & Joachims, 2015; Nachum et al., 2019b; Lee et al., 2021; Zhang et al., 2020c;b). Another category involves learning conservative values

such as conservative Q-learning (Kumar et al., 2020), fitted Q-iteration with conservative update (Liu et al., 2020), subtracting penalties (Rezaeifar et al., 2022), and critic regularization (Kostrikov et al., 2021). The last category includes model-based methods such as learning pessimistic models (Kidambi et al., 2020; Guo et al., 2022), adversarial model learning (Rigter et al., 2022), forming penalties using model ensembles (Yu et al., 2020), or incorporating a combination of model and values (Yu et al., 2021).

On the theoretical side, as discussed in the introduction, the majority of works design pessimistic offline RL algorithms that rely on some form of uncertainty quantification (Yin & Wang, 2021; Uehara et al., 2021; Zhang et al., 2022; Yan et al., 2022; Yin et al., 2022; Kumar et al., 2021; Shi & Chi, 2022; Wang et al., 2022). One exception is the work of Zanette et al. (2021) that uses value-function perturbation with actor-critic in linear function approximation setting. Other examples include the recent theoretical works on MIS (Zhan et al., 2022; Chen & Jiang, 2022) and adversarially trained actor-critic (Cheng et al., 2022).

Most related to our work are methods that focus on provable conservative offline RL under general function approximation and partial coverage, summarized under the pessimistic algorithms segment of Table B.2. Uehara & Sun (2021) propose a pessimistic model-based algorithm that under a generalization of single-policy concentrability to bounded TV distance ratio, enjoys a $1/\sqrt{N}$ rate but is computationally intractable. The work of Xie et al. (2021) presents a pessimistic model-free algorithm under a variant of single-policy concentrability framework that requires a bounded ratio of average Bellman error and Bellman completeness. While the original version of the algorithm achieves the optimal $1/\sqrt{N}$ rate, it is computationally intractable. A practical version of the algorithm is presented and has a suboptimal $1/N^{1/5}$ guarantee. Another related work by Chen & Jiang (2022) studies MIS combined with value function approximation under $\pi^\star$-concentrability and proves a $1/\sqrt{\text{gap}(Q^\star)N}$ rate, yet the guarantee degrades with $Q^\star$ gap and the algorithm is computationally intractable. Cheng et al. (2022) propose an adversarially trained actor-critic method that enjoys provable $1/N^{1/3}$ rate under the single-policy concentrability definition of Xie et al. (2021) and Bellman completeness and performs well in offline RL benchmarks when combined with deep neural networks.

Table 2: Comparison of provable offline RL algorithms with general function approximation.

| Algorithm | Computation | Uncertainty | Assumption | Coverage | Unbiased | Suboptimality |
|---|---|---|---|---|---|---|
| **Uniform coverage algorithms** | | | | | | |
| Munos & Szepesvári (2008) | Efficient | N/A | Completeness | all-policy | Yes | $O\left(\frac{1}{\sqrt{N}}\right)$ |
| Antos et al. (2008) | Efficient | N/A | Completeness | all-policy | Yes | $O\left(\frac{1}{\sqrt{N}}\right)$ |
| **Pessimistic algorithms** | | | | | | |
| Xie et al. (2021) | Intractable | Required | Completeness | single-policy | Yes | $O\left(\frac{1}{\sqrt{N}}\right)$ |
| Uehara & Sun (2021) | Intractable | Required | Completeness | single-policy | Yes | $O\left(\frac{1}{\sqrt{N}}\right)$ |
| Chen & Jiang (2022) | Intractable | Required | Realizability only | single-policy | Yes | $O\left(\frac{1}{\sqrt{N}\text{gap}(Q^\star)}\right)$ |
| Zhan et al. (2022) | Efficient | No | Realizability only | two-policy | No | $O\left(\frac{1}{N^{1/6}}\right)$ |
| Cheng et al. (2022) | Efficient | No | Completeness | single-policy | No | $O\left(\frac{1}{N^{1/3}}\right)$ |
| ALMIS (this work) | Efficient | No | Completeness | single-policy | Yes | $O\left(\frac{1}{\sqrt{N}}\right)$ |

### B.3 OTHER TOPICS

Apart from RL, our work on bandits is related to the selection problem (Hong et al., 2021), though the majority of works in this area are in the online setting. Additionally, in our analysis, we solve a subset of stochastic optimization problems with possibly large or infinite stochastic constraints involving conditional expectations. To our knowledge, finite-sample properties of such stochastic optimization problems have not been addressed (Shapiro et al., 2021) and our work may open up avenues for further research in this area.

## C SUPPLEMENTARY MATERIALS FOR MULTI-ARMED BANDITS

We start by presenting a pseudocode for the behavior regularized MIS algorithm in Appendix C.1. In Appendix C.2, we characterize the bias caused by adding the behavior regularization in the primal-dual objective (2). In Appendix C.3, we prove Proposition 1 that demonstrates the failure of unregularized MIS for solving offline MABs, even when the optimal solutions are realizable and an optimal policy is covered in the offline data. Appendix C.4 is devoted to the proof of Theorem 1, which gives a tight performance upper bound of the PRO-MAB algorithm. Finally in Appendix C.5, we prove Proposition 2, showing that constraint satisfaction is sufficient for the success of unregularized MIS.

### C.1 PRIMAL-DUAL REGULARIZED OFFLINE MULTI-ARMED BANDITS (PRO-MAB)

---
**Algorithm 5** Primal-dual Regularized Offline Multi-Armed Bandits (PRO-MAB)
---
1: **Inputs:** Dataset $\mathcal{D} = \{(a_i, r_i)\}_{i=1}^{N}$, classes $\mathcal{V} = [-B_v, B_v]$ and $\mathcal{W}$, function $f(\cdot)$, parameter $\alpha$.
2: Find a solution $\hat{w}, \hat{v}$ to the following problem

$$\max_{w \in \mathcal{W}} \min_{v \in \mathcal{V}} \hat{L}_\alpha^{\mathrm{MAB}}(w, v) := \frac{1}{N} \sum_{i=1}^{N} w(a_i) r_i - \alpha f(w(a_i)) - v(w(a_i) - 1). \qquad (16)$$

3: **Return:** $\hat{\pi}(a) = \frac{\hat{w}(a)\mu(a)}{\sum_a \hat{w}(a)\mu(a)}$ if $\sum_a \hat{w}(a)\mu(a) > 0$, and $\frac{1}{|\mathcal{A}|}$ otherwise.

---

### C.2 SOLUTIONS TO THE PRIMAL-DUAL REGULARIZED OBJECTIVE

In the following lemma, we characterize the optimal solution $(w_\alpha^\star, v_\alpha^\star)$ to the behavior-regularized population objective (2) as well as the suboptimality of the policy induced by $w_\alpha^\star$.

**Lemma 1 (Regularized primal-dual solutions, MAB)** *Let $f$ be differentiable, strictly convex, nonnegative, and bounded by $B_f$. Denote $r^\star := \max_{a \in \mathcal{A}} r(a)$. Then, the following statements hold:*

(I) *$w_0^\star = w^\star$, where $w^\star$ is the importance weight corresponding to an optimal policy;*

(II) *$v_\alpha^\star = r^\star - c\alpha$, where $0 \le c \le f'(C^\star)$;*

(III) *policy $\pi_\alpha^\star := \pi_{w_\alpha^\star}$ satisfies $J(\pi^\star) - J(\pi_\alpha^\star) \le \alpha B_f$.*

*Proof.* Part (I) follows directly by strong duality. For part (II), notice that KKT conditions imply the following relation between $w_\alpha^\star(a)$ and $v_\alpha^\star$:

$$w_\alpha^\star(a) = \max\left\{0, (f')^{-1}\left(\frac{r(a) - v_\alpha^\star}{\alpha}\right)\right\}.$$

Since $f$ is strictly convex, $f'$ is a monotonically increasing function. Therefore, the optimal arm $a^\star$ has the largest $w_\alpha^\star(a)$, which should be nonzero due to realizability of $w_\alpha^\star$. In other words,

$$w_\alpha^\star(a^\star) = (f')^{-1}\left(\frac{r^\star - v_\alpha^\star(s)}{\alpha}\right) \Rightarrow v_\alpha^\star = r^\star - \alpha f'(w_\alpha^\star(a^\star)). \qquad (17)$$

We now proceed to find a bound on $f'(w_\alpha^\star(a^\star))$. Since $w_\alpha^\star$ is the optimal solution to (2), it must satisfy the constraint

$$\sum_{a \in \mathcal{A}} \mu(a) w_\alpha^\star(a) = 1 \Rightarrow w_\alpha^\star(a^\star) \le \frac{1}{\mu(a^\star)} \le C^\star,$$

where the last inequality stems from the single-policy concentrability assumption of $\pi^*$. Since $f'$ is an increasing function, we have $f'(w_\alpha^\star(a^\star)) \le f'(C^\star)$, which combined with (17) yields the following lower bound on $v_\alpha^\star$

$$v_\alpha^\star \ge r^\star - \alpha f'(C^\star).$$

Moreover, the convexity of $f$ immediately gives the upper bound on $v_\alpha^\star \leq r^\star$, which completes the proof of part (II).

We now prove the last part. Since $w_\alpha^\star$ is the optimal solution to the regularized population objective (2), by strong duality, we have

$$\mathbb{E}_{a \sim d_{w_\alpha^\star}}[r(a)] - \alpha \mathbb{E}_{a \sim \mu}[f(w_\alpha^\star(a))] \geq \mathbb{E}_{a \sim d^\star}[r(a)] - \alpha \mathbb{E}_{a \sim \mu}[f(w^\star(a))]$$

where $d_{w_\alpha^\star}(a) = \mu(a) w_\alpha^\star(a)$ by definition given in Section 2 and we used the fact that $\mathbb{E}_{a \sim \mu}[w_\alpha^\star(a)] - 1 = \mathbb{E}_{a \sim \mu}[w^\star(a)] - 1 = 0$. Therefore, the suboptimality of $\pi_\alpha^\star$ can be bounded as follows

$$
\begin{aligned}
J(\pi^\star) - J(\pi_\alpha^\star) &= \mathbb{E}_{a \sim d^\star}[r(a)] - \mathbb{E}_{a \sim d_{w_\alpha^\star}}[r(a)] \\
&\leq \alpha \mathbb{E}_{a \sim \mu}[f(w^\star(a))] - \alpha \mathbb{E}_{a \sim \mu}[f(w_\alpha^\star(a))] \\
&\leq \alpha \mathbb{E}_{a \sim \mu}[f(w^\star(a))] \leq \alpha f(C^\star) \leq \alpha B_f,
\end{aligned}
$$

where in the second to last inequality we used the non-negativity of $f$ and in the last equality, we used the boundedness of $f$. $\qquad\square$

### C.3 PROOF OF PROPOSITION 1

Consider a 2-armed bandit instance with the following reward distributions, data distribution, and function classes.

- *Reward distributions:* The first arm is optimal with deterministic reward and the second arm has a Bernoulli distribution:

$$r(1) = \frac{1}{2} \quad \text{w.p. 1}, \qquad r(2) \sim \text{Bernoulli}(1/3).$$

- *Data distribution:* We consider a scenario where most data are concentrated on the optimal arm:

$$\mu(1) = 1 - \frac{2}{N}, \qquad \mu(2) = \frac{2}{N}.$$

Here, the single-policy concentrability coefficient is $C^\star = 1/\mu(1)$ and is finite for $N > 2$. Let $N(a)$ denote the number of samples on arm $a$. To obtain upper and lower bounds on $N(a)$, we resort to the following lemma, which is a direct consequence of the Chernoff bound for binomial variables.

**Lemma 2 (Chernoff bounds, binomial)**

(I) *With probability at least $1 - \exp(-N\mu(a)\delta_u^2/(2 + \delta_u))$, one has $N(a) \leq (1 + \delta_u)N\mu(a)$ for any $\delta_u > 0$;*

(II) *With probability at least $1 - \exp(-N\mu(a)\delta_l^2/2)$, one has $(1 - \delta_l)N\mu(a) \leq N(a)$ for any $0 < \delta_l < 1$.*

We condition on the event that the number of samples on the second arm is between 1 and 5 which occurs with probability larger than $1 - \exp\left(-2 \cdot \frac{0.9^2}{2}\right) - \exp\left(-2 \cdot \frac{1.95^2}{1.95+2}\right) \geq 0.4$ due to Lemma 2 when setting $\delta_l = 0.9$ and $\delta_u = 1.95$:

$$(1 - 0.9) \cdot N\mu(2) \leq N(2) \leq (1 + 1.95) \cdot N\mu(2) \quad \Rightarrow \quad 1 \leq N(2) \leq 5.$$

- *Function classes:* Assume that $\mathcal{W} = \{w_1 = (C^\star, 0), w_2 = (0, B_w)\}$ and $\mathcal{V} = \{1/2\}$. By Lemma 1, we have $v_0^\star = r^\star = 1/2$. Therefore, the problem is realizable as $v_0^\star \in \mathcal{V}$ and $w_0^\star = w^\star = (C^\star, 0) \in \mathcal{W}$. Furthermore, notice that for the second candidate $w_2 = (0, B_w) \in \mathcal{W}$, the normalization factor is small for a constant $B_w$ as $d_{w_2} = \sum_a w_2(a)\mu(a) = 2B_w/N$.

Consider the case where all $N(a)$ samples on the second arm observe a reward of 1, which happens with a probability of at least $\frac{1}{3^5}$ as we conditioned on the event that $1 \leq N(2) \leq 5$. We now compute

$\hat{w}$ by solving the empirical objective (16) with $\alpha = 0$. Note that since $|\mathcal{V}| = 1$, it suffices to compute $\hat{w} = \arg\max_{w \in \mathcal{W}} \hat{L}_0^{\text{MAB}}(w, v = 1/2)$. We have

$$\hat{L}_0^{\text{MAB}}(w_1, 1/2) = \frac{N(1)}{N}\left[C^\star \cdot \frac{1}{2} - \frac{1}{2}(C^\star - 1)\right] + \frac{N(2)}{2N} = \frac{1}{2}$$

$$\hat{L}_0^{\text{MAB}}(w_2, 1/2) = \frac{N(1)}{2N} + \frac{N(2)}{N}\left[B_w - \frac{1}{2}(B_w - 1)\right] = \frac{1}{2} + \frac{N(2)B_w}{2N}$$

Since we conditioned on the event with $N(2) \geq 1$, solving the optimization problem $\max_{w \in \mathcal{W}} \hat{L}_0^{\text{MAB}}(w, v = 1/2)$ finds $\hat{w} = (0, B_w)$, leading to a policy that picks the second arm with probability one. Therefore, with constant probability of $0.4 \times 1/3^5 > 0.001$, we have

$$J(\pi^\star) - J(\hat{\pi}) = \frac{1}{2} - \frac{1}{3} = \frac{1}{6}.$$

### C.4   PROOF OF THEOREM 1

Before embarking on the main proof, we present two lemmas related to the primal-dual regularized approach. The first lemma shows the closeness of population objective (2) to its empirical approximation used in Algorithm 1, which is a direct consequence of Hoeffding's inequality. We also show that closeness of objectives results in the closeness of $w_\alpha^\star$ and $\hat{w}$, which are respectively the optimums to (2) and (16). The proof of this lemma is deferred to the end of this subsection.

**Lemma 3 (Empirical and population closeness, PRO-MAB)** *Fix $\delta > 0$ and define*

$$\epsilon_{stat,\alpha}^{MAB} := ((B_w + 1)(B_v + 1) + \alpha B_f)\sqrt{\frac{\log|\mathcal{V}||\mathcal{W}|/\delta}{N}}. \tag{18}$$

*For any $w \in \mathcal{W}$ and $v \in \mathcal{V}$, the following bounds hold with probability at least $1 - \delta$*

*(I)* $|L_\alpha^{MAB}(w, v) - \hat{L}_\alpha^{MAB}(w, v)| \leq \epsilon_{stat,\alpha}^{MAB}$;

*(II)* $L_\alpha^{MAB}(w_\alpha^\star, v) - L_\alpha^{MAB}(\hat{w}, v) \leq 2\epsilon_{stat,\alpha}^{MAB}$.

The second lemma finds a lower bound on the occupancy normalization factor $d_{\hat{w}} = \sum_a \hat{w}(a)\mu(a)$ enforced by the behavior regularization.

**Lemma 4 (Occupancy validity enforced by behavior regularization)** *Let $f$ be an $M_f$-strongly-convex function and fix $\delta > 0$. Then, with probability at least $1 - \delta$, one has*

$$d_{\hat{w}} \geq 1 - \sqrt{\frac{4\epsilon_{stat,\alpha}^{MAB}}{\alpha M_f}},$$

*where $\epsilon_{stat,\alpha}^{MAB}$ is defined in (18).*

For the rest of this proof, we condition on the high probability events of Lemmas 3 and 4. Define

$$\epsilon_{\hat{w},r} := \sum_a w_\alpha^\star(a)\mu(a)r(a) - \hat{w}(a)\mu(a)r(a). \tag{19}$$

By part (II) of Lemma 3, we have $L_\alpha^{\text{MAB}}(v_\alpha^\star, w_\alpha^\star) - L_\alpha^{\text{MAB}}(v_\alpha^\star, \hat{w}) \leq 2\epsilon_{stat,\alpha}^{\text{MAB}}$. Therefore,

$$\epsilon_{\hat{w},r} - \alpha\mathbb{E}_\mu[f(w_\alpha^\star(a)) - f(\hat{w}(a))] + v_\alpha^\star(d_{\hat{w}} - 1) \leq 2\epsilon_{stat,\alpha}^{\text{MAB}}. \tag{20}$$

Recall from Lemma 1 that we have $v_\alpha^\star = r^\star - \alpha c$, where $c \leq f'(C^\star)$. Thus, combined with (20), we write

$$\epsilon_{\hat{w},r} + r^\star(d_{\hat{w}} - 1) \leq 2\epsilon_{stat,\alpha}^{\text{MAB}} + \alpha\mathbb{E}_\mu[f(w_\alpha^\star(a)) - f(\hat{w}(a))] + \alpha c(d_{\hat{w}} - 1)$$
$$\leq 2\epsilon_{stat,\alpha}^{\text{MAB}} + \alpha(2B_f + \alpha f'(C^\star)B_w), \tag{21}$$

where in the second line we used the bounds $|f(x)| \leq B_f$ and $d_{\hat{w}} \leq B_w$. Note that setting $\alpha = 16\epsilon_{stat,1}^{\text{MAB}}/M_f$, Lemma 4 asserts that $d_{\hat{w}} \geq 1/2$. Since $d_{\hat{w}} \geq 1/2$, the learned policy is written as

$\hat{\pi} = \hat{w}(a)\mu(a)/d_{\hat{w}}$. With simple algebraic manipulations, we find the following expression for the suboptimality of $\hat{\pi}$ with respect to $\pi_\alpha^\star$:

$$
\begin{aligned}
J(\pi_\alpha^\star) - J(\hat{\pi}) &= \sum_a w_\alpha^\star(a)\mu(a)r(a) - \frac{1}{d_{\hat{w}}}\hat{w}(a)\mu(a)r(a) \\
&= \sum_a w_\alpha^\star(a)\mu(a)r(a) - \hat{w}(a)\mu(a)r(a) + \sum_a \left(1 - \frac{1}{d_{\hat{w}}}\right)\hat{w}(a)\mu(a)r(a) \\
&= \epsilon_{\hat{w},r} + (d_{\hat{w}} - 1)\sum_a \frac{1}{d_{\hat{w}}}\hat{w}(a)\mu(a)r(a) \\
&= \epsilon_{\hat{w},r} + (d_{\hat{w}} - 1)J(\hat{\pi}) \\
&= \epsilon_{\hat{w},r} + (d_{\hat{w}} - 1)J(\pi_\alpha^\star) - (d_{\hat{w}} - 1)\left[J(\pi_\alpha^\star) - J(\hat{\pi})\right].
\end{aligned}
$$

Let $\epsilon_{\mathrm{reg}} = J(\pi^\star) - J(\pi_\alpha^\star) = r^\star - J(\pi_\alpha^\star)$ denote the suboptimality suffered due to behavior regularization. Suboptimality $J(\pi_\alpha^\star) - J(\hat{\pi})$ can be expressed as

$$
\begin{aligned}
J(\pi_\alpha^\star) - J(\hat{\pi}) &= \frac{1}{d_{\hat{w}}}\left(\epsilon_{\hat{w},r} + (d_{\hat{w}} - 1)J(\pi_\alpha^\star)\right) \\
&= \frac{1}{\hat{d}}\left(\epsilon_{\hat{w},r} + (d_{\hat{w}} - 1)(r^\star - \epsilon_{\mathrm{reg}})\right) \\
&\leq \frac{1}{d_{\hat{w}}}\left(\epsilon_{\hat{w},r} + (d_{\hat{w}} - 1)r^\star\right) - \frac{1}{d_{\hat{w}}}(d_{\hat{w}} - 1)\epsilon_{\mathrm{reg}}.
\end{aligned}
$$

We use the above inequality to bound the suboptimality with respect to an optimal policy:

$$
\begin{aligned}
J(\pi^\star) - J(\hat{\pi}) &= J(\pi^\star) - J(\pi_\alpha^\star) + J(\pi_\alpha^\star) - J(\hat{\pi}) \\
&= \epsilon_{\mathrm{reg}} + J(\pi_\alpha^\star) - J(\hat{\pi}) \\
&\leq \epsilon_{\mathrm{reg}} + \frac{1}{d_{\hat{w}}}\left(\epsilon_{\hat{w},r} + (d_{\hat{w}} - 1)r^\star\right) - \frac{1}{d_{\hat{w}}}(d_{\hat{w}} - 1)\epsilon_{\mathrm{reg}} \\
&\leq \frac{1}{d_{\hat{w}}}\left(\epsilon_{\hat{w},r} + (d_{\hat{w}} - 1)r^\star\right) + \frac{1}{d_{\hat{w}}}\epsilon_{\mathrm{reg}}.
\end{aligned}
$$

Recall that we have $1/d_{\hat{w}} \leq 2$ and that $\epsilon_{\mathrm{reg}}$ is bounded by $\alpha B_f$ by Lemma 1. Therefore,

$$
\begin{aligned}
J(\pi^\star) - J(\hat{\pi}) &\leq \frac{1}{d_{\hat{w}}}\left(\epsilon_{\hat{w},r} + (d_{\hat{w}} - 1)r^\star\right) + \frac{1}{d_{\hat{w}}}\epsilon_{\mathrm{reg}} \\
&\leq 2\left(\epsilon_{\hat{w},r} + (d_{\hat{w}} - 1)r^\star\right) + 2\alpha B_f \\
&\leq 4\epsilon_{\mathrm{stat},\alpha}^{\mathrm{MAB}} + \alpha(4B_f + 2f'(C^\star)B_w)) + 2\alpha B_f \\
&\lesssim \alpha(B_f + f'(C^\star)B_w).
\end{aligned}
$$

where the penultimate inequality relies on the bound derived in (21).

*Proof of Lemma 3.* $\hat{L}_\alpha^{\mathrm{MAB}}(w, v)$ is an empirical average over independent and bounded random variables, where the bound on individual variables is computed as

$$
\begin{aligned}
|w(a_i)r_i - \alpha f(w(a_i)) - v(w(a_i) - 1)| &\leq B_w + \alpha B_f + B_v(B_w + 1) \\
&\leq (B_w + 1)(B_v + 1) + \alpha B_f.
\end{aligned}
$$

It is easy to see that $\mathbb{E}_{\mathcal{D}}[\hat{L}_\alpha^{\mathrm{MAB}}(w, v)] = L_\alpha^{\mathrm{MAB}}(w, v)$, where the expectation is taken with respect to the randomness in dataset $\mathcal{D}$. Part (I) of this lemma is proved by applying Hoeffding's inequality along with a union bound on $w$ and $v$.

The proof of part (II) is similar to Lemma 7 of Zhan et al. (2022) and relies on decomposing the objective difference and using the fact that $(\hat{w}, \hat{v})$ correspond to the saddle points of $L_\alpha^{\mathrm{MAB}}$ and $\hat{L}_\alpha^{\mathrm{MAB}}$. For any $w \in \mathcal{W}$, define

$$
\hat{v}_w = \arg\min_{v \in \mathcal{V}} \hat{L}_\alpha^{\mathrm{MAB}}(w, v) \tag{22}
$$

We write

$$
\begin{aligned}
L_\alpha^{\mathrm{MAB}}(w_\alpha^\star, v) - L_\alpha^{\mathrm{MAB}}(\hat{w}, v) = {} & \underbrace{L_\alpha^{\mathrm{MAB}}(w_\alpha^\star, v) - L_\alpha^{\mathrm{MAB}}(w_\alpha^\star, \hat{v}_{w_\alpha^\star})}_{:=T_1} + \underbrace{L_\alpha^{\mathrm{MAB}}(w_\alpha^\star, \hat{v}_{w_\alpha^\star}) - \hat{L}_\alpha^{\mathrm{MAB}}(w_\alpha^\star, \hat{v}_{w_\alpha^\star})}_{:=T_2} \\
& + \underbrace{\hat{L}_\alpha^{\mathrm{MAB}}(w_\alpha^\star, \hat{v}_{w_\alpha^\star}) - \hat{L}_\alpha^{\mathrm{MAB}}(\hat{w}, \hat{v})}_{:=T_3} + \underbrace{\hat{L}_\alpha^{\mathrm{MAB}}(\hat{w}, \hat{v}) - \hat{L}_\alpha^{\mathrm{MAB}}(\hat{w}, v)}_{:=T_4} \\
& + \underbrace{\hat{L}_\alpha^{\mathrm{MAB}}(\hat{w}, v) - L_\alpha^{\mathrm{MAB}}(\hat{w}, v)}_{:=T_5},
\end{aligned}
$$

Each term is bounded as follows:

- $T_1 = 0$ because $w_\alpha^\star$ satisfies the constraint $\sum_a w_\alpha^\star(a)\mu(a) = 1$ and for any $v_1, v_2$ we have $L_\alpha^{\mathrm{MAB}}(w_\alpha^\star, v_1) = L_\alpha^{\mathrm{MAB}}(w_\alpha^\star, v_2)$.
- $T_2 \le \epsilon_{\mathrm{stat}}$ due to Lemma 3.
- $T_3 \le 0$ because $\hat{w} = \arg\max_{w\in\mathcal{W}} \hat{L}_\alpha(\hat{v}_w, w)$.
- $T_4 \le 0$ because $\hat{v} = \arg\min_{v\in\mathcal{V}} \hat{L}_\alpha^{\mathrm{MAB}}(v, \hat{w})$.
- $T_5 \le \epsilon_{\mathrm{stat}}$ due to Lemma 3.

Summing up the bounds on each term yields the desired bound. $\qquad\square$

*Proof of Lemma 4.* This lemma is a direct consequence of Lemma 8 in Zhan et al. (2022). For completeness, we present a simplified proof for the multi-armed bandit setting.

First, observe that since $f$ is $M_f$-strongly-convex, the function $L_\alpha^{\mathrm{MAB}}(v_\alpha^\star, w)$ is $\alpha M_f$-strongly-concave with respect to $w$ and norm $\|\cdot\|_{2,\mu}$. Furthermore, since $w_\alpha^\star = \arg\max_w L_\alpha^{\mathrm{MAB}}(v^\star, w)$, we have

$$
\|\hat{w} - w_\alpha^\star\|_{2,\mu} \le \sqrt{\frac{2(L_\alpha^{\mathrm{MAB}}(w_\alpha^\star, v_\alpha^\star) - L_\alpha^{\mathrm{MAB}}(\hat{w}, v_\alpha^\star))}{\alpha M_f}}.
$$

The above bound along with the bound on $L_\alpha^{\mathrm{MAB}}(w_\alpha^\star, v_\alpha^\star) - L_\alpha^{\mathrm{MAB}}(\hat{w}, v_\alpha^\star) \le 2\epsilon_{\mathrm{stat},\alpha}^{\mathrm{MAB}}$ showed in Lemma 1, give the following bound on $|d_{\hat{w}} - 1|$

$$
|d_{\hat{w}} - 1| = \left| \sum_a \hat{w}(a)\mu(a) - \sum_a w_\alpha^\star(a)\mu(a) \right| \le \|\hat{w} - w_\alpha^\star\|_{1,\mu} \le \|\hat{w} - w_\alpha^\star\|_{2,\mu} \le \sqrt{\frac{4\epsilon_{\mathrm{stat},\alpha}^{\mathrm{MAB}}}{\alpha M_f}},
$$

which completes the proof. $\qquad\square$

### C.5 PROOF OF PROPOSITION 2

Consider the difference between population objective with $\alpha = 0$ at $w_0^\star = w^\star$ and $\hat{w}$, which is bounded by Lemma 3:

$$
L(w^\star, v^\star) - L(\hat{w}, v^\star) = \mathbb{E}_{a\sim\mu}[r(a)(w^\star(a) - \hat{w}(a))] - v^\star \mathbb{E}_{a\sim\mu}[w^\star(a) - \hat{w}(a)] \lesssim \epsilon_{\mathrm{stat},\alpha}^{\mathrm{MAB}}. \tag{23}
$$

We have $\mathbb{E}_{a\sim\mu}[w^\star(a)] = 1$ due to realizability and $\mathbb{E}_{a\sim\mu}[\hat{w}(a)] = 1$ is our assumption. Thus the second term in (23) is zero. Moreover, note that $\hat{\pi}(a) = \hat{w}(a)\mu(a)/\mathbb{E}_{a\sim\mu}[\hat{w}(a)] = \hat{w}(a)$. Substituting the expression for $\epsilon_{\mathrm{stat},\alpha}^{\mathrm{MAB}}$ from (18) with $\alpha = 0$, we obtain

$$
J(\pi^\star) - J(\hat{\pi}) = \mathbb{E}_{a\sim\mu}[r(a)(w^\star(a) - \hat{w}(a))] \lesssim B_w(B_v + 1)\sqrt{\frac{\log|\mathcal{V}||\mathcal{W}|/\delta}{N}},
$$

where we used the fact that $B_w \asymp B_w + 1$ since $B_w \ge 1$ due to realizability of $w^\star$.

## D PROOFS FOR CONTEXTUAL BANDITS

This section of the appendix is organized as follows. In Appendix D.1, we present details of the PRO-CB algorithm. Appendix D.2 is devoted to the proof of Proposition 1, which shows that the PRO-CB algorithm fails to achieve the statistically optimal rate of $1/\sqrt{N}$. The proof of suboptimality upper bound for the conservative offline CB algorithm with ALM is presented in Theorem 3.

## D.1 PRIMAL-DUAL REGULARIZED OFFLINE CONTEXTUAL BANDITS (PRO-CB)

Define importance weights $w(s, a) = d(s, a)/\mu(s, a)$ to denote the ratio of occupancy and data distribution. The primal-dual regularized approach (Zhan et al., 2022) solves the following population objective

$$\max_{w \geq 0} \min_{v} L_\alpha^{\text{CB}}(w, v) := \mathbb{E}_{s,a\sim\mu}\left[w(s, a)r(s, a)\right] - \mathbb{E}_{s,a\sim\mu}[v(s)(w(s, a) - 1)] - \alpha\mathbb{E}_{s,a\sim\mu}\left[f\left(w(s, a)\right)\right], \tag{24}$$

The above optimization problem satisfies strong duality. We define $w_\alpha^\star, v_\alpha^\star$ to respectively denote the optimal solutions to the primal and dual variables. Approximating $w, v$ to belong to function classes $\mathcal{W}, \mathcal{V}$ and solving the empirical version of objective (24) leads to the PRO-CB given in Algorithm 6.

---

**Algorithm 6** Primal-dual Regularized Offline Contextual Bandits (PRO-CB)

1: **Inputs:** Dataset $\mathcal{D} = \{(s_i, a_i, r_i)\}_{i=1}^N$, function classes $\mathcal{W}, \mathcal{V}$, function $f(\cdot)$, parameter $\alpha$
2: Find a solution $\hat{w}, \hat{v}$ to the following problem

$$\max_{w \in \mathcal{W}} \min_{v \in \mathcal{V}} \hat{L}_\alpha^{\text{CB}}(w, v) := \frac{1}{N} \sum_{i=1}^N w(s_i, a_i)r_i - \alpha f(w(s_i, a_i)) - v(s_i)(w(s_i, a_i) - 1). \tag{25}$$

3: **Return:** $\hat{\pi} = \pi_{\hat{w}}$.

---

## D.2 PROOF OF PROPOSITION 3

We separate the proof into two cases: $\alpha \geq N^\beta$ for $\beta > -1/2$ and $\alpha \leq \widetilde{O}(N^{-1/2})$. When $\alpha$ is large, we show that the large bias caused by regularization results in suboptimality of $\alpha$ even in MABs. When $\alpha$ is small, we construct a two-state CB instance (as the single-state case is indeed successful due to Theorem 1), showing that such small $\alpha$ does not sufficiently enforce occupancy validity in states with a relatively small but still significant state distribution $\rho(s)$.

### D.2.1 PROOF FOR LARGE $\alpha$

If there exists $-\frac{1}{2} < \beta$ such that $\alpha \geq N^\beta$, then we consider a simple single-state two-arm contextual bandit (equivalently multi-armed bandit) instance:

- *Reward distribution*: Both arms have deterministic rewards and the suboptimal arm has a value gap of $\alpha$:

$$r(1) = 1 \quad \text{w.p. } 1, \quad r(2) = \max\{0, 1 - \alpha\} \quad \text{w.p. } 1.$$

- *Data distribution*: We construct the data distribution such that both arms have constant probability density, which implies a constant concentrability ratio $C^\star$. Here we assume $M_f < 100$ for convenience, but if $M_f$ is larger we can use the same construction with an even larger constant as the denominator.

$$\mu(1) = \frac{M_f}{100}, \quad \mu(2) = 1 - \frac{M_f}{100}.$$

- *Function classes*: We assume both $\mathcal{W}$ and $\mathcal{V}$ contain only the optimal regularized solutions $(w_\alpha^\star, v_\alpha^\star)$ and the optimal unregularized solutions $(w^\star, v^\star)$, which satisfy the realizability requirements of PRO-CB:

$$\mathcal{W} = \{w_\alpha^\star, w^\star\}, \quad \mathcal{V} = \{v_\alpha^\star, v^\star\}.$$

Our argument is broken down in two steps. In the first step, we show that the suboptimality of the optimal regularized policy, which is the policy induced by the regularized optimal weights $\pi_\alpha^\star := \pi_{w_\alpha^\star}$, is at least of order $\min\{1, \alpha\}$. Then, in the second step, we prove that $w_\alpha^\star$ is chosen with a constant probability.

**Step 1: Suboptimality of $\pi_\alpha^\star$.** In the particular offline bandit instance above, we show the following lower bound on suboptimality of $\pi_\alpha^\star$

$$J(\pi^\star) - J(\pi_\alpha^\star) = \pi_\alpha^\star(2) \cdot (r(1) - r(2)) = \mu(2)w_\alpha^\star(2) \cdot \min\{1, \alpha\} = \Omega(\min\{1, \alpha\}). \tag{26}$$

To establish (26), we show that $w_\alpha^\star(2) > c$ for a fixed constant $c = \frac{1}{2}$. We prove this by contradiction. Suppose

$$w_\alpha^\star(2) \leq c. \tag{27}$$

By KKT conditions we have

$$w_\alpha^\star(2) = \max\left\{0, (f')^{-1}\left(\frac{r(2) - v_\alpha^\star}{\alpha}\right)\right\} \geq (f')^{-1}\left(\frac{r(2) - v_\alpha^\star}{\alpha}\right).$$

Therefore, using the fact that $f'$ is strictly increasing since $f$ is strictly convex, we lower bound $v_\alpha^\star$ according to

$$v_\alpha^\star \geq r(2) - \alpha f'(w_\alpha^\star(2)) \geq r(1) - (r(1) - r(2)) - \alpha f'(c).$$

Combining the above bound on $v_\alpha^\star$ with the KKT condition on $w_\alpha^\star(1)$, we then obtain

$$w_\alpha^\star(1) = (f')^{-1}\left(\frac{r(1) - v_\alpha^\star}{\alpha}\right) \leq (f')^{-1}\left(\frac{r(1) - r(2)}{\alpha} + f'(c)\right). \tag{28}$$

Here, we used the fact that $v_\alpha^\star \geq r^\star = r(1)$ and that $f(0) = 0$ so $(f')^{-1}((r(1) - v_\alpha^\star)/\alpha) \geq 0$. Moreover, since the regularization function $f$ is $M_f$-strongly convex, we write

$$f'\left(\frac{1 - c\mu(2)}{\mu(1)}\right) - f'(c) \geq M_f\left(\frac{1 - c\mu(2)}{\mu(1)} - c\right) = M_f\frac{1 - c}{\mu(1)} = 100(1 - c) > 1,$$

$$\Rightarrow \frac{r(1) - r(2)}{\alpha} + f'(c) \leq 1 + f'(c) < f'\left(\frac{1 - c\mu(2)}{\mu(1)}\right). \tag{29}$$

Therefore, we can continue to upper bound the RHS of (28):

$$w_\alpha^\star(1) \leq (f')^{-1}\left(\frac{r(1) - r(2)}{\alpha} + f'(c)\right) \underbrace{<}_{\text{by (29)}} (f')^{-1}\left(f'\left(\frac{1 - c\mu(2)}{\mu(1)}\right)\right) = \frac{1 - c\mu(2)}{\mu(1)},$$

which further implies that

$$w_\alpha^\star(1)\mu(1) < 1 - c\mu(2) \underbrace{\leq}_{\text{by (27)}} 1 - w_\alpha^\star(2)\mu(2) \Rightarrow \sum_a w_\alpha^\star(a)\mu(a) < 1. \tag{30}$$

Note that (30) contradicts with the fact that $(w_\alpha^\star, v_\alpha^\star)$ is the optimal min-max solution of $L_\alpha^{\text{MAB}}$ because it violates the constraint $\mathbb{E}_\mu[w(a)] = 1$. Therefore, (27) should not hold in the first place, and we must have

$$J(\pi^\star) - J(\pi_\alpha^\star) = \mu(2)w_\alpha^\star(2) \cdot (r(1) - r(2)) > c\left(1 - \frac{M_f}{100}\right)\min\{1, \alpha\} \gtrsim \min\{1, \alpha\} \tag{31}$$

**Step 2: $w_\alpha^\star$ is picked with large probability.** We now show that $w_\alpha^\star$ is picked by the algorithm with at least a constant probability. Note that since $w_\alpha^\star$ and $w^\star$ both satisfy the constraint $\mathbb{E}_\mu[w] - 1 = 0$, objectives $L_\alpha^{\text{MAB}}(w_\alpha^\star, v)$ and $L_\alpha^{\text{MAB}}(w_\alpha, v)$ do not depend on the Lagrange multiplier variable $v$. We argue that at the population level, we have the following lower bound on the gap $L_\alpha^{\text{MAB}}(w_\alpha^\star, v) - L_\alpha^{\text{MAB}}(w^\star, v) \gtrsim \alpha$. Using the definition of $L_\alpha^{\text{MAB}}$, one has

$$L_\alpha^{\text{MAB}}(w_\alpha^\star, \cdot) - L_\alpha^{\text{MAB}}(w^\star, \cdot)$$
$$= \alpha\mathbb{E}_\mu[f(w^\star(a)) - f(w_\alpha^\star(a))] - \mu(2)w_\alpha^\star(2)(r(1) - r(2))$$
$$= \alpha\left(\mu(1)\Big(f(w^\star(1)) - f(w_\alpha^\star(1))\Big) + \mu(2)\Big(f(w^\star(2)) - f(w_\alpha^\star(2))\Big)\right) - \mu(2)w_\alpha^\star(2)(r(1) - r(2))$$
$$\geq \alpha\left(\mu(1)\Big(w^\star(1) - w_\alpha^\star(1)\Big) \cdot f'(w_\alpha^\star(1)) - \mu(2)f(w_\alpha^\star(2))\right) - \mu(2)w_\alpha^\star(2)(r(1) - r(2)) \tag{32}$$
$$= \alpha\mu(2)\left(f'(w_\alpha^\star(1)) \cdot w_\alpha^\star(2) - f(w_\alpha^\star(2)) - w_\alpha^\star(2) \cdot \frac{r(1) - r(2)}{\alpha}\right), \tag{33}$$

In (32), we used the convexity of regularization function $f$ as well as the fact that $f(w^\star(2)) = f(0) = 0$. Moreover, (33) holds because

$$\mu(1)\left(w^\star(1) - w_\alpha^\star(1)\right) = \mu(1)\left(\frac{1}{\mu(1)} - w_\alpha^\star(1)\right) = 1 - \mu(1)w_\alpha^\star(1) = \mu(2)w_\alpha^\star(2).$$

By KKT conditions we also have

$$f'(w_\alpha^\star(1)) = \frac{r(1) - v_\alpha^\star}{\alpha} = \frac{r(1) - r(2) + \alpha f'(w_\alpha^\star(2))}{\alpha} = \frac{r(1) - r(2)}{\alpha} + f'(w_\alpha^\star(2)). \quad (34)$$

Plugging (34) back into (33), we obtain

$$L_\alpha^{\mathrm{MAB}}(w_\alpha^\star, \cdot) - L_\alpha^{\mathrm{MAB}}(w^\star, \cdot) \geq \alpha\mu(2)\Big(f'(w_\alpha^\star(2)) \cdot w_\alpha^\star(2) - f(w_\alpha^\star(2))\Big)$$

$$\geq \alpha\mu(2) \cdot \frac{M_f}{2}w_\alpha^\star(2)^2 > \alpha\mu(2) \cdot \frac{M_f}{2}c^2 \gtrsim \alpha, \quad (35)$$

where (35) is based on the fact that $f$ is $M_f$-strongly convex, and that $w_\alpha^\star(2) > c$ proved in Step 1. We now prove that such large lower bound on population objective difference leads the algorithm to select $w_\alpha^\star$. Recall from Lemma 3 that with at least constant probability (e.g. setting $\delta = 0.1$), for any $v \in \mathcal{V}, w \in \mathcal{W}$, one has the following bound on difference between the population and empirical objectives

$$\left|L_\alpha^{\mathrm{MAB}}(w, v) - \hat{L}_\alpha^{\mathrm{MAB}}(w, v)\right| \lesssim 2\epsilon_{\mathrm{stat},\alpha}^{\mathrm{MAB}},$$

where $\epsilon_{\mathrm{stat},\alpha}^{\mathrm{MAB}}$ is of order $1/\sqrt{N}$ as defined in (18). Combining the above inequality with (35), for any $v, v' \in \mathcal{V}$ we have

$$\hat{L}_\alpha^{\mathrm{MAB}}(w_\alpha^\star, v) - \hat{L}_\alpha^{\mathrm{MAB}}(w^\star, v')$$
$$\gtrsim \alpha - \epsilon_{\mathrm{stat},\alpha}^{\mathrm{MAB}} \gtrsim \alpha - (1 + \alpha)N^{-\frac{1}{2}} \gtrsim N^\beta - N^{-\frac{1}{2}}.$$

Therefore, since $\beta > -1/2$, we conclude that $w_\alpha^\star$ is chosen by the algorithm with constant probability:

$$\min_{v \in \mathcal{V}} \hat{L}_\alpha^{\mathrm{MAB}}(w_\alpha^\star, v) - \min_{v \in \mathcal{V}} \hat{L}_\alpha^{\mathrm{MAB}}(w^\star, v) > 0 \Rightarrow w_\alpha^\star = \underset{w \in \mathcal{W}}{\operatorname{argmax}} \min_{v \in \mathcal{V}} \hat{L}_\alpha^{\mathrm{MAB}}(w, v).$$

Combining the above result with the suboptimality lower bound of $\pi_\alpha^\star$ in (31) completes the proof for $\alpha \geq N^\beta$.

### D.2.2 PROOF FOR SMALL $\alpha$

Now suppose $\alpha \leq \widetilde{O}(N^{-\frac{1}{2}})$, where $\widetilde{O}$ hides the logarithmic factors. In this case, we consider the following two-state two-arm contextual bandit instance:

- *State and reward distributions*: We construct the states such that state 1 has a very small probability mass. For state 1, the first arm is optimal with a Bernoulli-distributed reward and the second arm is suboptimal with a deterministic reward. For state 2, both arms have deterministic rewards. Importantly, state 1 has a constant value gap in its suboptimal action.

$$\rho(1) = N^{-\frac{1}{4}}, \quad r(1,1) \sim \mathrm{Bernoulli}\left(\frac{1}{2}\right), \ r(1,2) \equiv \frac{1}{3};$$

$$\rho(2) = 1 - N^{-\frac{1}{4}}, \quad r(2,1) \equiv \frac{1}{2}, \ r(2,2) \equiv \frac{1}{3}.$$

- *Data distribution*: We assume that for both states, most of the probability density is concentrated on the optimal arm.

$$\mu(s) = \rho(s), \ s = 1, 2.$$

$$\mu(1|1) = \mu(1|2) = 1 - \frac{2}{N}, \quad \mu(2|1) = \mu(2|2) = \frac{2}{N}.$$

- *Function classes*: Let $w$ be defined as $\tilde{w}(2, a) = w_\alpha^\star(2, a)$ and $\tilde{w}(1, a) = 0$ for $a = 1, 2$. Consider the following function classes $\mathcal{W}$ and $\mathcal{V}$:

$$\mathcal{W} = \{w_\alpha^\star, \tilde{w}\}, \ \mathcal{V} = \{v_\alpha^\star, v^\star\}. \tag{36}$$

The proof is broken down into 4 steps. In the first step, we show that when $\alpha \leq \widetilde{O}(N^{-\frac{1}{2}})$ and $N$ is sufficiently large, the regularized optimal policy is the same as the unregularized optimal policy, i.e., $w_\alpha^\star = w^\star$. Therefore, the function class $\mathcal{W}$ defined in (36) is realizable $w_\alpha^\star = w^\star \in \mathcal{W}$. In the second step, we prove that with constant probability $v_\alpha^\star = \arg\min_{v \in \mathcal{V}} \hat{L}_\alpha^{CB}(\tilde{w}, v)$. Then, we show that solving the saddle point of the empirical objective $\hat{L}_\alpha^{CB}(w, v)$ selects $\tilde{w}$ over $w_\alpha^\star$ with a constant probability. Finally, we prove that $\tilde{w}$ induces a policy $\pi_{\tilde{w}}$ that suffers from suboptimality of order $N^{-1/4}$, which completes the proof.

**Step 1: Regularized optimal weights coincides with unregularized optimal weights.** Since the population optimization problem (24) is independent across states at a population level, we can use the result of Lemma 1 to conclude that

$v_\alpha^\star(s) = r^\star(s) - c(s)\alpha$, and

$$w_\alpha^\star(s, a) = \max\left\{0, (f')^{-1}\left(\frac{r(s, a) - v_\alpha^\star(s)}{\alpha}\right)\right\} = \max\left\{0, (f')^{-1}\left(c(s) - \frac{r^\star(s) - r(s, a)}{\alpha}\right)\right\},$$

where $0 \leq c(s) \leq f'(C^\star)$ for $s \in \{1, 2\}$. Since $r^\star(s) - r(s, 2) = \frac{1}{6} = \Theta(1)$, for $N \geq (6f'(C^\star))^2$, we have $w_\alpha^\star(s, 2) = 0$ for the suboptimal arm 2. Thus $w_\alpha^\star(s) = w^\star(s) = \frac{1}{\mu(1|s)}$. Correspondingly, we can use the KKT conditions to compute $v_\alpha^\star(s) = r^\star(s) - \alpha f'\left(\frac{1}{\mu(1|s)}\right)$.

**Step 2: $v_\alpha^\star = \arg\min_{v \in \mathcal{V}} \hat{L}_\alpha^{CB}(\tilde{w}, v)$ with constant probability.** Let $\hat{\mu}$ denote the empirical state-arm distribution and $\hat{r}$ denote the empirical mean reward. Define the following event:

$$\mathcal{E} := \left\{\sum_a \hat{\mu}(a|s)\tilde{w}(s, a) \leq 1 \ \text{ for } \ s \in \{1, 2\}\right\}. \tag{37}$$

Recall that we defined $\tilde{w}(1, a) = 0$ and $\tilde{w}(2, a) = w^\star(2, a)$. Thus, the above event can be equivalently written as

$$\sum_a \hat{\mu}(a|2)w_\alpha^\star(2, a) \leq 1 \iff \sum_a (\hat{\mu}(a|2) - \mu(a|2)) w_\alpha^\star(2, a) \leq 0. \tag{38}$$

Here we used the fact that $\sum_a \mu(a|2)w_\alpha^\star(s, 2) = 1$. Moreover, in Step 1 we showed that $w_\alpha^\star = w^\star$, thus $w_\alpha^\star(2, 2) = 0$ and (38) corresponds to the following event

$$\mathcal{E} = \{\hat{\mu}(1|2) - \mu(1|2) \leq 0\}. \tag{39}$$

Since $\hat{\mu}(1|2)$ is an empirical version of the conditional probability $\mu(1|2)$, event $\mathcal{E}$ happens with probability $\frac{1}{2}$.

We condition on the event $\mathcal{E}$ for the rest of the proof. Using the fact that $v_\alpha^\star(s) \leq r^\star(s) = v^\star(s)$, we conclude that

$$\hat{L}_\alpha^{CB}(\tilde{w}, v_\alpha^\star) \leq \hat{L}_\alpha^{CB}(\tilde{w}, v^\star) \Rightarrow \hat{L}_\alpha^{CB}(\tilde{w}, v_\alpha^\star) = \min_{v \in \mathcal{V}} \hat{L}_\alpha^{CB}(\tilde{w}, v). \tag{40}$$

**Step 3: Analyzing the probability of picking $w_\alpha^\star$.** Now we compare the value of $\hat{L}_\alpha^{CB}(\cdot, v_\alpha^\star)$ evaluated at $\tilde{w}$ and $w_\alpha^\star$. We use the definition $\tilde{w}(2, a) = w_\alpha^\star(2, a)$ and write

$$\hat{L}_\alpha^{CB}(w_\alpha^\star, v_\alpha^\star) - \hat{L}_\alpha^{CB}(\tilde{w}, v_\alpha^\star)$$
$$= \hat{\mu}(1)\left[\hat{r}(1, 1)\hat{\mu}(1|1)w_\alpha^\star(1, 1) + \alpha\hat{\mu}(1|1)f(w_\alpha^\star(1, 1)) + v_\alpha^\star(1)\left(\sum_a \hat{\mu}(a|1)(w(1, a) - w_\alpha^\star(1, a))\right)\right]$$

Noting that $v_\alpha^\star(1) = r(1,1) - \alpha f'(w_\alpha^\star(s_1, a_1))$, $\tilde{w}(1,a) = 0$, and $w_\alpha^\star(1,1) = w^\star(1,1) = \frac{1}{\mu(1|1)}$, we further simplify the above equation

$$
\hat{L}_\alpha^{\mathrm{CB}}(w_\alpha^\star, v_\alpha^\star) - \hat{L}_\alpha^{\mathrm{CB}}(\tilde{w}, v_\alpha^\star)
$$

$$
= \hat{\mu}(1)\left[ \left( \hat{r}(1,1) - r(1,1) + \alpha f'\left(\frac{1}{\mu(1|1)}\right) \right) \frac{\hat{\mu}(1|1)}{\mu(1|1)} + \alpha\hat{\mu}(1|1) f\left(\frac{1}{\mu(1|1)}\right) \right] \tag{41}
$$

$$
= \hat{\mu}(1,1)\left[ \frac{\hat{r}(1,1) - r(1,1)}{\mu(1|1)} + \alpha \cdot \left( \frac{1}{\mu(1|1)} f'\left(\frac{1}{\mu(1|1)}\right) + f\left(\frac{1}{\mu(1|1)}\right) \right) \right]. \tag{42}
$$

We then prove that with constant probability, the first term in (42) is negative with a magnitude larger than the second term:

$$
\frac{\hat{r}(1,1) - r(1,1)}{\mu(1|1)} \lesssim -N^{-3/8}. \tag{43}
$$

The proof of this inequality relies on anti-concentration bounds of binomial random variables and is presented at the end of this section. By Inequality (43) combined with (40), we conclude that

$$
\min_{v \in \mathcal{V}} \hat{L}_\alpha^{\mathrm{CB}}(\tilde{w}, v) = \hat{L}_\alpha^{\mathrm{CB}}(\tilde{w}, v_\alpha^\star) > \hat{L}_\alpha^{\mathrm{CB}}(w_\alpha^\star, v_\alpha^\star) \geq \min_{v \in \mathcal{V}} \hat{L}_\alpha^{\mathrm{CB}}(w_\alpha^\star, v), \tag{44}
$$

which guarantees that the algorithm picks $\tilde{w}$ with a constant probability.

**Step 4: Suboptimality of $\pi_w$** Finally, for the policy $\pi_w$ induced by $w$, we have

$$
J(\pi_\alpha^\star) - J(\pi_w) = \mu(s_1)\pi_w(2|1)(r(1,1) - r(1,2)) = \frac{N^{-\frac{1}{4}}}{12} \geq \Omega(N^\beta),
$$

for $\beta = -\frac{1}{4} > -\frac{1}{2}$, as desired. The proof for small $\alpha$ is thus complete.

*Proof of Inequality* (43). Using the Chernoff bounds for binomial random variables given in Lemma 2 (adapted from Proposition 7.3.2 of Matoušek & Vondrák (2001)), one can conclude that the following event $\mathcal{E}'$ happens with probability at least 0.5:

$$
\mathcal{E}' := \left\{ N(1,1) \geq 0.1 N\mu(1,1) \geq 0.05 N^{\frac{3}{4}} \right\}. \tag{45}
$$

Furthermore, $\mathcal{E}$ and $\mathcal{E}'$ are independent because the random variable $\hat{r}(s_1, a_1)$ is independent from the arm distribution within state $s_2$. Therefore, conditioning on $\mathcal{E} \cap \mathcal{E}'$ which happens with probability $0.5 \times 0.5 = 0.25$, we use the anti-concentration bounds for Binomial random variables Lemma 5 to obtain the following lower bound:

$$
\Pr\left( \hat{r}(1,1) - r(1,1) \leq -\sqrt{\frac{\log(2c_1)}{c_2 N(1,1)}} \leq -c' N^{-\frac{3}{8}} \,\middle|\, \mathcal{E} \cap \mathcal{E}' \right) \geq 0.5, \tag{46}
$$

where $c' = \sqrt{\frac{20\log(2c_1)}{c_2}}$ is a universal constant. Therefore, we have established that (43) holds with constant probability.

**Lemma 5 (Anti-concentration of Binomial random variables)** *Let $X_1, \cdots, X_n$ be independent random variables following the Bernoulli distribution with mean $\frac{1}{2}$, and let $\overline{X} = \frac{1}{n}\sum_{i=1}^n X_i$ be the empirical mean. Then we have that for any $t \in [0, \frac{1}{8}]$ and universal constants $c_1, c_2$,*

$$
\Pr\left( \overline{X} \leq \mathbb{E}[\overline{X}] - t \right) \geq c_1 e^{-c_2 t^2 n}. \tag{47}
$$

### D.3 PROOF OF THEOREM 3

Proof of this theorem largely follows similar steps as the proof we presented for Theorem 1. In particular, we start by presenting two lemmas. The first lemma leverages Hoeffding's inequality to establish the closeness of the population objective (7) and empirical objective (8). Additionally, we show that this result leads to the closeness of population objective at $w^\star$ and $\hat{w}$. Proof of this lemma is presented at the end of this subsection.

**Lemma 6 (Empirical and population closeness, CB)** *Fix $\delta > 0$ and define*

$$\epsilon_{stat}^{CB} := 3(B_w + 1)^2(B_v + 1)\sqrt{\frac{\log(|\mathcal{W}||\mathcal{V}|/\delta)}{N}}. \tag{48}$$

*For any $w \in \mathcal{W}$ and $v \in \mathcal{V}$, the following statements hold with probability at least $1 - \delta$*

*(I)* $\left| L_{AL}^{CB}(w, v) - \hat{L}_{AL}^{CB}(w, v) \right| \leq \epsilon_{stat}^{CB}$;

*(II)* $L_{AL}^{CB}(w^\star, v) - L_{AL}^{CB}(\hat{w}, v) \leq 2\epsilon_{stat}^{CB}$.

In the second lemma, we prove that the ALM term enforces a lower bound on normalization factors $\frac{d_{\hat{w}}}{\mu}(s) := \sum_a \hat{w}(s, a)\mu(a|s)$ for significant states.

**Lemma 7 (Occupancy validity enforced by the ALM)** *Define the state space subset*

$$\mathcal{S}_s := \left\{ s \,\middle|\, \frac{d_{\hat{w}}}{\mu}(s) \leq \frac{1}{2} \right\}. \tag{49}$$

*For any fixed $\delta > 0$, the following statements hold with probability at least $1 - \delta$,*

*(I)* $\mathbb{E}_{s,a\sim\mu}\left[(r^\star(s) - r(s,a))\hat{w}(s,a)\right] \lesssim \epsilon_{stat}^{CB}$;

*(II)* $\sum_{s \in \mathcal{S}_s} \mu(s) \lesssim \epsilon_{stat}^{CB}$;

*where $\epsilon_{stat}^{CB}$ is defined in (48).*

Given the two lemmas above, our suboptimality analysis can be broken down into two simple steps. First, we partition the states based on $\mathcal{S}_s$ defined in (49) and decompose the policy suboptimality accordingly:

$$\sum_s \mu(s)V^\star(s) - \sum_s \mu(s)V^{\hat{\pi}}(s) = \sum_{s \in \mathcal{S}_s} \mu(s)(V^\star(s) - V^{\hat{\pi}}(s)) + \sum_{s \notin \mathcal{S}_s} \mu(s)(V^\star(s) - V^{\hat{\pi}}(s)),$$

$$\lesssim \epsilon_{stat}^{CB} + \sum_{s \notin \mathcal{S}_s} \mu(s)(V^\star(s) - V^{\hat{\pi}}(s)) \tag{50}$$

$$\leq \epsilon_{stat}^{CB} + 2\sum_s d_{\hat{w}}(s)(V^\star(s) - V^{\hat{\pi}}(s)) \tag{51}$$

In (50), we used part (II) in Lemma 7 to bound the first term and (51) uses the fact that by definition, for all $s \notin \mathcal{S}_s$ we have $\mu(s) < 2\hat{d}(s)$ and $V^\star(s) - V^{\hat{\pi}}(s) \geq 0$. Moreover, the second term in (51) is bounded by part (I) of Lemma 7 since

$$\sum_s d_{\hat{w}}(s)(V^\star(s) - V^{\hat{\pi}}(s)) = \sum_{s:d_{\hat{w}}(s)>0} d_{\hat{w}}(s)\left(r^\star(s) - \sum_a \hat{\pi}(a|s)r(s,a)\right)$$

$$= \sum_{s:d_{\hat{w}}(s)>0} \sum_a d_{\hat{w}}(s,a)r^\star(s) - d_{\hat{w}}(s)\frac{\hat{w}(s,a)\mu(s,a)}{d_{\hat{w}}(s)}r(s,a)$$

$$= \sum_{s:d_{\hat{w}}(s)>0} \sum_a \hat{w}(s,a)\mu(s,a)r^\star(s) - \hat{w}(s,a)\mu(s,a)r(s,a)$$

$$\leq \sum_{s,a} \mu(s,a)\hat{w}(s,a)(r^\star(s) - r(s,a)) \lesssim \epsilon_{stat}^{CB},$$

where the equations follow from the definition of $\hat{\pi}$. The final suboptimality bound is proved by noting that $(B_w + 1)^2 \asymp B_w^2$ since $B_w \geq 1$ due to realizability of $w^\star(s, a^\star) \geq 1$.

*Proof of Lemma 6.* To prove part (I), notice that $\mathbb{E}_\mu\left[\hat{L}_{\text{AL}}^{\text{CB}}(w,v)\right] = L_{\text{AL}}^{\text{CB}}(w,v)$. Furthermore, $\hat{L}_{\text{AL}}^{\text{CB}}(w,v)$ is an empirical average of i.i.d. random variables which are bounded by

$$\left| w(s,a)r(s,a) - v(s)(w(s,a)-1) - \left(\sum_a w(s,a)\mu(a|s) - 1\right)^2 \right|$$
$$\leq B_w + B_v(B_w + 1) + B_w^2$$
$$\leq 3(B_w + 1)^2(B_v + 1)$$

Applying Hoeffding's inequality along with a union bound on $w \in \mathcal{W}$ and $v \in \mathcal{V}$ finishes the proof of part (I).

We now prove part (II). For the primal-dual objective without the AL term

$$\max_{w \geq 0} \min_v L^{\text{CB}}(w,v) := \mathbb{E}_{s,a\sim\mu}[w(s,a)r(s,a)] - \mathbb{E}_{s,a\sim\mu}[v(s)(w(s,a)-1)],$$

we have $(w^\star, v^\star) \in \text{argmax}_{w \geq 0} \text{argmin}_v L^{\text{CB}}(w,v)$ by strong duality. Moreover, since $w^\star$ is realizable, it satisfies the validity constraint $\mathbb{E}_{a\sim\mu(\cdot|s)}[w^\star(s,a)] = 1$ for all $s$. Therefore, by Lemma 14 adding the ALM term does not change the optimal solution and we have $(w^\star, v^\star) \in \text{argmax}_{w \geq 0} \text{argmin}_v L_{\text{AL}}^{\text{CB}}(w,v)$.

We follow similar steps as in the proof of Lemma 1 and decompose $L_{\text{AL}}^{\text{CB}}(w^\star, v) - L_{\text{AL}}^{\text{CB}}(\hat{w}, v)$ according to

$$L_{\text{AL}}^{\text{CB}}(w^\star, v) - L_{\text{AL}}^{\text{CB}}(\hat{w}, v)$$
$$= \underbrace{L_{\text{AL}}^{\text{CB}}(w^\star, v) - L_{\text{AL}}^{\text{CB}}(w^\star, \hat{v}(w^\star))}_{:=T_1} + \underbrace{L_{\text{AL}}^{\text{CB}}(w^\star, \hat{v}(w^\star)) - \hat{L}_{\text{AL}}^{\text{CB}}(w^\star, \hat{v}(w^\star))}_{:=T_2}$$
$$+ \underbrace{\hat{L}_{\text{AL}}^{\text{CB}}(w^\star, \hat{v}(w^\star)) - \hat{L}_{\text{AL}}^{\text{CB}}(\hat{w}, \hat{v})}_{:=T_3} + \underbrace{\hat{L}_{\text{AL}}^{\text{CB}}(\hat{w}, \hat{v}) - \hat{L}_{\text{AL}}^{\text{CB}}(\hat{w}, v)}_{:=T_4}$$
$$+ \underbrace{\hat{L}_{\text{AL}}^{\text{CB}}(\hat{w}, v) - L_{\text{AL}}^{\text{CB}}(\hat{w}, v)}_{:=T_5},$$

where $\hat{v}_w = \arg\min_{v \in \mathcal{V}} \hat{L}_{\text{AL}}^{\text{CB}}(w,v)$. Each term is bounded as follows:

- $T_1 = 0$ because $w^\star$ satisfies the optimization constraints.

- $T_2 \leq \epsilon_{\text{stat}}^{\text{CB}}$ due to Lemma 6.

- $T_3 \leq 0$ because $\hat{w} = \arg\max_{w \in \mathcal{W}} \hat{L}_{\text{AL}}^{\text{CB}}(\hat{v}_w, w)$.

- $T_4 \leq 0$ because $\hat{v} = \arg\min_{v \in \mathcal{V}} \hat{L}_{\text{AL}}^{\text{CB}}(v, \hat{w})$.

- $T_5 \leq \epsilon_{\text{stat}}^{\text{CB}}$ due to Lemma 6.

Summing up the bounds on each term proves part (II). $\qquad\square$

*Proof of Lemma 7.* We leverage the closeness of the objective at $w^\star$ and $\hat{w}$ established in Lemma 6 to show that the ALM term at $\hat{w}$ is small. Since $w^\star$ satisfies the validity constraints, the objective at $w^\star$ simplifies to

$$L_{\text{AL}}^{\text{CB}}(w^\star, v) = \mathbb{E}_{s,a\sim\mu}[r(s,a)w^\star(s,a)] + \underbrace{\mathbb{E}_{s,a\sim\mu}[v(s)(1 - w^\star(s,a))]}_{=0} - \underbrace{\mathbb{E}_{s\sim\mu}[(\mathbb{E}_{a\sim\mu(\cdot|s)}[w(s,a)] - 1)^2]}_{=0}$$
$$= \mathbb{E}_{s,a\sim\mu}[r(s,a)w^\star(s,a)].$$

Consider the objective difference at $v(s) = r^\star(s) := \max_a r(s,a)$:

$$L_{\text{AL}}^{\text{CB}}(w^\star, r^\star) - L_{\text{AL}}^{\text{CB}}(\hat{w}, r^\star)$$

$$= \sum_s \mu(s) r^\star(s) - \sum_{s,a} \mu(s,a) r(s,a) \hat{w}(s,a) - \sum_s r^\star(s) \left( \mu(s) - \sum_a \mu(s,a) \hat{w}(s,a) \right)$$

$$+ \mathbb{E}_{s \sim \mu} \left[ \left( \frac{d_{\hat{w}}}{\mu}(s) - 1 \right)^2 \right]$$

$$= \sum_{s,a} \mu(s,a)[r^\star(s) - r(s,a)] \hat{w}(s,a) + \mathbb{E}_{s \sim \mu} \left[ \left( \frac{d_{\hat{w}}}{\mu}(s) - 1 \right)^2 \right].$$

Since $L_{\text{AL}}^{\text{CB}}(w^\star, v) - L_{\text{AL}}^{\text{CB}}(\hat{w}, v) \lesssim \epsilon_{\text{stat}}^{\text{CB}}$ by Lemma 6, we conclude that

$$\sum_{s,a} \mu(s,a)[r^\star(s) - r(s,a)] \hat{w}(s,a) + \mathbb{E}_{s \sim \mu} \left[ \left( \frac{d_{\hat{w}}}{\mu}(s) - 1 \right)^2 \right] \lesssim \epsilon_{\text{stat}}^{\text{CB}}$$

Moreover, since the first term is nonnegative due to $\hat{w}(s,a) \geq 0$ and $r^\star(s) - r(s,a) \geq 0$, both of the terms in the above inequality are bounded by $\epsilon_{\text{stat}}^{\text{CB}}$ and thereby proving part (I).

The above result also allows us to bound the mass on the subset $\mathcal{S}_s$ that contains the states that violate state occupancy validity

$$\epsilon_{\text{stat}}^{\text{CB}} \gtrsim \sum_s \mu(s) \left[ \left( \frac{d_{\hat{w}}}{\mu}(s) - 1 \right)^2 \right] \geq \sum_{s \in \mathcal{S}_s} \mu(s) \left[ \left( \frac{d_{\hat{w}}}{\mu}(s) - 1 \right)^2 \right] \geq \frac{1}{4} \sum_{s \in \mathcal{S}_s} \mu(s) \gtrsim \sum_{s \in \mathcal{S}_s} \mu(s),$$

where we used the fact that $\frac{d_{\hat{w}}}{\mu}(s) \leq \frac{1}{2}$ and thus $\left( \frac{d_{\hat{w}}}{\mu}(s) - 1 \right)^2 \geq \frac{1}{4}$ by definition of $\mathcal{S}_s$. This concludes the proof of part (II). $\qquad \square$

# E   PROOFS FOR MDPS

In this section, we begin by introducing some additional notation. The original primal-dual objective without ALM term is given by

$$\max_{w \geq 0} \min_v L^{\text{MDP}}(w, v) := (1 - \gamma) \mathbb{E}_{s \sim \rho}[v(s)] + \mathbb{E}_{s,a \sim \mu} [w(s,a) e_v(s,a)]. \qquad (52)$$

Define $w^\star(s,a) = d^{\pi^\star}(s,a)/\mu(s,a)$ and $v^\star(s) = V^\star(s)$. By strong duality, one has $(w^\star, v^\star) \in \arg\max_{w \geq 0} \arg\min_v L^{\text{MDP}}(w, v)$. Additionally, define $\zeta_{w,u}^\star = \arg\max_{\zeta < 0} L_{\text{AL}}^{\text{model-free}}(w, v, u, \zeta), \forall w \in \mathcal{W}, u \in \mathcal{U}$ and $\zeta_w^\star = \zeta_{w,u_w^\star}^\star \; \forall w \in \mathcal{W}$ where $u_w^\star$ is defined in Theorem 4. Also, denote $\zeta^\star = \zeta_{w^\star}^\star$ and $u^\star = u_{w^\star}^\star$.

The rest of this section is organized as follows. In Appendix E.1, we provide some details regarding practical implementation of the offline learning algorithm with ALM. In Appendix E.2, we derive the objective of model-free ALMIS algorithm. Appendix E.3 contains the proof of performance upper bound on model-based and model-free ALMIS algorithms (Theorem 4), which relies on several lemmas subsequently proved in Appendices E.4 through E.7.

## E.1   ON PRACTICAL IMPLEMENTATIONS

In our algorithms for CB and MDP, we need to compute summations of form $\sum_{a \in \mathcal{A}}$. This can be implemented efficiently when $|\mathcal{A}|$ is small. When $|\mathcal{A}|$ is large or even infinite, one can utilize numerical methods to estimate the summation with desired precision. Additionally, in Algorithm 3, we need to evaluate a term $\sum_{s',a'} P(s'|s,a) \pi_w(a'|s') u(s',a')$. In practice, we can evaluate this term by numerical integration.

For MAB, CB, and model-based RL, our algorithms need to solve a max-min(-min) problem. For model-free RL, the max-min-min-max can be converted to a max(-max)-min(-min) problem. This is

because we can first exchange $\min_u$ and $\max_\zeta$ since $L_{\text{AL}}^{\text{model-free}}$ as defined in (54) is convex-concave w.r.t. $(u, \zeta)$. Then, we can exchange $\min_v$ and $\max_\zeta$ since $v$ and $\zeta$ are not coupling in $L_{\text{AL}}^{\text{model-free}}$. Therefore, our algorithms only require a max-min oracle, which is also required in prior works on provable conservative offline RL with general function approximators such as (Zhan et al., 2022). Moreover, many practically successful offline RL algorithms also solve minimax problems such as the DICE family (Nachum et al., 2019b;a; Yang et al., 2020; Lee et al., 2021)

## E.2 DERIVATION OF THE MODEL-FREE ALMIS OBJECTIVE

For $f(x) = (x - 1)^2$, the Fenchel conjugate $f_*$ is given by

$$f_*(x) = \max_y (xy - f(y)) = \max_y \left( xy - y^2 + 2y - 1 \right) = \left( \frac{x + 2}{2} \right)^2 - 1. \tag{53}$$

Since $d_w(s)/d^{\pi_w}(s) \geq 0$, we have $x_w^\star(s, a) \geq -2$ and thus it is sufficient to only consider domain $x(s, a) \geq -2$, over which $f_*(x)$ is invertible.

Let $g(x) = -f_*^{-1}(x) = 2 - 2\sqrt{x + 1}$, which is a convex function on $[-1, +\infty)$. Similar to Nachum et al. (2019a), we use Fenchel duality to estimate $g\left(u(s, a) - \gamma \mathbb{P}^{\pi_w} u(s, a)\right)$. By Fenchel duality, any convex function $g(x)$ can be written as $g(x) = \max_\zeta x\zeta - g_*(\zeta)$. In the case of $g(x)$, the Fenchel conjugate is given by $g_*(x) = -x - 2 - 1/x$ with domain $x < 0$. Therefore, we write

$$\mathbb{E}_\mu[w(s, a)g\left(u(s, a) - \gamma \mathbb{P}^{\pi_w} u(s, a)\right)]$$
$$= \mathbb{E}_\mu[w(s, a) \max_{\zeta < 0} \left(u(s, a) - \gamma(\mathbb{P}^{\pi_w} u)(s, a)\right)\zeta - g_*(\zeta)]$$
$$= \mathbb{E}_\mu[w(s, a) \max_{\zeta < 0} \left(u(s, a) - \gamma(\mathbb{P}^{\pi_w} u)(s, a)\right)\zeta + \zeta + 1/\zeta + 2].$$

The interchangeability principle (Rockafellar & Wets, 2009; Dai et al., 2017) allows us to convert the inner maximization step over scalar $\zeta$ to an overall maximization over $\zeta : \mathcal{S} \times \mathcal{A} \to \mathbb{R}^-$. Replacing this term in the objective (13) results in the following objective:

$$\max_{w \geq 0} \min_v \min_u \max_{\zeta < 0} L_{\text{AL}}^{\text{model-free}}(w, v, u, \zeta) = (1 - \gamma)\mathbb{E}_{s \sim \rho} \left[ v(s) + \sum_a u(s, a)\pi_w(a|s) \right] \tag{54}$$
$$+ \mathbb{E}_{(s, a, s') \sim \mu, a' \sim \pi_w(\cdot|s')}[w(s, a)\left(e_v(s, a) + (u(s, a) - \gamma u(s', a'))\zeta(s, a) - g_*(\zeta(s, a))\right)],$$

## E.3 PROOF OF THEOREM 4

We start by deriving an expression for $x_w^\star$ and characterizing bounds on $u_w^\star$ and $\zeta_{w,v}^\star$ in the following lemma. The proof is presented in Appendix E.4.

**Lemma 8** *For any $w \in \mathcal{W}$ and $v \in \mathcal{V}$, one has $x_w^\star(s, a) = 2d_w(s)/d^{\pi_w}(s) - 2$, $|u_w^\star(s, a)| \leq \frac{1}{1-\gamma}(B_x^2/4 + B_x)$, and $|\zeta_{w,v}^\star(s, a)| \in \left[ \frac{2}{2+B_x}, \frac{2}{2-B_x} \right]$.*

Bounding the suboptimality of policies returned by both model-based and model-free variants of ALMIS follow a similar analysis. We first characterize the statistical error in approximating population objectives by their empirical versions and use it to establish the closeness of $\hat{w}$ and $w^\star$. The lemma below captures these approximation errors for the model-based objective, whose proof can be found Appendix E.5.

**Lemma 9 (Empirical and population closeness, model-based ALMIS)** *Fix $\delta > 0$ and define*

$$\epsilon_{\text{stat}}^{\text{model-based}} := (B_u + (1 + B_v)B_w)\sqrt{\frac{B_u \log(|\mathcal{P}||\mathcal{U}||\mathcal{W}||\mathcal{V}|/\delta)}{N}}. \tag{55}$$

*For any $w \in \mathcal{W}, v \in \mathcal{V}$, and $u \in \mathcal{U}$, the following statements hold with probability at least $1 - \delta$*

*(I)* $\left| L_{\text{AL}}^{\text{model-based}}(w, v, u) - \hat{L}_{\text{AL}}^{\text{model-based}}(w, v, u) \right| \leq \epsilon_{\text{stat}}^{\text{model-based}}$;

*(II)* $L_{\text{AL}}^{\text{model-based}}(w^\star, v^\star, u^\star) - L_{\text{AL}}^{\text{model-based}}(\hat{w}, v^\star, u_{\hat{w}}^\star) \leq 2\epsilon_{\text{stat}}^{\text{model-based}}$.

In Appendix E.6, we prove a similar lemma for the model-free objective.

**Lemma 10 (Empirical and population closeness, model-free ALMIS)** *Fix $\delta > 0$ and define*

$$\epsilon_{stat}^{model\text{-}free} := (B_u + (1 + B_v + B_\zeta(B_u + 1))B_w)\sqrt{\frac{\log(|\mathcal{U}||\mathcal{W}||\mathcal{V}||\mathcal{Z}|/\delta)}{N}}. \tag{56}$$

*For any $w \in \mathcal{W}, v \in \mathcal{V}$, and $u \in \mathcal{U}$, the following statements hold with probability at least $1 - \delta$*

*(I)* $\left| L_{AL}^{model\text{-}free}(w, v, u) - \hat{L}_{AL}^{model\text{-}free}(w, v, u) \right| \leq \epsilon_{stat}^{model\text{-}free}$;

*(II)* $L_{AL}^{model\text{-}free}(w^\star, v^\star, u^\star) - L_{AL}^{model\text{-}free}(\hat{w}, v^\star, u_{\hat{w}}^\star) \leq 2\epsilon_{stat}^{model\text{-}free}$.

The final key lemma demonstrates that in model-based and model-free ALMIS, the ALM terms enforce lower bounds on the ratio of the estimated occupancy of learned weights $d_{\hat{w}}(s)$ and the actual occupancy of the learned policy $d^{\pi_{\hat{w}}}(s)$ in most states. The proof of this lemma is given in Appendix E.7.

**Lemma 11 (Occupancy validity by the ALM, MDP)** *For $\hat{w}$ computed by the model-based ALMIS Algorithm 3, define the state space subspace $\mathcal{S}_s := \left\{ s \,\middle|\, d_{\hat{w}}(s) \leq \frac{1}{2}d^{\pi_{\hat{w}}}(s) \right\}$. For any fixed $\delta > 0$, the following statements hold with probability at least $1 - \delta$*

*(I)* $\mathbb{E}_{s,a\sim\mu}\left[-A^\star(s,a)\hat{w}(s,a)\right] \lesssim \epsilon_{stat}^{model\text{-}based}$;

*(II)* $\sum_{s\in\mathcal{S}_s} d^{\pi_{\hat{w}}}(s) \lesssim (1-\gamma)^{-2}\epsilon_{stat}^{model\text{-}based}$.

*Similarly, for $\hat{w}$ computed by the model-free ALMIS Algorithm 4, define the state space subspace $\mathcal{S}_s := \left\{ s \,\middle|\, d_{\hat{w}}(s) \leq \frac{1}{2}d^{\pi_{\hat{w}}}(s) \right\}$. For any fixed $\delta > 0$, the following statements hold with probability at least $1 - \delta$*

*(I)* $\mathbb{E}_{s,a\sim\mu}\left[-A^\star(s,a)\hat{w}(s,a)\right] \lesssim \epsilon_{stat}^{model\text{-}free}$;

*(II)* $\sum_{s\in\mathcal{S}_s} d^{\pi_{\hat{w}}}(s) \lesssim (1-\gamma)^{-2}\epsilon_{stat}^{model\text{-}free}$.

Given the above lemmas, we proceed to prove the suboptimality bounds in terms of statistical errors defined in (55) and (56). In the rest of this section, we drop the superscripts model-based and model-free from statistical errors to avoid cluttered notation.

In view of the performance difference lemma in Kakade & Langford (2002, Lemma 6.1), one has

$$J(\pi^\star) - J(\hat{\pi}) = \mathbb{E}_{s\sim d^{\hat{\pi}}}\left[\sum_a A^\star(s,a)\left(\pi^\star(a|s) - \hat{\pi}(a|s)\right)\right] = \mathbb{E}_{s\sim d^{\hat{\pi}}}\left[\sum_a -A^\star(s,a)\hat{\pi}(a|s)\right],$$

where $d^{\hat{\pi}} = d^{\pi_{\hat{w}}}$. Here, we used the fact that the expectation of the optimal advantage over optimal policy is zero $\sum_a A^\star(s,a)\pi^\star(a|s) = 0$. Lemma 11 links an expectation of $-A^\star(s,a)$ to the statistical error. With this lemma at hand and using the definition $\mathcal{S}_s = \{s \mid d_{\hat{w}}(s) \leq d^{\hat{\pi}}(s)/2\}$, we continue to decompose and bound the suboptimality

$$\mathbb{E}_{s\sim d^{\hat{\pi}}}\left[\sum_a -A^\star(s,a)\hat{\pi}(a|s)\right]$$

$$= \sum_{s\in\mathcal{S}_s} d^{\hat{\pi}}(s)\left[\sum_a -A^\star(s,a)\hat{\pi}(a|s)\right] + \sum_{s\notin\mathcal{S}_s} d^{\hat{\pi}}(s)\left[\sum_a -A^\star(s,a)\hat{\pi}(a|s)\right]$$

$$\lesssim \frac{1}{(1-\gamma)^3}\epsilon_{stat} + \sum_{s\notin\mathcal{S}_s, d_{\hat{w}}(s)\neq 0} \frac{d^{\hat{\pi}}(s)}{d_{\hat{w}}(s)}\left[\sum_a -A^\star(s,a)\hat{w}(s,a)\mu(s,a)\right] \tag{57}$$

$$+ \sum_{s\notin\mathcal{S}_s, d_{\hat{w}}(s)=0} d^{\hat{\pi}}(s)\left[\sum_a -\frac{1}{|\mathcal{A}|}A^\star(s,a)\right]$$

$$\leq \frac{1}{(1-\gamma)^3}\epsilon_{stat} + 2\sum_{s\notin\mathcal{S}_s}\left[\sum_a -A^\star(s,a)\hat{w}(s,a)\mu(s,a)\right] \tag{58}$$

In (57), we used part (II) in Lemma 11 and that $-A^\star(s,a) \leq 1/(1-\gamma)$ and in (58) we used the definition of $\mathcal{S}_s$ to bound the ratio $d^{\hat\pi}(s)/d_{\hat w}(s)$ by 2 and the fact that $d_{\hat w}(s) = 0$ implies $d^{\hat\pi}(s) = 0$ for $s \notin \mathcal{S}_s$. We then apply part (I) in in Lemma 11 to bound the second term by $\mathbb{E}_{s,a\sim\mu}[-A^\star(s,a)\hat w(s,a)]$ and thus the overall suboptimality:

$$J(\pi^\star) - J(\hat\pi) \lesssim \frac{1}{(1-\gamma)^3}\epsilon_{\text{stat}} + \mathbb{E}_{s,a\sim\mu}\left[-A^\star(s,a)\hat w(s,a)\right] \lesssim \frac{1}{(1-\gamma)^3}\epsilon_{\text{stat}}.$$

### E.4    PROOF OF LEMMA 8

**Derivation of $x_w^\star$.**    Recall from Appendix E.2 that for $f(x) = (x-1)^2$, the Fenchel conjugate is $f_*(x) = \left(\frac{x+2}{2}\right)^2 - 1$. Therefore, for any $(s,a)$,

$$x_w^\star(s,a) = \arg\max_x \left(d_w(s)x - d^{\pi_w}(s)\left(\left(\frac{x+2}{2}\right)^2 - 1\right)\right) = 2\frac{d_w(s)}{d^{\pi_w}(s)} - 2$$

$$\Rightarrow \tilde x_w(s,a) = \text{clip}\left(2\frac{d_w(s)}{d^{\pi_w}(s)} - 2, -B_x, B_x\right).$$

**Bound on $u_w^\star$.**    Recall that $u_w^\star$ is defined as the fixed point of the following Bellman-like equation

$$u(s,a) = f_*(\tilde x_w(s,a)) + \gamma(\mathbb{P}^{\pi_w}u)(s,a). \tag{59}$$

The above equation has a solution since $f_*(\tilde x_w(s,a))$ is bounded

$$\left(\frac{2-B_x}{2}\right)^2 - 1 \leq f_*(\tilde x_w(s,a)) \leq \left(\frac{B_x+2}{2}\right)^2 - 1.$$

One can view $u_w^\star$ as the Q-function of policy $\pi_w$ with the reward function $f_*(\tilde x_w(s,a))$, which leads to $|u_w^\star(s,a)| \leq \frac{1}{1-\gamma}\max\left\{1 - \left(\frac{2-B_x}{2}\right)^2, \left(\frac{B_x+2}{2}\right)^2 - 1\right\} = \frac{1}{1-\gamma}(B_x^2/4 + B_x)$.

**Bound on $\zeta_{w,u}^\star$.**    To see the bound on $\zeta_{w,u}^\star$, recall that by definition,

$$\zeta_{w,u}^\star = \arg\max_{\zeta<0}\mathbb{E}_{(s,a,s')\sim\mu,a'\sim\pi_w(\cdot|s')}[w(s,a)\left((u(s,a) - \gamma u(s',a') + 1)\zeta(s,a) + 1/\zeta(s,a)\right)]. \tag{60}$$

It is easy to show that $|\zeta_{w,u}^\star(s,a)| = (u(s,a) - \gamma(\mathbb{P}^{\pi_w}u)(s',a') + 1)^{-1/2} = (f_*(\tilde x_w(s,a)) + 1)^{-1/2}$. Since $\tilde x_w(s,a) \in [-B_x, B_x]$, we have $|\zeta_{w,u}^\star(s,a)| \in \left[\frac{2}{2+B_x}, \frac{2}{2-B_x}\right]$.

### E.5    PROOF OF LEMMA 9

#### E.5.1    PROOF OF PART (I)

We decompose the difference between the population and empirical objective into three terms $L_{\text{AL}}^{\text{model-based}} - \hat L_{\text{AL}}^{\text{model-based}} = T_1 + T_2 + T_3$ defined as follows

$$T_1 := (1-\gamma)\mathbb{E}_\rho\left[v(s) + \sum_a u(s,a)\pi_w(a|s)\right] - (1-\gamma)\frac{1}{N_0}\sum_{i=1}^{N_0}\left(v(s_i) + \sum_a u(s_i,a)\pi_w(a|s_i)\right)$$

$$T_2 := \mathbb{E}_\mu\left[w(s,a)(r(s,a) + \gamma\sum_{s'}P(s'|s,a)v(s') - v(s))\right] - \frac{1}{N}\sum_{i=1}^N w(s_i,a_i)\left[r_i + \gamma v(s_i') - v(s_i)\right]$$

$$T_3 := \mathbb{E}_\mu\left[w(s,a)\left(f_*^{-1}\left(u(s,a) - \gamma P^{\pi_w}u(s,a)\right)\right)\right] - \frac{1}{N}\sum_{i=1}^N w(s_i,a_i)\left[f_*^{-1}\left(u(s_i,a_i) - \gamma\hat P^{\pi_w}u(s_i,a_i)\right)\right]$$

We subsequently show that the absolute values of the above error terms satisfy the following high probability upper bounds:

$$|T_1| \lesssim (B_v + B_u) \sqrt{\frac{\log|\mathcal{V}||\mathcal{U}|/\delta}{N_0}}, \tag{61a}$$

$$|T_2| \lesssim (1 + B_v) B_w \sqrt{\frac{\log|\mathcal{V}||\mathcal{W}|/\delta}{N}}, \tag{61b}$$

$$|T_3| \lesssim B_w \sqrt{\frac{B_u \log|\mathcal{P}||\mathcal{U}||\mathcal{W}|/\delta}{N}} \tag{61c}$$

Taking $N_0 = N$ and noting that $B_w \geq 1$ due to realizability of $w^\star$ yield that

$$\left| L_{\mathrm{AL}}^{\text{model-based}} - \hat{L}_{\mathrm{AL}}^{\text{model-based}} \right| \lesssim (B_v + B_u) \sqrt{\frac{\log|\mathcal{V}||\mathcal{U}|/\delta}{N_0}} + (1 + B_v) B_w \sqrt{\frac{B_u \log(|\mathcal{P}||\mathcal{U}||\mathcal{W}||\mathcal{V}|/\delta)}{N}}$$

$$\lesssim \epsilon_{\text{stat}}^{\text{model-based}}.$$

**Proof of bound** (61a) **on** $|T_1|$**.** Since $|v(s)| \leq B_v, |u(s,a)| \leq B_u$ for all $v \in \mathcal{V}$ and $u \in \mathcal{U}$ and $s_i$ are independent, we can apply Hoeffding's inequality and union bound to conclude the advertised bound (61a) on $|T_1|$.

**Proof of the bound** (61b) **on** $|T_2|$**.** By boundedness of $w, v$, we have

$$|w(s,a)(r(s,a) + \gamma v(s') - v(s))| \leq B_w(1 + (\gamma+1)B_v) \leq B_w(1+\gamma)(1+B_v).$$

As before, due to boundedness and independence of variables $w(s_i, a_i)[r_i + \gamma v(s'_i) - v(s_i)]$, Hoeffding's inequality can be applied, giving the bound (61b) on $|T_2|$.

**Proof of the bound** (61c) **on** $|T_3|$**.** We decompose $T_3 = T_{3,1} + T_{3,2}$, where $T_{3,1}$ and $T_{3,2}$ are defined as

$$T_{3,1} := \mathbb{E}_\mu \left[ w(s,a) \left( f_*^{-1} \left( u(s,a) - \gamma (\mathbb{P}^{\pi_w} u)(s,a) \right) \right) \right] - \mathbb{E}_\mu \left[ w(s,a) \left( f_*^{-1} \left( u(s,a) - \gamma (\hat{\mathbb{P}}^{\pi_w} u)(s,a) \right) \right) \right]$$

$$T_{3,2} := \mathbb{E}_\mu \left[ w(s,a) \left( f_*^{-1} \left( u(s,a) - \gamma (\hat{\mathbb{P}}^{\pi_w} u)(s,a) \right) \right) \right]$$

$$- \frac{1}{N} \sum_{i=1}^N w(s_i, a_i) \left[ f_*^{-1} \left( u(s_i, a_i) - \gamma (\hat{\mathbb{P}}^{\pi_w} u)(s_i, a_i) \right) \right]$$

Recall that $f_*^{-1}(x) = 2\sqrt{x+1} - 2$ from Appendix E.2. The absolute value of $T_{3,2}$ can be immediately bounded using Hoeffding's inequality:

$$|T_{3,2}| \lesssim B_w \sqrt{\frac{B_u \log|\mathcal{W}||\mathcal{U}|\delta}{N}}. \tag{62}$$

To bound $|T_{3,1}|$, we first use the inequality given in Lemma 13, setting $b_i, x_i, y_i$ for each $(s,a)$ according to

$$b_i = \begin{cases} 1 + u(s,a) & i = 0 \\ \gamma \sum_{a'} \pi_w(a'|s') u(s'|a') & 1 \leq i \leq |\mathcal{S}| \end{cases}$$

$$x_i = \begin{cases} 1 & i = 0 \\ P(s'|s,a) & 1 \leq i \leq |\mathcal{S}| \end{cases}, \quad y_i = \begin{cases} 1 & i = 0 \\ P(s'|s,a) & 1 \leq i \leq |\mathcal{S}| \end{cases}$$

Thus by Lemma 13, we obtain the following bound on $T_{3,1}^2$

$$
\begin{aligned}
T_{3,1}^2 &= \left( \mathbb{E}_\mu \left[ w(s,a) \left( f_*^{-1} \left( u(s,a) - \gamma \mathbb{P}^{\pi_w} u(s,a) \right) \right) \right] - \mathbb{E}_\mu \left[ w(s,a) \left( f_*^{-1} \left( u(s,a) - \gamma \hat{\mathbb{P}}^{\pi_w} u(s,a) \right) \right) \right] \right)^2 \\
&\lesssim B_w \Bigg( \mathbb{E}_\mu \left[ \sqrt{1 + u(s,a) - \gamma \sum_{s'} P(s'|s,a) \sum_{a'} \pi_w(a'|s') u(s',a')} \right] \\
&\qquad - \mathbb{E}_\mu \left[ \sqrt{1 + u(s,a) - \gamma \sum_{s'} \hat{P}(s'|s,a) \sum_{a'} \pi_w(a'|s') u(s',a')} \right] \Bigg)^2 \\
&\leq B_w^2 B_u \mathbb{E}_\mu \left[ \sum_{s'} \left( \sqrt{P(s'|s,a)} - \sqrt{\hat{P}(s'|s,a)} \right)^2 \right].
\end{aligned}
\tag{63}
$$

Note that the terms under square root are always nonnegative because for any transition $P$

$$
1 + u(s,a) - \gamma \sum_{s'} P(s'|s,a) \sum_{a'} \pi_w(a'|s') u(s',a') \geq 1 - B_u - \gamma B_u \geq 1 - 2B_u \geq 0.
$$

Then, we use the concentration result on maximum likelihood model estimation stated in Theorem 6 and a union bound on $w \in \mathcal{W}$ and $v \in \mathcal{V}$ to conclude that

$$
|T_{3,1}| \lesssim B_w \sqrt{\frac{B_u \log |\mathcal{P}||\mathcal{U}||\mathcal{W}|/\delta}{N}}.
\tag{64}
$$

### E.5.2 PROOF OF PART (II)

To prove the second part, let $\hat{v}_w$ and $\hat{u}_w$ denote the solutions to the model-based empirical objective

$$
\hat{v}_w, \hat{u}_w = \underset{v \in \mathcal{V}}{\operatorname{argmin}} \underset{u \in \mathcal{U}}{\operatorname{argmin}} \hat{L}_{\text{AL}}^{\text{model-based}}(w, v, u)
$$

Decompose the objective difference according to

$$
\begin{aligned}
L_{\text{AL}}^{\text{model-based}}&(w^\star, v^\star, u^\star) - L_{\text{AL}}^{\text{model-based}}(\hat{w}, v^\star, u_{\hat{w}}^\star) \\
&= L_{\text{AL}}^{\text{model-based}}(w^\star, v^\star, u^\star) - L_{\text{AL}}^{\text{model-based}}(w^\star, \hat{v}_{w^\star}, \hat{u}_{w^\star}) \quad := T_1 \\
&\quad + L_{\text{AL}}^{\text{model-based}}(w^\star, \hat{v}_{w^\star}, \hat{u}_{w^\star}) - \hat{L}_{\text{AL}}^{\text{model-based}}(w^\star, \hat{v}_{w^\star}, \hat{u}_{w^\star}) \quad := T_2 \\
&\quad + \hat{L}_{\text{AL}}^{\text{model-based}}(w^\star, \hat{v}_{w^\star}, \hat{u}_{w^\star}) - \hat{L}_{\text{AL}}^{\text{model-based}}(\hat{w}, \hat{v}_{\hat{w}}, \hat{u}_{\hat{w}}) \quad := T_3 \\
&\quad + \hat{L}_{\text{AL}}^{\text{model-based}}(\hat{w}, \hat{v}_{\hat{w}}, \hat{u}_{\hat{w}}) - \hat{L}_{\text{AL}}^{\text{model-based}}(\hat{w}, v^\star, u_{\hat{w}}^\star) \quad := T_4 \\
&\quad + \hat{L}_{\text{AL}}^{\text{model-based}}(\hat{w}, v^\star, u_{\hat{w}}^\star) - L_{\text{AL}}^{\text{model-based}}(\hat{w}, v^\star, u_{\hat{w}}^\star) \quad := T_5
\end{aligned}
$$

We bound each term:

- $T_1 \leq 0$ because $v^\star, u^\star = \arg\min_v \arg\min_u L_{\text{AL}}^{\text{model-based}}(w^\star, v, u)$;

- $T_2 \leq \epsilon_{\text{stat}}^{\text{model-based}}$ by Lemma 9;

- $T_3 \leq 0$ because $\hat{w} = \arg\max_{w \in \mathcal{W}} \hat{L}_{\text{AL}}^{\text{model-based}}(w, \hat{v}_w, \hat{u}_w)$;

- $T_4 \leq 0$ because $\hat{v}_w, \hat{u}_w = \arg\min_{v \in \mathcal{V}} \arg\min_{u \in \mathcal{U}} \hat{L}_{\text{AL}}^{\text{model-based}}(w, v, u)$;

- $T_5 \leq \epsilon_{\text{stat}}^{\text{model-based}}$ by Lemma 9.

### E.6 PROOF OF LEMMA 10

#### E.6.1 PROOF OF PART (I)

We decompose the difference $L_{\mathrm{AL}}^{\text{model-free}} - \hat{L}_{\mathrm{AL}}^{\text{model-free}} = T_1 + T_2 + T_3$ into three error terms

$$T_1 := (1-\gamma)\mathbb{E}_\rho \left[ v(s) + \sum_a u(s,a)\pi_w(a|s) \right] - (1-\gamma)\frac{1}{N_0}\sum_{i=1}^{N_0}\left( v(s_i) + \sum_a u(s_i,a)\pi_w(a|s_i) \right)$$

$$T_2 := \mathbb{E}_\mu \left[ w(s,a)(r(s,a) + \gamma\sum_{s'} P(s'|s,a)v(s') - v(s)) \right] - \frac{1}{N}\sum_{i=1}^{N} w(s_i,a_i)\left[ r_i + \gamma v(s_i') - v(s_i) \right]$$

$$T_3 := \mathbb{E}_{(s,a,s')\sim\mu,a'\sim\pi_w(\cdot|s')}[w(s,a)\left((u(s,a)-\gamma u(s',a'))\zeta(s,a) - g_\star(\zeta(s,a))\right)]$$

$$- \frac{1}{N}\sum_{i=1}^{N} w(s_i,a_i)\left[ \left( u(s_i,a_i) - \gamma\sum_{a'\in\mathcal{A}} u(s_i',a')\pi_w(a'|s_i') \right)\zeta(s_i,a_i) - g_\star(\zeta(s_i,a_i)) \right].$$

The absolute values of the error terms above satisfy the following upper bounds with high probability

$$|T_1| \lesssim (B_v + B_u)\sqrt{\frac{\log(|\mathcal{V}||\mathcal{U}|/\delta)}{N_0}}, \tag{65a}$$

$$|T_2| \lesssim (1 + B_v)B_w\sqrt{\frac{\log(|\mathcal{V}||\mathcal{W}|/\delta)}{N}}, \tag{65b}$$

$$|T_3| \lesssim (1 + B_\zeta(B_u+1))B_w\sqrt{\frac{\log|\mathcal{U}||\mathcal{W}||\mathcal{Z}|/\delta}{N}}. \tag{65c}$$

The bounds on the first two error terms $|T_1|$ and $|T_2|$ are already shown in Appendix E.5.1. To bound $|T_3|$, recall that $g_\star(x) = -x - 2 - \frac{1}{x}$, $\forall x < 0$. Also, $|\zeta(s,a)| \in (B_{\zeta,L}, B_{\zeta,U})$ for any $\zeta \in \mathcal{Z}$ and any $(s,a)$, and $B_\zeta \triangleq \max\{B_{\zeta,U}, B_{\zeta,L}^{-1}\}$. Therefore, the individual error terms in $|T_3|$ satisfy the following bound

$$|w(s,a)\left((u(s,a)-\gamma u(s',a'))\zeta(s,a) - g_\star(\zeta(s,a))\right)| \le B_w((1+\gamma)B_u B_{\zeta,U} + B_{\zeta,U} + B_{\zeta,L}^{-1} + 2).$$

Thus, by Hoeffding's inequality and a union bound on $\mathcal{W}, \mathcal{U}$, and $\mathcal{Z}$, we obtain the upper bound (65c) on $|T_3|$. Summing up the bounds given in (65a), (65b), and (65c) and noting that $B_w \ge 1$ due to realizability of $w^\star$, we obtain

$$L_{\mathrm{AL}}^{\text{model-free}}(w,v,u,\zeta) - \hat{L}_{\mathrm{AL}}^{\text{model-free}}(w,v,u,\zeta)$$

$$\lesssim (B_v + B_u)\sqrt{\frac{\log|\mathcal{V}||\mathcal{U}|/\delta}{N_0}} + (1 + B_v + B_\zeta(B_u+1))B_w\sqrt{\frac{\log|\mathcal{U}||\mathcal{W}||\mathcal{V}||\mathcal{Z}|/\delta}{N}}$$

$$\lesssim \epsilon_{\text{stat}}^{\text{model-free}}.$$

#### E.6.2 PROOF OF PART (II)

Define the following solutions to the empirical model-free objective

$$\hat{v}_w, \hat{u}_w, \hat{\zeta}_w = \arg\min_{v\in\mathcal{V}} \arg\min_{u\in\mathcal{U}} \arg\max_{\zeta\in\mathcal{Z}} \hat{L}_{\mathrm{AL}}^{\text{model-free}}(w,v,u,\zeta), \quad \forall w \in \mathcal{W}$$

$$\hat{\zeta}(w,u) = \arg\max_{\zeta\in\mathcal{Z}} \hat{L}_{\mathrm{AL}}^{\text{model-free}}(w,v,u,\zeta) \quad \forall w \in \mathcal{W}, u \in \mathcal{U}$$

Decompose the objective difference according to

$$
\begin{aligned}
L_{\mathrm{AL}}^{\text{model-free}}&(w^\star, v^\star, u^\star, \zeta^\star) - L_{\mathrm{AL}}^{\text{model-free}}(\hat{w}, v^\star, u^\star_{\hat{w}}, \zeta^\star_{\hat{w}}) \\
&= L_{\mathrm{AL}}^{\text{model-free}}(w^\star, v^\star, u^\star, \zeta^\star) - L_{\mathrm{AL}}^{\text{model-free}}(w^\star, \hat{v}_{w^\star}, \hat{u}_{w^\star}, \zeta^\star(w^\star, \hat{u}_{w^\star})) \quad \coloneqq T_1 \\
&\quad + L_{\mathrm{AL}}^{\text{model-free}}(w^\star, \hat{v}_{w^\star}, \hat{u}_{w^\star}, \zeta^\star_{w^\star, \hat{u}_{w^\star}}) - \hat{L}_{\mathrm{AL}}^{\text{model-free}}(w^\star, \hat{v}_{w^\star}, \hat{u}_{w^\star}, \zeta^\star_{w^\star, \hat{u}_{w^\star}}) \quad \coloneqq T_2 \\
&\quad + \hat{L}_{\mathrm{AL}}^{\text{model-free}}(w^\star, \hat{v}_{w^\star}, \hat{u}_{w^\star}, \zeta^\star_{w^\star, \hat{u}_{w^\star}}) - \hat{L}_{\mathrm{AL}}^{\text{model-free}}(w^\star, \hat{v}_{w^\star}, \hat{u}_{w^\star}, \hat{\zeta}_{w^\star}) \quad \coloneqq T_3 \\
&\quad + \hat{L}_{\mathrm{AL}}^{\text{model-free}}(w^\star, \hat{v}_{w^\star}, \hat{u}_{w^\star}, \hat{\zeta}_{w^\star}) - \hat{L}_{\mathrm{AL}}^{\text{model-free}}(\hat{w}, \hat{v}_{\hat{w}}, \hat{u}_{\hat{w}}, \hat{\zeta}_{\hat{w}}) \quad \coloneqq T_4 \\
&\quad + \hat{L}_{\mathrm{AL}}^{\text{model-free}}(\hat{w}, \hat{v}_{\hat{w}}, \hat{u}_{\hat{w}}, \hat{\zeta}_{\hat{w}}) - \hat{L}_{\mathrm{AL}}^{\text{model-free}}(\hat{w}, v^\star, u^\star_{\hat{w}}, \hat{\zeta}_{\hat{w}, u^\star_{\hat{w}}}) \quad \coloneqq T_5 \\
&\quad + \hat{L}_{\mathrm{AL}}^{\text{model-free}}(\hat{w}, v^\star, u^\star_{\hat{w}}, \hat{\zeta}_{\hat{w}, u^\star_{\hat{w}}}) - L_{\mathrm{AL}}^{\text{model-free}}(\hat{w}, v^\star, u^\star_{\hat{w}}, \hat{\zeta}_{\hat{w}, u^\star_{\hat{w}}}) \quad \coloneqq T_6 \\
&\quad + L_{\mathrm{AL}}^{\text{model-free}}(\hat{w}, v^\star, u^\star_{\hat{w}}, \hat{\zeta}_{\hat{w}, u^\star_{\hat{w}}}) - L_{\mathrm{AL}}^{\text{model-free}}(\hat{w}, v^\star, u^\star_{\hat{w}}, \zeta^\star_{\hat{w}}) \quad \coloneqq T_7
\end{aligned}
$$

We bound each term:

- $T_1 \leq 0$ because $v^\star, u^\star = \arg\min_{v,u} L_{\mathrm{AL}}^{\text{model-free}}(w^\star, v, u, \zeta^\star(w^\star, u))$;

- $T_2 \leq \epsilon_{\text{model-free}}$ by part (I);

- $T_3 \leq 0$ because $\hat{\zeta}_{w^\star} = \hat{\zeta}_{w^\star, \hat{u}_{w^\star}} = \arg\max_{\zeta \in \mathcal{Z}} \hat{L}_{\mathrm{AL}}^{\text{model-free}}(w^\star, \hat{v}_{w^\star}, \hat{u}_{w^\star}, \zeta)$;

- $T_4 \leq 0$ because $\hat{w} = \arg\max_{w \in \mathcal{W}} \hat{L}_{\mathrm{AL}}^{\text{model-free}}(w, \hat{v}_w, \hat{u}_w, \hat{\zeta}_w)$;

- $T_5 \leq 0$ because $\hat{v}_{\hat{w}}, \hat{u}_{\hat{w}} = \arg\min_{v \in \mathcal{V}, u \in \mathcal{U}} \hat{L}_{\mathrm{AL}}^{\text{model-free}}(\hat{w}, v, u, \hat{\zeta}_{\hat{w}, u})$;

- $T_6 \leq \epsilon_{\text{model-free}}$ by part (I);

- $T_7 \leq 0$ because $\zeta^\star_{\hat{w}} = \zeta^\star_{\hat{w}, u^\star_{\hat{w}}} = \arg\max_{\zeta < 0} L_{\mathrm{AL}}^{\text{model-free}}(\hat{w}, v^\star, u^\star_{\hat{w}}, \zeta)$.

### E.7 PROOF OF LEMMA 11

We provide proof only for the model-based algorithm and let $\hat{w} = \hat{w}^{\text{model-based}}$ for notation convenience. The proof for a model-free algorithm follows analogously, noting the fact that $L_{AL}^{\text{model-free}}(w, v^\star, u^\star_w, \zeta^\star_w) = L_{AL}^{\text{model-based}}(w, v^\star, u^\star_w)$ and we can replace Lemma 9 with Lemma 10 to prove the model-free version.

#### E.7.1 PROOF OF PART (I)

Consider the expression of the model-based objective $L_{\mathrm{AL}}^{\text{model-based}}(w^\star, v^\star, u^\star)$ at the optimal solution where $u^\star \coloneqq u^\star_{w^\star}$:

$$
\begin{aligned}
L_{\mathrm{AL}}^{\text{model-based}}&(w^\star, v^\star, u^\star) \\
&= (1-\gamma)\mathbb{E}_{s \sim \rho}[v^\star(s)] + \mathbb{E}_{s, a \sim \mu}\left[w^\star(s,a) e_{v^\star}(s,a)\right] - \mathbb{E}_{s \sim d^{\pi_{w^\star}}}\left(\frac{d_{w^\star}(s)}{d^{\pi_{w^\star}}(s)} - 1\right)^2 \\
&= (1-\gamma)\mathbb{E}_{s \sim \rho}[V^\star(s)] + \mathbb{E}_{s, a \sim \mu}\left[w^\star(s,a) A^\star(s,a)\right]
\end{aligned}
\tag{66}
$$

The first equation comes from the fact that $u^\star$ is the optimal solution to the variational lower bound, making it equal to the $f$-divergence. To see this, recall from Lemma 8 that $x^\star_w(s,a) = 2d_w(s)/d^{\pi_w}(s) - 2$ and $\tilde{x}_w(s,a) = \text{clip}(x^\star_w(s,a), -B_x, B_x)$. Since $x^\star_{w^\star}(s,a) = 0$, we have $\tilde{x}_{w^\star}(s,a) = x^\star_{w^\star}(s,a)$ and thus $u^\star$ recovers the $f$-divergence.

In Equation (66), we wrote $v^\star(s) = V^\star(s)$ since $v^\star(s)$ is the optimal solution to the primal-dual program without the ALM term and is equal to the optimal value function (Zhan et al., 2022). We also used the fact that $e_{v^\star}(s,a) = r(s,a) + \gamma \sum_{s'} P(s'|s,a)v^\star(s') - v^\star(s) = A^\star(s,a)$ is the optimal advantage function, and that $d_{w^\star}(s) = d^{\pi_{w^\star}}(s)$ by definition and realizability of $w^\star$. Moreover, the second term in (66) is zero since it captures the optimal advantage of optimal policy. Therefore, we conclude that

$$
L_{\mathrm{AL}}^{\text{model-based}}(w^\star, v^\star, u^\star) = (1-\gamma)\mathbb{E}_{s \sim \rho}[V^\star(s)].
\tag{67}
$$

Given the above expression of the objective at $(w^\star, v^\star, u^\star)$, we write the following objective difference

$$
\begin{aligned}
&L_{\mathrm{AL}}^{\text{model-based}}(w^\star, v^\star, u^\star) - L_{\mathrm{AL}}^{\text{model-based}}(\hat{w}, v^\star, u_{\hat{w}}^\star) \\
&= (1-\gamma)\mathbb{E}_{s \sim \rho}[V^\star(s)] - (1-\gamma)\mathbb{E}_{s \sim \rho}[v^\star(s)] - \mathbb{E}_{s, a \sim \mu}[\hat{w}(s, a)e_{v^\star}(s, a)] \\
&\quad + (1-\gamma)\mathbb{E}_{s \sim \rho, a \sim \pi_{\hat{w}}}[u_{\hat{w}}^\star(s, a)] + \mathbb{E}_\mu\left[\hat{w}(s, a)f_*^{-1}\left(u_{\hat{w}}^\star(s, a) - \gamma(\mathbb{P}^{\pi_{\hat{w}}}u_{\hat{w}}^\star)(s, a)\right)\right] \\
&= -\mathbb{E}_{s, a \sim \mu}[\hat{w}(s, a)A^\star(s, a)] - \mathbb{E}_{d^{\pi_{\hat{w}}}}[f_*(\tilde{x}_{\hat{w}}(s, a))] + \mathbb{E}_{d_{\hat{w}}}[\tilde{x}_{\hat{w}}(s, a)]
\end{aligned}
\tag{68}
$$

The last line uses $e_{v^\star}(s, a) = A^\star(s, a)$ as well as the definition of $u_{\hat{w}}^\star$ as the fixed point solution to

$$
u_{\hat{w}}^\star(s, a) := f_*(\tilde{x}_{\hat{w}}(s, a)) + \gamma(\mathbb{P}^{\pi_{\hat{w}}}u_{\hat{w}}^\star)(s, a),
$$

which allows us to write (68) in the original $f$-divergence variational form (11) with $\tilde{x}_{\hat{w}}$ as variable. Lemma 9 asserts that $L_{\mathrm{AL}}^{\text{model-based}}(w^\star, v^\star, u^\star) - L_{\mathrm{AL}}^{\text{model-based}}(\hat{w}, v^\star, u_{\hat{w}}^\star) \lesssim \epsilon_{\text{stat}}^{\text{model-based}}$. Therefore,

$$
-\mathbb{E}_{s, a \sim \mu}[\hat{w}(s, a)A^\star(s, a)] - \mathbb{E}_{d^{\pi_{\hat{w}}}}[f_*(\tilde{x}_{\hat{w}}(s, a))] + \mathbb{E}_{d_{\hat{w}}}[\tilde{x}_{\hat{w}}(s, a)] \lesssim \epsilon_{\text{stat}}^{\text{model-based}}.
\tag{69}
$$

We next argue that both terms in inequality above are nonnegative and conclude that

$$
-\mathbb{E}_{s, a \sim \mu}[\hat{w}(s, a)A^\star(s, a)] \lesssim \epsilon_{\text{stat}}^{\text{model-based}}
\tag{70a}
$$

$$
-\mathbb{E}_{d^{\pi_{\hat{w}}}}[f_*(\tilde{x}_{\hat{w}}(s, a))] + \mathbb{E}_{d_{\hat{w}}}[\tilde{x}_{\hat{w}}(s, a)] \lesssim \epsilon_{\text{stat}}^{\text{model-based}}
\tag{70b}
$$

The first term is nonnegative because for the optimal advantage function we have $A^\star(s, a) \leq 0$ for all $s \in \mathcal{S}$ and $a \in \mathcal{A}$. We write the second term as

$$
-\mathbb{E}_{d^{\pi_{\hat{w}}}}[f_*(\tilde{x}_{\hat{w}}(s, a))] + \mathbb{E}_{d_{\hat{w}}}[\tilde{x}_{\hat{w}}(s, a)] = \mathbb{E}_{d^{\pi_{\hat{w}}}}\left[\frac{d_{\hat{w}}(s)}{d^{\pi_{\hat{w}}}(s)}\tilde{x}_{\hat{w}}(s, a) - f_*(\tilde{x}_{\hat{w}}(s, a))\right].
$$

We then show that each term inside the expectation is nonnegative:

$$
\frac{d_w(s)}{d^{\pi_w}(s)}\tilde{x}_w(s, a) - f_*(\tilde{x}_w(s, a)) \geq 0 \quad \forall s \in \mathcal{S}, w \in \mathcal{W}.
\tag{71}
$$

**Proof of bound** (71). we separate the argument into three cases and use the expression of $\tilde{x}_w$ given in Lemma 8.

1. When $1 - B_x/2 \leq \frac{d_w(s)}{d^{\pi_w}(s)} \leq B_x/2 + 1$, we have $\tilde{x}_w(s, a) = \left(2\frac{d_w(s)}{d^{\pi_w}(s)} - 2\right)$ and therefore

$$
\frac{d_w(s)}{d^{\pi_w}(s)}\tilde{x}_w(s, a) - f_*(\tilde{x}_w(s, a)) = \left(\frac{d_w(s)}{d^{\pi_w}(s)} - 1\right)^2 \geq 0.
$$

2. When $\frac{d_w(s)}{d^{\pi_w}(s)} > B_x/2 + 1$, substitute $\tilde{x}_w(s, a) = B_x$ to arrive at

$$
\frac{d_w(s)}{d^{\pi_w}(s)}B_x - \left(\left(\frac{B_x}{2} + 1\right)^2 - 1\right) \geq \left(\frac{B_x}{2} + 1\right)B_x - \frac{B_x^2}{4} - B_x = \frac{B_x^2}{4} \geq 0.
$$

3. Similarly, when $\frac{d_w(s)}{d^{\pi_w}(s)} < 1 - B_x/2$, substitute $\tilde{x}_w(s, a) = -B_x$ to arrive at

$$
-\frac{d_w(s)}{d^{\pi_w}(s)}B_x - \left(\left(1 - \frac{B_x}{2}\right)^2 - 1\right) \geq \left(\frac{B_x}{2} - 1\right)B_x - \frac{B_x^2}{4} + B_x = \frac{B_x^2}{4} \geq 0.
\tag{72}
$$

### E.7.2 PROOF OF PART (II)

We derive the second part by using the bound (70b) restricted on the set $\mathcal{S}_s$. When $s \in \mathcal{S}_s$, we have $\frac{d_{\hat{w}}(s)}{d^{\pi_{\hat{w}}}(s)} \leq \frac{1}{2}$ and thus the variational form falls into the case 3 in the proof of bound (71). Therefore, for $s \in \mathcal{S}_s$, we have $\tilde{x}_{\hat{w}}(s, a) = -B_x$ and

$$
\frac{d_{\hat{w}}(s)}{d^{\pi_{\hat{w}}}(s)}\tilde{x}_{\hat{w}}(s, a) - f_*(\tilde{x}_{\hat{w}}(s, a)) \gtrsim (1-\gamma)^2 \quad \forall s \in \mathcal{S}_s.
\tag{73}
$$

We use the bound in (70b) as well as (73) to conclude that

$$
\begin{aligned}
\epsilon_{\text{stat}}^{\text{model-based}} &\gtrsim \mathbb{E}_{d_{\hat{w}}}[\tilde{x}_{\hat{w}}(s,a)] - \mathbb{E}_{d^{\pi_{\hat{w}}}}[f_*(\tilde{x}_{\hat{w}}(s,a))] \\
&= \sum_s d^{\pi_{\hat{w}}}(s) \left( \frac{d_{\hat{w}}(s)}{d^{\pi_{\hat{w}}}(s)} \tilde{x}_{\hat{w}}(s,a) - f_*(\tilde{x}_{\hat{w}}(s,a)) \right) \\
&\gtrsim \sum_{s \in \mathcal{S}_s} (1-\gamma)^2 d^{\pi_{\hat{w}}}(s),
\end{aligned}
$$

which leads to the second advertised claim $\sum_{s \in \mathcal{S}_s} d^{\pi_{\hat{w}}}(s) \lesssim (1-\gamma)^{-2} \epsilon_{\text{stat}}$.

### E.8 ALM BASED ON BELLMAN FLOW ERROR CONSTRAINT IS INSUFFICIENT

We demonstrated that Lagrange multipliers are not sufficient to enforce occupancy validity and we need to use additional penalty terms. Furthermore, we discussed why ensuring ratio-based occupancy validity is compatible with the single-policy concentrability definition, resulting in learning a policy whose actual occupancy is within the data distribution.

However, one might wonder whether a more standard application of the ALM term, which involves adding a squared penalty on Bellman flow error, leads to a similar policy validity guarantee. This idea is appealing because it avoids variational lower bound and additional variables. However, here we provide an intuitive argument that a penalty term on Bellman flow error does not appear to be sufficient to ensure a ratio-based occupancy validity guarantee.

We use $\epsilon(s)$ to denote Bellman flow error defined as

$$
\epsilon(s) = (1-\gamma)\rho(s) + \gamma \sum_{s',a'} P(a|s',a')\mu(s',a')w(s',a') - \sum_a w(s,a)\mu(s,a).
$$

We show that even such a strong state-wise guarantee on Bellman error cannot generally lead to $\frac{d^{\pi_{\hat{w}}}(s)}{d_{\hat{w}}(s)}$ being bounded by a constant. We argue this by contradiction. Assume that for $0 \leq c < 1$

$$
\frac{d^{\pi_{\hat{w}}}(s)}{d_{\hat{w}}(s)} \leq \frac{1}{1-c} \iff d^{\pi_{\hat{w}}}(s) - d_{\hat{w}}(s) \leq cd^{\pi_{\hat{w}}}(s). \tag{74}
$$

Since $d^{\pi_{\hat{w}}}$ satisfies the Bellman flow equations, we can show $d^{\pi_{\hat{w}}} - d_{\hat{w}} = (I - \gamma P_{\pi_{\hat{w}}})^{-1}\epsilon$. Moreover, we have $d^{\pi_{\hat{w}}} = (I - \gamma P_{\pi_{\hat{w}}})^{-1}\rho$. Substituting these equations to (74), we conclude that for $0 \leq c < 1$

$$
(I - \gamma P_{\pi_{\hat{w}}})^{-1}\epsilon \leq c(I - \gamma P_{\pi_{\hat{w}}})^{-1}\rho \iff \epsilon \leq c\rho.
$$

Therefore, to ensure a constant bound on $\frac{d^{\pi_{\hat{w}}}(s)}{d_{\hat{w}}(s)}$, we require Bellman flow error to be pointwise smaller than the initial distribution. For state $s$ with $\rho(s) = 0$, this means that the Bellman flow error is required to be nonpositive: $\epsilon(s) \leq 0$. However, even state-wise minimization of squared penalty terms such as $\epsilon^2(s)$ can only ensure $|\epsilon(s)|$ to be small.

## F ROBUSTNESS TO MODEL MISSPECIFICATION AND OPTIMIZATION ERROR

In this section, we study the sample complexity of our algorithm in the presence of model misspecification and optimization error similar to Zhan et al. (2022).

Since in practice, it might be the case that our function classes $\mathcal{W}, \mathcal{V}$ do not contain $w^\star, v^\star$, similar to Zhan et al. (2022), we measure the approximation errors of $\mathcal{W}$ and $\mathcal{V}$ by

$$
\begin{aligned}
\epsilon_{r,v} &= \min_{v \in \mathcal{V}} \|v - v^\star\|_{1,\rho} + \|v - v^\star\|_{1,\mu} + \|v - v^\star\|_{1,\mu'}, \\
\epsilon_{r,w,w^\star} &= \min_{w \in \mathcal{W}} \|w - w^\star\|_{1,\mu},
\end{aligned} \tag{75}
$$

where $w^\star = d^\star/\mu$ and $d^\star$ is the (discounted) occupancy frequency of any optimal policy $\pi^\star$, $\|\cdot\|_{1,\rho}$ is weighted $l_1$ norm w.r.t. $\rho$, and $\mu'(s) = \sum_{s',a'} P(s|s',a')\mu(s',a')$. The model misspecification error is measured in $l_1$ norm, which is weaker than $l_\infty$ norm.

Furthermore, we also consider the optimization error of practical optimization algorithms since in real-world scenarios it is unlikely that an algorithm can recover the *exact* optimal solution. Instead, a typical optimization algorithm is able to find an approximate solution that is close enough to the true optimal solution. Formally, we assume that the solution $(\hat{w}, \hat{v})$ that the optimizer obtained satisfies

$$
\begin{aligned}
\hat{L}(\hat{w}, \hat{v}) - \min_{v \in \mathcal{V}} \hat{L}(\hat{w}, v) &\leq \epsilon_{o,v}, \\
\max_{w \in \mathcal{W}} \min_{v \in \mathcal{V}} \hat{L}(w, v) - \min_{v \in \mathcal{V}} \hat{L}(\hat{w}, v) &\leq \epsilon_{o,w},
\end{aligned}
\tag{76}
$$

where the objective $\hat{L}$ can be substituted by any objective with ALM term in different settings (e.g., $\hat{L} = \hat{L}_{\mathrm{AL}}^{\mathrm{CB}}$ in contextual bandits).

The assumption above is also similar to Zhan et al. (2022), and it assumes that $\hat{L}(\hat{w}, \hat{v}) \approx \max_{w \in \mathcal{W}} \min_{v \in \mathcal{V}} \hat{L}(w, v)$, which shows that $(\hat{w}, \hat{v})$ is an approximate max-min solution of $\hat{L}$.

With definition (75) and (76), the main result of this section is stated as follows:

**Theorem 5 (Robust version of Theorem 3)** *Assume concentrability of an optimal policy $\pi^\star$ (Definition 1) and let $w^\star(s,a) = d^{\pi^\star}(s,a)/\mu(s,a)$, $v^\star = J(\pi^\star)$. Assume that $|v(s)| \leq B_v$ for $v \in \mathcal{V}$ and $0 \leq w(s,a) \leq B_w$ for $w \in \mathcal{W}$. Moreover, assume (75) and (76) hold. Then for any fixed $\delta > 0$, policy $\hat{\pi}$ returned by Algorithm 2 (where $(\hat{w}, \hat{v})$ satisfies (76) with $\hat{L} = \hat{L}_{AL}^{CB}$ instead of the exact max-min solution as in (8)) achieves*

$$
J(\pi^\star) - J(\hat{\pi}) \lesssim (B_w + 1)^2 (B_v + 1) \sqrt{\frac{\log(|\mathcal{W}||\mathcal{V}|/\delta)}{N}} + \epsilon_{opt} + \epsilon_{mis}.
$$

*with probability at least $1 - \delta$, where $\epsilon_{opt} = \epsilon_{o,w} + \epsilon_{o,v}$ and $\epsilon_{mis} = (B_w + B_v + 3)\epsilon_{r,w,w^\star} + (B_w + 1)\epsilon_{r,v}$.*

Note that we only present the robustness result for contextual bandit settings for conciseness. Similar results also hold for MAB, model-based MDP, and model-free MDP settings.

*Proof.* Note that the proof is almost the same as Appendix D.3, except for part (II) of Lemma 6. Part (I) of Lemma 6 directly holds in model misspecification and optimization error by the same proof. Now we show part (II) of Lemma 6. For convenience, we use $L$, $\hat{L}$ to represent $L_{\mathrm{AL}}^{\mathrm{CB}}$, $\hat{L}_{\mathrm{AL}}^{\mathrm{CB}}$ respectively, and let $\epsilon_{\mathrm{stat}} = 3(B_w + 1)^2 (B_v + 1)\sqrt{\frac{\log(|\mathcal{W}||\mathcal{V}|/\delta)}{N}}$. Also, let

$$
\begin{aligned}
v_{\mathcal{V}}^\star &= \arg\min_{v \in \mathcal{V}} \|v - v^\star\|_{1,\rho} + \|v - v^\star\|_{1,\mu} + \|v - v^\star\|_{1,\mu'}, \\
w_{\mathcal{W}}^\star &= \arg\min_{w \in \mathcal{W}} \|w - w^\star\|_{1,\mu}.
\end{aligned}
$$

By the same argument as in Appendix D.3, $(w^\star, v^\star) \in \mathrm{argmax}_{w \geq 0} \mathrm{argmin}_v L(w, v)$. Also, we can decompose $L(w^\star, v^\star) - L(\hat{w}, v^\star)$ according to

$$
\begin{aligned}
&L(w^\star, v^\star) - L(\hat{w}, v^\star) \\
&= \underbrace{L(w^\star, v^\star) - L(w^\star, \hat{v}(w_{\mathcal{W}}^\star))}_{:=T_1} + \underbrace{L(w^\star, \hat{v}(w_{\mathcal{W}}^\star)) - L(w_{\mathcal{W}}^\star, \hat{v}(w_{\mathcal{W}}^\star))}_{:=T_2} \\
&\quad + \underbrace{L(w_{\mathcal{W}}^\star, \hat{v}(w_{\mathcal{W}}^\star)) - \hat{L}(w_{\mathcal{W}}^\star, \hat{v}(w_{\mathcal{W}}^\star))}_{:=T_3} + \underbrace{\hat{L}(w_{\mathcal{W}}^\star, \hat{v}(w_{\mathcal{W}}^\star)) - \hat{L}(\hat{w}, \hat{v})}_{:=T_4} \\
&\quad + \underbrace{\hat{L}(\hat{w}, \hat{v}) - \hat{L}(\hat{w}, v_{\mathcal{V}}^\star)}_{:=T_5} + \underbrace{\hat{L}(\hat{w}, v_{\mathcal{V}}^\star) - L(\hat{w}, v_{\mathcal{V}}^\star)}_{:=T_6} + \underbrace{L(\hat{w}, v_{\mathcal{V}}^\star) - L(\hat{w}, v^\star)}_{:=T_7},
\end{aligned}
$$

where $\hat{v}(w) = \arg\min_{v \in \mathcal{V}} \hat{L}(w, v)$. Each term is bounded as follows:

- $T_1 = 0$ because $w^\star$ satisfies the optimization constraints.
- $T_2 \leq (B_w + B_v + 3)\epsilon_{r,w,w^\star}$ by Lemma 12.
- $T_3 \leq \epsilon_{\mathrm{stat}}$ due to part (I) of Lemma 6.

- $T_4 \leq \epsilon_{o,w}$ because $\max_{w \in \mathcal{W}} \min_{v \in \mathcal{V}} \hat{L}(w,v) - \min_{v \in \mathcal{V}} \hat{L}(\hat{w}, v) \leq \epsilon_{o,w}$ and $w^\star_{\mathcal{W}} \in \mathcal{W}$.

- $T_5 \leq \epsilon_{o,v}$ because $\hat{L}(\hat{w}, \hat{v}) - \min_{v \in \mathcal{V}} \hat{L}(\hat{w}, v) \leq \epsilon_{o,v}$ and $v^\star_{\mathcal{V}} \in \mathcal{V}$.

- $T_6 \leq \epsilon_{\text{stat}}$ due to part (I) of Lemma 6.

- $T_7 \leq (B_w + 1)\epsilon_{r,v}$ by Lemma 12.

Summing up the bounds on each term, we have

$$L(w^\star, v^\star) - L(\hat{w}, v^\star) \leq 2\epsilon_{\text{stat}} + \epsilon_{\text{opt}} + \epsilon_{\text{mis}}.$$

The remaining steps are the same as Appendix D.3, and we can finally obtain that

$$J(\pi^\star) - J(\hat{\pi}) \lesssim (B_w + 1)^2 (B_v + 1)\sqrt{\frac{\log(|\mathcal{W}||\mathcal{V}|/\delta)}{N}} + \epsilon_{\text{opt}} + \epsilon_{\text{mis}}.$$

$\square$

**Remark 1** *Note that the suboptimality caused by model misspecification and optimization error in our algorithm is of order $O(\epsilon_{opt} + \epsilon_{mis})$. This is much better than the result of Zhan et al. (2022) where the suboptimality caused by model misspecification and optimization error is of order $O(\sqrt{(\epsilon_{opt} + \epsilon_{mis})/\alpha})$.*

Finally, we show and prove the following lemma which is key to the proof of our main theorem (Theorem 5) in this section. This is also similar to Zhan et al. (2022).

**Lemma 12** *Under the same setting as in Theorem 5, for any $v \in \mathcal{V}$ and any $w_1, w_2 \in \mathcal{W}$, it holds that*

$$|L(w, v_1) - L(w, v_2)| \leq (B_w + 1)(\|v_1 - v_2\|_{1,\rho} + \|v_1 - v_2\|_{1,\mu} + \|v_1 - v_2\|_{1,\mu'}).$$

*Also, for any $v_1, v_2 \in \mathcal{V}$ and any $w \in \mathcal{W}$, it holds that*

$$|L(w_1, v) - L(w_2, v)| \leq (B_w + B_v + 3)\|w_1 - w_2\|_{1,\mu}.$$

*Proof.* Recall that

$$L(w, v) = \mathbb{E}_\mu\left[w(s,a)r(s,a)\right] - \mathbb{E}_\mu[v(s)(w(s,a) - 1)] - \mathbb{E}_{s \sim \mu}[(\mathbb{E}_{a \sim \mu(\cdot|s)}[w(s,a)] - 1)^2],$$

and in contextual bandits we have $\rho = \mu$. Therefore, by definition,

$$
\begin{aligned}
&|L(w, v_1) - L(w, v_2)| \\
=&|\mathbb{E}_\mu[(v_1(s) - v_2(s))(w(s,a) - 1)]| \\
\leq&|\mathbb{E}_\mu[(v_1(s) - v_2(s))w(s,a)]| + |\mathbb{E}_\rho[v_1(s) - v_2(s)]| \\
\leq& B_w\|v_1 - v_2\|_{1,\mu} + \|v_1 - v_2\|_{1,\rho} \\
\leq&(B_w + 1)(\|v_1 - v_2\|_{1,\rho} + \|v_1 - v_2\|_{1,\mu} + \|v_1 - v_2\|_{1,\mu'}).
\end{aligned}
$$

Similarly, we have

$$
\begin{aligned}
&|L(w_1, v) - L(w_2, v)| \\
\leq&|\mathbb{E}_\mu[(w_1(s,a) - w_2(s,a))(r(s,a) - v(s))]| \\
&+ \mathbb{E}_{s \sim \mu}[|(\mathbb{E}_{a \sim \mu(\cdot|s)}[w_1(s,a)] - 1)^2 - (\mathbb{E}_{a \sim \mu(\cdot|s)}[w_2(s,a)] - 1)^2|] \\
\leq&(B_v + 1)\|w_1 - w_2\|_{1,\mu} \\
&+ \mathbb{E}_{s \sim \mu}[|\mathbb{E}_{a \sim \mu(\cdot|s)}[w_1(s,a) - w_2(s,a)](\mathbb{E}_{a \sim \mu(\cdot|s)}[w_1(s,a) + w_2(s,a)] - 2)|] \\
\leq&(B_v + 1)\|w_1 - w_2\|_{1,\mu} + (B_w + 2)\|w_1 - w_2\|_{1,\mu} \\
=&(B_w + B_v + 3)\|w_1 - w_2\|_{1,\mu}.
\end{aligned}
$$

$\square$

## G  AUXILIARY RESULTS

**Theorem 6 (Convergence of MLE for learning transitions ([Van de Geer](#), [2000]))** *Given a realizable model class $\mathcal{P} = \{P : (\mathcal{S}, \mathcal{A}) \to \Delta(\mathcal{S})\}$ that contains the true model $P^\star$ and a dataset $\mathcal{D}_m = \{(s_i, a_i, s_i')\}$ with $(s_i, a_i) \overset{iid}{\sim} \mu, s_i' \sim P^\star(\cdot|s_i, a_i)$, let $\hat{P}$ be*

$$\hat{P} = \arg\max_{P \in \mathcal{P}} \sum_{i=1}^{N} \ln P(s_i'|s_i, a_i).$$

*Fix the failure probability $\delta > 0$. Then, with probability at least $1 - \delta$, we have the following concentration on the squared Hellinger distance between $\hat{P}$ and $P^\star$:*

$$\mathbb{E}_{s,a\sim\mu} \left[ \sum_{s'} \left( \sqrt{\hat{P}(s'|s,a)} - \sqrt{P^\star(s'|s,a)} \right)^2 \right] \lesssim \frac{\log(|\mathcal{P}|/\delta)}{N}.$$

**Lemma 13** *For any $0 \leq b_i \leq B$ and $x_i, y_i \geq 0$ for $i \in \{0, \dots, n\}$, the following holds*

$$\left( \sqrt{\sum_{i=0}^{n} b_i x_i} - \sqrt{\sum_{i=0}^{n} b_i y_i} \right)^2 \leq B \sum_{i=0}^{n} (\sqrt{x_i} - \sqrt{y_i})^2. \tag{77}$$

*Proof.* We expend the left-hand side of (77), use Cauchy-Schwarz inequality, and then complete the square:

$$\left( \sqrt{\sum_{i=1}^{n} b_i x_i} - \sqrt{\sum_{i=1}^{n} b_i y_i} \right)^2 = \sum_i b_i x_i + \sum_i b_i y_i - 2\sqrt{\left( \sum_i b_i x_i \right)\left( \sum_i b_i y_i \right)}$$

$$\leq \sum_i b_i x_i + \sum_i b_i y_i - 2 \sum_i b_i \sqrt{x_i y_i}$$

$$= \sum_i (\sqrt{b_i x_i} - \sqrt{b_i y_i})^2 \leq B \sum_i (\sqrt{x_i} - \sqrt{y_i})^2.$$

$\square$

**Lemma 14** *For any two arbitrary sets $\mathcal{X}, \mathcal{Y}$, let $f(\cdot, \cdot) : \mathcal{X} \times \mathcal{Y} \to \mathbb{R}$ be an arbitrary function. Let $\mathcal{X}_0 = \{x \in \mathcal{X} \mid \inf_{y \in \mathcal{Y}} f(x, y) > -\infty\}$ and assume $\mathcal{X}_0$ is non-empty. For any $x \in \mathcal{X}_0$, assume there exists $y^*(x) \in \mathcal{Y}$ s.t. $f(x, y^*(x)) = \min_{y \in \mathcal{Y}} f(x, y)$. Also, let $\mathcal{X}_p^* = \{x \in \mathcal{X}_0 \mid x \in \arg\max_{x \in \mathcal{X}_0} f(x, y^\star(x))\}$ and assume $\mathcal{X}_p^*$ is non-empty. For a nonnegative function $A(\cdot)$ on $\mathcal{X}$, let $\mathcal{X}^* = \{x \in \mathcal{X}_p^* \mid A(x) = 0\}$ and assume $\mathcal{X}^*$ is non-empty. Define $f^{AL}(x, y) = f(x, y) - A(x)$. Then*

$$x \in \mathcal{X}_0 \iff \inf_{y \in \mathcal{Y}} f^{AL}(x, y) > -\infty.$$

*and for any $x \in \mathcal{X}_0$,*

$$x \in \mathcal{X}^* \iff x \in \arg\max_{x \in \mathcal{X}_0} \min_{y \in \mathcal{Y}} f^{AL}(x, y).$$

*Proof.* Note that for any fixed $x$, $f^{AL}(x, y)$ is a constant shift of $f(x, y)$, which implies that $\inf_{y \in \mathcal{Y}} f(x, y) > -\infty \iff \inf_{y \in \mathcal{Y}} f^{AL}(x, y) > -\infty$. This also implies that for any $x \in \mathcal{X}_0$, $\arg\min_{y \in \mathcal{Y}} f(x, y) = \arg\min_{y \in \mathcal{Y}} f^{AL}(x, y)$.

For any $x \in \mathcal{X}_0$, let $y^*(x)$ denote any one of $y \in \mathcal{Y}$ s.t. $f(x, y^*(x)) = \min_{y \in \mathcal{Y}} f(x, y)$.

Now for any $x^* \in \mathcal{X}^*$, we have

$$f(x^*, y^*(x^*)) \geq f(x, y^*(x)), \ \forall x \in \mathcal{X}_0.$$
$$\Longrightarrow f(x^*, y^*(x^*)) - A(x^*) \geq f(x, y^*(x)) - A(x), \ \forall x \in \mathcal{X}_0.$$
$$\Longrightarrow f^{AL}(x^*, y^*(x^*)) \geq f^{AL}(x, y^*(x)), \ \forall x \in \mathcal{X}_0.$$
$$\Longrightarrow \min_{y \in \mathcal{Y}} f^{AL}(x^*, y) \geq \min_{y \in \mathcal{Y}} f^{AL}(x, y), \ \forall x \in \mathcal{X}_0.$$
$$\Longrightarrow x^* \in \arg\max_{x \in \mathcal{X}_0} \min_{y \in \mathcal{Y}} f^{AL}(x, y).$$

For the other direction, given any $x_0 \in \arg\max_{x \in \mathcal{X}_0} \min_{y \in \mathcal{Y}} f^{AL}(x, y)$, we have

$$
\begin{aligned}
\min_{y \in \mathcal{Y}} f^{AL}(x_0, y) &\geq \min_{y \in \mathcal{Y}} f^{AL}(x, y), \ \forall x \in \mathcal{X}_0. \\
\Longrightarrow f^{AL}(x_0, y^*(x_0)) &\geq f^{AL}(x, y^*(x)), \ \forall x \in \mathcal{X}_0.
\end{aligned}
\tag{78}
$$

Fix any $x^* \in \mathcal{X}^* \subseteq \mathcal{X}_0$, we have $f(x^*, y^*(x^*)) \geq f(x_0, y^*(x_0))$ and $-A(x^*) \geq -A(x_0)$ by definition. Now assume $x_0 \notin \mathcal{X}^*$. Then either $f(x^*, y^*(x^*)) > f(x_0, y^*(x_0))$ if $x_0 \notin \mathcal{X}_p^*$, or $-A(x^*) > -A(x_0)$ if $x_0 \in \mathcal{X}_p^* \backslash \mathcal{X}^*$. Either one of the above two conditions implies that

$$
f(x^*, y^*(x^*)) - A(x^*) > f(x_0, y^*(x_0)) - A(x_0) \Longrightarrow f^{AL}(x^*, y^*(x^*)) > f^{AL}(x_0, y^*(x_0)),
$$

which contradicts with (78). Therefore, $x_0 \in \mathcal{X}^*$. $\qquad\square$

