# OpenReview forum: "Optimal Conservative Offline RL with General Function Approximation via Augmented Lagrangian"
_ICLR.cc/2023/Conference — ICLR 2023 notable top 25%_

### Official Review · Reviewer_f5U7 · 2022-10-25

**Confidence:** 3
**Clarity, Quality, Novelty And Reproducibility:** The paper is generally well-written a…
**Correctness:** 3
**Technical Novelty And Significance:** 3
**Empirical Novelty And Significance:** Not applicable
**Recommendation:** 6

**Strength And Weaknesses:**

Strengths:
- The paper addresses an important problem of offline reinforcement learning.
- The theoretical results are original in this setting.
- While the MIS formulation has been used in PRO-RL (Zhan et al., 2022) to achieve the sample efficiency of offline RL under weak assumptions, this paper shows that the MIS formulation combined with the augmented Lagrangian method can reach the optimality in sample efficiency of offline RL. This is an interesting result.

Weaknesses:
- The paper makes many typos. For example, the first sentence in Section 3.2: "... cements..."; a typos in Proposition 2; $B_v$ has not been defined in Theorem 2;...
- It would be nice if the authors can explain intuitively why replacing behavior regularization with the augmented Lagrangian is key for the optimality of their proposed algorithms.
- The authors claim that their algorithms are practical but do not provide any experiments.




**Summary Of The Paper:**

This paper addresses the offline reinforcement learning problem with general function approximations and the partial coverage of offline data. Existing works either are computationally intractable or are suboptimal in statistical rates. The paper presents the first offline RL algorithms that are practical and optimal in regret under the assumption of single-policy concentrability. To obtain these properties, the paper uses the marginalized importance sampling (MIS) formulation combined with the augmented Lagrangian method.





**Summary Of The Review:**

This is a good paper. I recommend the acceptance.

---

> ### Author Response · Authors · 2022-11-18
> **Response (Part 1/2)**
>
> We thank the reviewer for the time spent reviewing our work, thoughtful comments, and positive feedback. We provide our response to the points raised by the reviewer below. All references are to the revised version of the paper.
>
> ### Intuition on why replacing behavior regularization with ALM is the key to optimality
>
> >  It would be nice if the authors can explain intuitively why replacing behavior regularization with the augmented Lagrangian is key for the optimality of their proposed algorithms.
>
> In our original submission, we provided some intuitive explanation of why behavior regularization fails to achieve the optimal rate in paragraph 6 of page 3. We have improved this paragraph in the revised paper and below, we expand more on intuition and compare behavior regularization with the workings of the ALM.
>
> **Short answer:** The short intuitive answer to the above question is as follows: in contrast to the augmented Lagrangian method which is unbiased, behavior regularization introduces bias in the objective. The bias introduced by behavior regularization is the culprit behind the suboptimal rate.
>
> **Long answer:** We provide intuition on several aspects of behavior regularization and ALM:
>
> 1. *Behavior regularization is biased and ALM is not.* This is because behavior regularization adds the term $\alpha E_\mu \Big[ f \Big( \frac{d(s,a)}{\mu(s,a)}\Big)\Big]$. This term is not necessarily zero at the optimal solution $d^\star(s,a)$ since $f(d^\star(s,a)/\mu(s,a)) \neq 0$. On the other hand, the ALM term is $\mathbb{E}_{d^{\pi_w}} \Big(\frac{d_w(s)}{d^{\pi_w}(s)} - 1\Big)^2$, which is zero when $w = w^\star$ as the optimal solution to optimization problem (8) must satisfy the validity constraints.
>
> 2. *Why behavior regularization is suboptimal beyond MABs.*
> Since the term $E_\mu \Big[ f \Big( \frac{d^\star(s,a)}{\mu(s,a)}\Big)\Big]$ might be large even at the optimal solution (for example a constant like $f(C^\star)$), to control the bias, one needs to set $\alpha = O \Big( \frac{1}{\sqrt{N}} \Big)$ (as we do so in the PRO-MAB guarantee in Theorem 1). Furthermore, to control the error in estimating the expected return $E_{d^{\pi_w}}[r(s,a)]$ by $E_{d_w}[r(s,a)]$ (expectation over a different measure), one needs to ensure occupancy validity: $d_w(s)$ is not too far from $d^{\pi_w}(s)$. In the case of MAB, we were successfully able to set $\alpha$ to be small and still ensure occupancy validity. However, as we show in Appendix D.2.2, setting $\alpha = O \Big( \frac{1}{\sqrt{N}} \Big)$ is not strong enough to ensure that $d_w(s)$ is not too far from $d^{\pi_w}(s)$ in (almost) every state.
>
> To put differently, behavior regularization introduces curvature to the optimization problem, which reduces variance (thus improves over unregularized MIS) but introduces bias, and beyond the MAB setting, even balancing bias-variance tradeoff by carefully setting $\alpha$ leads to a rate worse than $1/\sqrt{N}$ (Proposition 3). On the contrary, the augmented Lagrangian method does not introduce any bias. Thus, we can set the coefficient of the ALM to be large enough to ensure that the variance is sufficiently reduced without impacting the bias and thereby achieving the optimal sample complexity.
>
> 3. *Why ALM works.* In the paper, we have provided intuition on why ALM combined with MIS works (e.g. the paragraph below Equation (10) or paragraph 9 on page 3). We also explain it here in two steps:
>
> * Step 1: *MIS leads to bounded weights.* In the MIS formulation, we learn $w(s,a) = \frac{d_w(s,a)}{\mu(s,a)}$, where $w$ belongs to a function class with bounded values $w(s,a) \leq B_w$ (a typical and necessary assumption). Therefore,
>
> $ \qquad \qquad \frac{d_w(s,a)}{\mu(s,a)} \leq B_w \qquad (1)$
>
> * Step 2: *ALM leads to occupancy validity.* Consider the objective (10) for simplicity, which also minimizes the following ALM term:
>
> $\qquad \qquad E_{d^{\pi_w}} \Big(\frac{d_w(s)}{d^{\pi_w}(s)} - 1\Big)^2$
>
> Minimizing the above term ensures that for states with significant $d^{\pi_w}(s)$, the ratio $\frac{d_w(s)}{d^{\pi_w}(s)}$ cannot be too small. In other words, the reverse of this ratio must be smaller than a constant (as we formally prove in Lemma 11):
>
> $ \qquad \qquad \frac{d^{\pi_w}(s)}{d_w(s)} = \frac{d^{\pi_w}(s) \pi_w(a|s)}{d_w(s)\pi_w(a|s)} = \frac{d^{\pi_w}(s,a)}{d_w(s,a)} \leq c \qquad (2)$
>
> Multiplying both sides of (1) by (2), we conclude that
>
> $ \qquad \qquad \frac{d_w(s,a)}{\mu(s,a)} \times \frac{d^{\pi_w}(s,a)}{d_w(s,a)} = \frac{d^{\pi_w}(s,a)}{\mu(s,a)} \leq c B_w$
>
> In summary, MIS combined with ALM results in learning $w$ such that the *actual* occupancy $d^{\pi_w}$ of the induced policy $\pi_w$, has a bounded concentrability coefficient, i.e. it learns a policy covered by data.

---

> > ### Author Response · Authors · 2022-11-18
> > **Response (Part 2/2)**
> >
> > ### On the practicality of our algorithm.
> >
> > > The authors claim that their algorithms are practical but do not provide any experiments.
> >
> > We clarify that by practicality, we refer to the fact that our provable algorithm can be implemented and combined with general function approximators such as deep neural networks. This is in contrast to the majority of existing provable offline RL algorithms that are impractical, for example,
> > * They require oracle access to *uncertainty quantification*, such as Jin et al. 2021, However, existing empirical uncertainty estimation methods for nonlinear function approximators are unreliable as extensively discussed and demonstrated by Yu et al., 2021.
> > * They are *computationally intractable*, for example, they require enumeration over all functions in the hypothesis class or rely on an oracle that correctly eliminates all hypothesis functions in the class that disagree with the offline data with high probability (Xie et al., (2021), Uehara and Sun (2022).
> >
> > We plan to carry out experiments on our algorithm in a future work.
> >
> > ### Typos
> > We thank the reviewer for pointing out the typos. We have revised the paper accordingly.
> >
> > ---
> >
> > **References.**
> >
> > Jin, Ying, Zhuoran Yang, and Zhaoran Wang. "Is pessimism provably efficient for offline RL?." International Conference on Machine Learning. PMLR, 2021.
> >
> > Yu, T., Kumar, A., Rafailov, R., Rajeswaran, A., Levine, S., and Finn, C. (2021). Combo: Conservative offline model-based policy optimization. Advances in neural information processing systems, 34, 28954-28967.
> >
> > Xie, T., Cheng, C. A., Jiang, N., Mineiro, P., and Agarwal, A. (2021). Bellman-consistent pessimism for offline reinforcement learning. Advances in neural information processing systems, 34, 6683-6694.
> >
> > Uehara, M., and Sun, W. (2021, September). Pessimistic Model-based Offline Reinforcement Learning under Partial Coverage. In International Conference on Learning Representations.
> >
> > ---
> >
> > We hope that we have integrated the reviewer's feedback in the revised draft and answered their questions. We are happy to answer any questions and discuss further in case the reviewer believes there are any missing details.

---

### Official Review · Reviewer_WYf4 · 2022-10-26

**Confidence:** 4
**Correctness:** 3
**Technical Novelty And Significance:** 3
**Empirical Novelty And Significance:** Not applicable
**Recommendation:** 8

**Clarity, Quality, Novelty And Reproducibility:**

## Clarity

The major unclear statement to me is related to "enforcers of state occupancy validity" (using ALM). If I understand correctly, the Bellman flow constraint in PRO-RL already enforces the state occupancy to be valid? I think using ALM in the MAB/CB makes sense as I feel the parameter w will have a scaling problem if no squared penalty is added. However, I don't quite understand the intuition and reason for that in the MDP setting.

As mentioned in the previous block, this paper has many results, and it is quite hard to capture them. For example, this paper discusses CB/MAB/MDP and PRO-RL(CB) with/without regularization + the ALM method proposed in the paper. Adding a table would be very helpful.

Minor: zeta on pg 8 seems to be undefined.

In the appendix (e.g., B.6), it seems better to use statistical error than closeness in the lemma title.

## Quality

Overall, the paper is of good quality.

## Novelty

The most novel part to me is the introduction of ALM and the study on the more special MAB/CB cases. The remaining part and the analysis seem to be more standard and adapted from the literature. In case I'm missing something, feel free to comment on that.

## Reproducibility

I don't have time to check the proof, so I'm not able to evaluate in this aspect.

## Others

Instead of assuming the coverage of pi* and comparing with it, can you handle the arbitrary data distribution (Sec 4.3 in Zhan et al. (2022)) or the robustness case as in Zhan et al. (2022) and Chen & Jiang (2022)?

The major weakness to me is that this paper requires additional assumption other than the realizable W class + realizable V class in the MDP setting. For model-based CORAL, an additional realizable model class is assumed. For the model-free CORAL, an additional U class is assumed. I want to comment that the assumption on the U class is completeness type because when we add more functions to the W class, the realizability of U class will be broken and we need to enlarge the U class as well. However, we do not know the u_w^* for w, so it is not easy to expand the U class. I understand (from the title and intro) the scope of the paper is only pi* concentrability, but it would be certainly great to also consider weak realizability type assumption on the function approximation side.

Related to that, I am curious that why you do not require similar additional assumption in MAB/CB settings? My intuition is that we only encounter completeness type assumption in the MDP case (e.g., more than one horizon). It would be great if you can comment on that.

Can you comment on the computational side of the algorithm? It seems that the algorithm is not very efficient as a max-min or a harder max-min(-min)-max optimization oracle is needed. A side comment is that similar oracles are used in [1][2].

[1] Model-free Representation Learning and Exploration in Low-rank MDPs

[2] Efficient reinforcement learning in block mdps: A model-free representation learning approach

**Strength And Weaknesses:**

## Strengths

Compared with Zhan et al. (2022), the rate is better and the realizability assumption is on the true density function instead of the regularized one.

Compared with Chen & Jiang (2022), this paper drops the gap assumption.

Some investigations in CB/MAB are interesting. For example, a lower bound on PRO-CB in prop 3 shows that PRO-RL cannot get the optimal rate.

## Weaknesses

In the MDP case, the assumption is stronger. Compared with prior works Zhan et al. (2022) and Chen & Jiang (2022), this paper requires an additional realizable model class for model-based CORAL or a completeness-type U class for model-free CORAL.

The results could be presented in a clearer way. This paper includes many results, and it is very hard to track and compare them. I think adding a table that summarize the assumptions and results in the current paper and the most related works will be very helpful and beneficial to readers.

**Summary Of The Paper:**

This paper studies offline RL (MAB/CB/MDP) under general function approximation and single-policy coverage. The authors leverage MIS formulation plus augmented Langragian method (ALM) and identify that the key for solving problem is to ensure that certain state occupancy validity constraints are nearly satisfied.

**Summary Of The Review:**

I lean towards acceptance. I would appreciate if the author could address my questions.

--
Score 6 --> 8 after rebuttal.

---

> ### Author Response · Authors · 2022-11-18
> **Response (Part 1/6)**
>
> We thank the reviewer for the time spent reviewing our paper, positive evaluation of our work, interesting questions, and thoughtful feedback that helped us improve our manuscript. We provide our response to the reviewer's comments and questions below. All references are to the latest version of the paper.
>
> ### On the realizability/completeness assumptions
>
> > In the MDP case, the assumption is stronger. Compared with prior works Zhan et al. (2022) and Chen & Jiang (2022), this paper requires an additional realizable model class for model-based CORAL or a completeness-type $U$ class for model-free CORAL.
>
> > The major weakness to me is that this paper requires additional assumption other than the realizable $W$ class + realizable V class in the MDP setting. For model-based CORAL, an additional realizable model class is assumed. For the model-free CORAL, an additional U class is assumed. I want to comment that the assumption on the U class is completeness type because when we add more functions to the $W$ class, the realizability of U class will be broken and we need to enlarge the U class as well. However, we do not know the $u_w^*$ for w, so it is not easy to expand the U class. I understand (from the title and intro) the scope of the paper is only pi* concentrability, but it would be certainly great to also consider weak realizability type assumption on the function approximation side.
>
>
> **Completeness-type assumption.** We agree with the reviewer that our assumption of $u^\star_w \in \mathcal{U}$ for $w \in \mathcal{W}$ is a completeness-type assumption. However, as we explain in the paragraph below Theorem 4, nearly all offline RL algorithms with general function approximation rely on it or similar assumptions. For example, Cheng et al. (2022) assume $\mathcal{T} f \in \mathcal{F}, \forall f \in \mathcal{F}$, Xie et al. (2021) assume $\mathcal{T}^\pi f \in \mathcal{F}, \forall f \in \mathcal{F}, \forall \pi \in \Pi$, and Jiang and Huang (2020)  require $w^\star \in \mathcal{W}, Q^\pi \in \mathcal{F}, \forall \pi \in \Pi$. Removing the completeness assumption is a major open problem in theory and the recent work of Foster et al. (2021) proved that even with all-policy concentrability and realizability of all value functions, without the completeness assumption, polynomial sample complexity in offline RL is impossible.
>
> We are aware of only three works that circumvent the completeness assumptions, but these come at the expense of other strong assumptions and limitations. We have discussed these works in Appendix B and further explain the limitations below:
>
> * The work of Xie and Jiang (2021) assumes a much stronger data coverage, even stronger than all-policy concentrability ("pushforward concentrability"), which requires $P(s'|s,a)/\mu(s') \leq C, \forall s,a,s'$. Moreover, the algorithm is computationally intractable as it requires enumeration over all functions in the hypothesis class.
>
> * Chen and Jiang (2022) additionally require a positive $Q^\star$ gap and the performance guarantee of the algorithm degrades with inverse of $Q^\star$ gap: $1/\sqrt{N}\text{gap}(Q^\star)$. The algorithm is also intractable and requires uncertainty quantifiers.
>
> * The algorithm of Zhan et al. (2022) achieves the rate of $1/N^{1/6}$ and requires concentrability of optimal policy $\pi^\star$ as well as optimal regularized policy $\pi^\star_\alpha$, which can be strong as we discuss in Section 5.3. Additionally, we want to bring up a subtle point regarding the realizability assumptions in Zhan et al. (2022). Note that in addition to realizability of optimal solutions $w^\star, v^\star$, Zhan et al. (2022) also require realizability of $w^\star_\alpha$ and $v^\star_\alpha$. Although Zhan et al. (2022) argue they only need two-policy realizability for a fixed $\alpha$, the typical definition of sample complexity is for any $\epsilon > 0$, with $O(1/\text{poly}(\epsilon))$ samples, the algorithm can obtain a policy with suboptimality at most $\epsilon$ with high probability. However, if the function class of Zhan et al. (2022) indeed contains only two policies $\pi^\star$ and $\pi^\star_\alpha$ for a fixed $\alpha$, they can only achieve sample complexity guarantee w.r.t. a fixed $\epsilon$. This is therefore a much weaker result than typical sample complexity results. To achieve the typical sample complexity result for any $\epsilon > 0$, the assumption of Zhan et al. (2022) should be modified to for any $\alpha \in [0,\bar{\alpha}]$ where $\bar{\alpha}$ is a small constant, $w_{\alpha}^*$ and $v_{\alpha}^*$ are realizable. This is actually stronger than a two-policy realizability assumption. In contrast, our approach does not suffer from this issue and guarantees hold for any $\epsilon > 0$.

---

> > ### Author Response · Authors · 2022-11-18
> > **Response (Part 2/6)**
> >
> > **Additional function classes and realizability assumptions.** Although our algorithm uses additional realizable function classes such as the model class $\mathcal{P}$ that contains the true transitions or the class $\mathcal{U}$, these additional realizable classes don't appear to pose a major hurdle in practice. For example, when handling complex observations, one can use a shared representation across these classes.
> >
> > > Related to that, I am curious that why you do not require similar additional assumption in MAB/CB settings? My intuition is that we only encounter completeness type assumption in the MDP case (e.g., more than one horizon). It would be great if you can comment on that.
> >
> > In short, additional assumptions in the MDP setting stem from the fact that the state occupancy is *not* fixed in MDPs and is affected by the policy due to transitions. This is in contrast to the MAB/CB setting, where the state occupancy is always fixed. Since state occupancy is a key part of the ALM term, additional assumptions in the MDP setting help ensure an accurate estimation of the ALM term is possible.
> >
> > In particular, recall that the ALM term aims at enforcing the true state occupancy to be close to the estimated state occupancy, where closeness is *weighted by the true state occupancy* since closeness is more important in states that are actually visited more.
> >
> > * In MAB/CB setting, since no transition kernel is involved, the state occupancy distribution is always equal to the state distribution in data $d^{\pi_w}(s) = \mu(s)$. Therefore, without any additional assumptions, the ALM term can be expressed as an expectation with respect to $\mu$ and accurately estimated from offline data.
> > * In the MDP setting, the ALM term involves an expectation w.r.t. the actual state occupancy $d^{\pi_w}$ of the policy $\pi_w$ induced from weights $w$. Since in the MDP case, we do not have direct samples from this state distribution, we estimate the ALM term via a variational lower bound, which only involves expectations over offline data. The additional assumption $u^\star_w \in \mathcal{U}$ for any $w \in \mathcal{W}$ allows us to ensure that the variational lower bound is accurate enough for estimating the ALM term for all possible $w$.
> >
> > ### Comparison and results tables
> >
> > > The results could be presented in a clearer way. This paper includes many results, and it is very hard to track and compare them. I think adding a table that summarize the assumptions and results in the current paper and the most related works will be very helpful and beneficial to readers.
> >
> > > As mentioned in the previous block, this paper has many results, and it is quite hard to capture them. For example, this paper discusses CB/MAB/MDP and PRO-RL(CB) with/without regularization + the ALM method proposed in the paper. Adding a table would be very helpful.
> >
> > We thank the reviewer for their suggestion. We have revised the paper accordingly:
> >
> > * We added Table 1 in Appendix A which summarizes our results on MAB, CB, and MDPs.
> >
> > * We added Table 2 in Appendix B.2 which compares various aspects of our offline RL algorithm with prior works.

---

> > > ### Author Response · Authors · 2022-11-18
> > > **Response (Part 3/6)**
> > >
> > > ### Occupancy validity
> > >
> > > > The major unclear statement to me is related to "enforcers of state occupancy validity" (using ALM). If I understand correctly, the Bellman flow constraint in PRO-RL already enforces the state occupancy to be valid? I think using ALM in the MAB/CB makes sense as I feel the parameter w will have a scaling problem if no squared penalty is added. However, I don't quite understand the intuition and reason for that in the MDP setting.
> > >
> > > The reviewer brings up an interesting point. To answer the reviewer's question, we make several comments:
> > >
> > > **We prove that Lagrange multipliers alone (= unregularized MIS) do not sufficiently enforce occupancies to be valid.**
> > >
> > > * In all settings MAB/CB/MDP, the constraints of the original optimization problem are indeed *occupancy validity* constraints. However, as we proved in Proposition 1, even in a simple 2-armed bandit case, solving the sample-based approximation of the Lagrange multiplier objective (unregularized MIS) does not sufficiently enforce these constraints and fails. Moreover, in Proposition 2, we prove that if we only search over $w$ with valid occupancies, then unregularized MIS is optimal. Therefore, we demonstrated that the reason unregularized MIS fails is *only* because occupancy validity constraints are not enforced enough with the Lagrange multiplier method. This is in stark contrast to prior work that believed unregularized MIS fails because there are no terms to "promote conservatism" for handling partial data coverage.
> > >
> > > * Therefore, since the PRO-RL algorithm, which is MIS with behavior regularization, has theoretical guarantees, its behavior regularizer term (and not the Bellman flow Lagrange multiplier term) is occupancy validity (one can conclude this based on Lemma 8 in Zhan et al. (2022)). However, this enforcement is done in a manner that additionally introduces bias and leads to a suboptimal statistical rate.
> > >
> > > **We use ALM to sufficiently enforce occupancy validity constraints in MAB, CB, and MDPs.** In all three settings, we use ALM which *adds an additional term* on violating the occupancy validity constraints, resulting in sufficiently enforcing the state occupancy validity. Additionally, we show that only in MABs (but not in CB/MDP), behavior regularization can also be used to optimally enforce occupancy validity constraints. With occupancy validity constraints enforced through these additional terms, we prove that the MIS achieves an optimal convergence rate. Therefore, we have demonstrated that the empirical observation that additional terms and regularizers that appear to be necessary in practice (Yang et al., 2020, Lee et al., 2021) are really "enforcers of occupancy validity" than "promoters of conservatism".
> > >
> > > **The right scaling.** The reviewer's understanding is correct that the penalty terms address scaling issues in $w$, which is particularly intuitive in the MAB/CB cases---note how correcting the scaling can be interpreted as $w$ inducing a valid occupancy $d_w = w \mu$ in MAB and CB settings. However, we found that correct scaling is not sufficient in MDPs and we require learned occupancy $d_w(s)$ not to be very small compared to the actual occupancy $d^{\pi_w}(s)$.
> > >
> > > **Bellman flow constraints (even added via both Lagrange multiplier and the ALM) is insufficient in enforcing occupancy validity.**
> > >
> > > > the Bellman flow constraint in PRO-RL already enforces the state occupancy to be valid?
> > >
> > > * We discussed above that Lagrange multipliers (which add Bellman flow constraints in the MDP case) are not sufficient to enforce occupancy validity and we need to use additional penalty terms. Moreover, the reviewer's question touches upon the following question: why not use squares of the original constraints (Bellman flow error) as the ALM term? The short answer is we find that penalty on Bellman flow error does not lead to the ratio guarantee on occupancy validity $\frac{d^{\pi_{\hat w}}(s)}{d_{\hat w}(s)} \leq \text{const}$, which we require in our proofs; see e.g., the expected advantage guarantee in part (I) of Lemma 11 and compare with performance difference lemma on page 31. We added an intuitive argument to Appendix E.8 of the revised paper to demonstrate the insufficiency of using Bellman flow errors in the ALM term.

---

> > > > ### Author Response · Authors · 2022-11-18
> > > > **Response (Part 4/6)**
> > > >
> > > > * We also point to another reason for the insufficiency of the Bellman flow constraints for enforcing occupancy validity. Even when considering the population objective, the objective expressed in terms of importance weights $w = d/\mu$ turns out not to be exactly equivalent to the original LP expressed in terms of $d$. The subtle issue arises when there exists a state $s$ with $\mu(s) = 0$. Consider the example in Section 5.3, in the original LP (in terms of $d$), there is a constraint on state $C$ enforcing the validity of the state occupancy. However, in the converted program, since $\mu(C) = 0$, the constraint on state $C$ is “disappeared”, which is the reason that even with infinite data, the unregularized algorithm still fails to recognize the optimal policy. Our ALM term can be expressed as the expectation w.r.t.  $d^{\pi_d}$, where $\pi_d(a|s) = \frac{d(s,a)}{\sum_a d(s,a)}$ is the policy computed by $d$ and $d^{\pi_d}$ is the state occupancy frequency of $\pi_d$. This ensures that even if there exists a state $s$ s.t. $\mu(s) = 0$, the ALM term still enforces constraint on that state. We want to emphasize that any penalty term that involves an expectation w.r.t. $\mu$ fails to work in this example. Note that the PRO-RL algorithm guarantee holds under the assumption $w_{\alpha}^\star \leq B_{w,\alpha}$, which implicitly enforces that $\mu(C) > 0$ in this example. Therefore, their algorithm actually fails in this example similar to the unregularized MIS algorithm, whereas our method with the ALM term succeeds.
> > > >
> > > > **Intuition behind the ratio guarantee.** Our particular ALM construction can be intuitively understood as follows: the MIS formulation learns bounded weights $\hat w(s,a) = d_{\hat w}(s,a)/\mu(s,a) \leq B_w$, and the ALM term ensures that $d_{\hat w}(s,a)/d^{\pi_{\hat w}}(s,a) = d_{\hat w}(s)/d^{\pi_{\hat w}}(s) = \Omega(1)$ for most states, which translates to ${d^{\pi_{\hat w}}(s,a)}/{\mu(s,a)} \lesssim B_w$.
> > > >
> > > >
> > > > ### Novelty
> > > >
> > > > > The most novel part to me is the introduction of ALM and the study on the more special MAB/CB cases. The remaining part and the analysis seem to be more standard and adapted from the literature. In case I'm missing something, feel free to comment on that.
> > > >
> > > > In addition to the introduction of the ALM to offline learning and our study in the special MAB and CB settings, we make the following novel contributions:
> > > > * **Our design of the ALM for offline RL is novel and highly non-trivial.** Recall that the standard augmented Lagrangian method simply adds the *unweighted* sum of squared constraints. Below, we discuss how our incorporation of the ALM in offline RL involves several novel extensions both on the ALM as well as on the way we applied ALM to MAB/CB.
> > > >
> > > >     (1) **We do not use the original constraints of the optimization problem.** While the ALM terms in the MAB/CB settings involve squared constraints of the original optimization problem, our proposed ALM term for the RL case is different. Note that in the LP formulation of offline RL, the constraints are Bellman flow equations. Thus, the standard ALM would involve squared Bellman flow error terms:
> > > >
> > > >     $\left((1-\gamma)\rho(s) + \gamma \sum_{s',a'} P(s|s',a')d(s',a') - d(s)\right)^2$
> > > >
> > > >     However, as we discussed above in the Occupancy validity segment and Appendix E.8, we found that using squared Bellman flow error as ALM is insufficient for obtaining our desired optimality guarantees. We addressed this by constructing the ALM via a new ratio-based occupancy validity constraint $\frac{d_w(s)}{d^{\pi_w}(s)} = 1$. We also prove that such changes to the augmented Lagrangian method are valid in Lemma 13. Therefore, instead of the Bellman flow errors, our ALM includes the following terms:
> > > >
> > > >     $ \qquad \qquad \left( \frac{d_w(s)}{d^{\pi_w}(s)} - 1\right)^2.$
> > > >
> > > >     (2) **We identify the correct ratio to be used in the ALM term.** Recall that the iterands in the ALM term in the CB setting are:
> > > >
> > > >     $ \qquad \qquad \left(\sum_a w(s,a) \mu(a|s) - 1\right)^2 = \left(\frac{d_w(s)}{\mu(s)} -1\right)^2$
> > > >
> > > >     It might appear that we can use the ratios $\frac{d_w(s)}{\mu(s)}$ in the offline RL setting as well, since they enforce conditionals to be valid distributions. Moreover, note that similar ratios of the form $\frac{d_w(s,a)}{\mu(s,a)}$ are widely-used in behavior regularization methods (Nachum et al., 2019; Lee et al., 2021; Zhan et al., 2022;). However, we identify that the correct extension to the MDP setting is using ratios $\frac{d_w(s)}{d^{\pi_w}(s)}$ instead of $\frac{d_w(s)}{\mu(s)}$.

---

> > > > > ### Author Response · Authors · 2022-11-18
> > > > > **Response (Part 5/6)**
> > > > >
> > > > > ### Novelty (Cont.)
> > > > > *
> > > > >    (3) **We add a sum of squared constraints, weighted by actual state occupancy.** While the standard ALM uses an unweighted sum of constraints and our application of ALM to the MAB/CB setting involves weights $\mu(s)$, we show that the correct extension to the MDP setting is weighting the terms according to the actual occupancy $d^{\pi_w}(s)$. Note that this is again in contrast to behavior regularization methods which use weights $\mu(s)$.
> > > > >
> > > > >    (4) **We interpret the ALM term as an $f$-divergence between the actual and learned occupancies to estimate it via offline data.** Our design of the ALM term is based on ensuring a bound on the ratio of $d_w$ and $d^{\pi_w}$ in actually visited states visited by $d^{\pi_w}$. However, this term is difficult to estimate as it both involves a ratio and expectation w.r.t. unknown $d^{\pi_w}$ (even statistically accurate estimation of $d^{\pi_w}$ would lead to large errors). To resolve this, we propose a novel idea of viewing the ALM as an $f$-divergence (or more specifically, Chi-squared) and inspired by Nguyen et al. (2010) and Nachum et al. (2019), we write the ALM term in the variational form (11) and then convert it to (13) which only involves expectations w.r.t. offline data.
> > > > >
> > > > >    (5) **Differences between penalty terms at a glance.** Below, we summarize different penalty options to highlight their distinctions:
> > > > >
> > > > >      - Weighted ALM applied to original LP formulation of RL:  $E_{\mu} \left[\left((1-\gamma)\rho(s) + \gamma \sum_{s',a'} P(s|s',a')d(s',a') - d(s)\right)^2\right]$
> > > > >
> > > > >      - Behavior regularization (Nachum et al., 2019; Lee et al., 2021; Zhan et al., 2022): ${E}_{s,a \sim \mu} \left[f\left(\frac{d_w(s,a)}{\mu(s,a)}\right) \right]$
> > > > >
> > > > >      - Our proposed ALM term for CB: ${E}_{s \sim \mu} \left[\left(\frac{d_w(s)}{\mu(s)} -1\right)^2\right]$  Notice the distinction with behavior regularization even when choosing $f(x) =(x-1)^2$. The above term enforces the learned state occupancy $d_w(s)$ to match the actual state occupancy which turns out to be equal to $\mu(s)$ in CBs.
> > > > >
> > > > >      - Our proposed ALM term for MDPs: ${E}_{s \sim d^{\pi_w}} \left[\left(\frac{d_w(s)}{d^{\pi_w}(s)}-1\right)^2\right]$ Notice further distinctions with behavior regularization: behavior regularization minimizes the $f$-divergence between learned occupancy and data distribution whereas our ALM method minimizes the $f$-divergence between learned occupancy and the actual occupancy of the induced policy.
> > > > >
> > > > > * **Our suboptimality analysis is novel and reveals new insights.** Specifically, the majority of prior works on conservative offline RL use lower confidence bound arguments on value function (Jin et al., 2021, Xie et al., 2021). The work of Zhan et al. (2022) bounds the suboptimality by $\|\hat w - w^\star_\alpha\|_{1, \mu}$, i.e., the $\ell_1$ norm of the difference between learned weights $\hat w$ and optimal (regularized) weights $w^\star_\alpha$. They then connect this norm to the statistical approximation error relying on the strong convexity of the objective induced by adding a strongly convex behavior regularizer.
> > > > >
> > > > >     In contrast, we avoid introducing norms and instead connect suboptimality to learned occupancy validity, i.e. closeness of $d_{\hat w}$ is to $d^{\pi_{\hat w}}$, which is a new technique (c.f. Lemma 11 and the main argument of Theorem 4).
> > > > >
> > > > >     Our new analysis technique leads to the following:
> > > > >
> > > > >     (1) Allows us to prove that behavior regularization can indeed achieve the optimal $1/\sqrt{N}$ rate in MABs and improve over the $1/N^{1/6}$ shown by Zhan et al. (2022).
> > > > >
> > > > >     (2) Reveals the importance of occupancy validity, which paved the way for our design of the ALM technique and may inspire new algorithm design in theory and practice.

---

> > > > > > ### Author Response · Authors · 2022-11-18
> > > > > > **Response (Part 6/6)**
> > > > > >
> > > > > > ### Robustness and competing with the best-covered policy
> > > > > >
> > > > > > > Instead of assuming the coverage of $\pi^\star$ and comparing with it, can you handle the arbitrary data distribution (Sec 4.3 in Zhan et al. (2022)) or the robustness case as in Zhan et al. (2022) and Chen and Jiang (2022)?
> > > > > >
> > > > > > **Robustness.** Yes, our algorithms enjoy robustness results with respect to an average $\ell_1$ norm (instead of the usual $\ell_\infty$ robustness) on both model misspecification and optimization error. Following the proof of Zhan et al. (2022), we proved the robustness of our algorithm in the CB setting in Appendix F of the revised paper. We believe that robustness for RL case can be proved similarly. We note that our algorithm enjoys better robustness of $O(\epsilon_{\text{opt}} + \epsilon_{\text{mis}})$ compared to behavior regularization, which achieves robustness of $O((\epsilon_{\text{opt}} + \epsilon_{\text{mis}})/\alpha)$ (e.g. $\alpha \asymp 1/N^6$) as analyzed by Zhan et al. (2022).
> > > > > >
> > > > > > **Competing with any covered policy.** Our upper bound analysis currently has one step that uses the optimality of the target policy $\pi^\star$. Specifically, we use the fact that when the target policy is optimal, its advantage is nonpositive $A^{\pi^\star}(s,a) \leq 0$. This allows us to prove that the ALM term must be small at learned weights $\hat w$ and thus enforces occupancy validity. We plan to investigate whether (variants of) the ALM approach can also compete with any covered policy in future work.
> > > > > >
> > > > > > ### Computational side
> > > > > >
> > > > > > > Can you comment on the computational side of the algorithm? It seems that the algorithm is not very efficient as a max-min or a harder max-min(-min)-max optimization oracle is needed. A side comment is that similar oracles are used in [1][2].
> > > > > >
> > > > > > For MAB, CB, and model-based RL, our algorithms need to solve a max-min(-min) problem. For model-free RL, the max-min-min-max can actually be converted to a max(-max)-min(-min) problem. This is because we can first exchange $\min_u$ and $\max_{\zeta}$ since $L_{\text{AL}}^{\text{model-free}}$ as defined in (53) is convex-concave w.r.t. $(u,\zeta)$. Then, we can exchange $\min_v$ and $\max_{\zeta}$ since $v$ and $\zeta$ are not coupling in  $L_{\text{AL}}^{\text{model-free}}$. Therefore, we only need to solve a max-min problem for the model-free RL as well.
> > > > > >
> > > > > > Thus, our algorithms only require a max-min oracle, which is also required in prior works on provable conservative offline RL with general function approximators (e.g., Zhan et al. 2022). Many other prior works require much stronger oracles such as ones that can identify the subset of function class compatible with offline dataset (e.g., Xie et al., 2021). Moreover, many practically successful offline RL algorithms also solve minimax problems such as DICE family (Nachum et al., 2019; Yang et al., 2020; Lee et al., 2021).
> > > > > >
> > > > > > ---
> > > > > >
> > > > > > We hope that we have integrated the reviewer's feedback and answered their questions. We are happy to answer any questions and discuss further in case the reviewer believes there are any missing details.

---

> > > > > > > ### Author Response · Authors · 2022-11-18
> > > > > > > **Response (References)**
> > > > > > >
> > > > > > > **References**
> > > > > > >
> > > > > > > Xie, T., \& Jiang, N. (2021, July). Batch value-function approximation with only realizability. In International Conference on Machine Learning (pp. 11404-11413). PMLR.
> > > > > > >
> > > > > > > Cheng, C. A., Xie, T., Jiang, N., \& Agarwal, A. (2022). Adversarially trained actor critic for offline reinforcement learning. arXiv preprint arXiv:2202.02446.
> > > > > > >
> > > > > > > Jiang, Nan, and Jiawei Huang. "Minimax value interval for off-policy evaluation and policy optimization." Advances in Neural Information Processing Systems 33 (2020): 2747-2758.
> > > > > > >
> > > > > > > Xie, T., Cheng, C. A., Jiang, N., Mineiro, P., and Agarwal, A. (2021). Bellman-consistent pessimism for offline reinforcement learning. Advances in neural information processing systems, 34, 6683-6694.
> > > > > > >
> > > > > > > Chen, Jinglin, and Nan Jiang. "Offline Reinforcement Learning Under Value and Density-Ratio Realizability: the Power of Gaps." In The 38th Conference on Uncertainty in Artificial Intelligence. 2022.
> > > > > > >
> > > > > > > Jin, Ying, Zhuoran Yang, and Zhaoran Wang. "Is pessimism provably efficient for offline rl?." In International Conference on Machine Learning, pp. 5084-5096. PMLR, 2021.
> > > > > > >
> > > > > > > Ofir Nachum, Bo Dai, Ilya Kostrikov, Yinlam Chow, Lihong Li, and Dale Schuurmans. AlgaeDICE: Policy gradient from arbitrary experience. arXiv preprint arXiv:1912.02074, 2019.
> > > > > > >
> > > > > > > Jongmin Lee, Wonseok Jeon, Byungjun Lee, Joelle Pineau, and Kee-Eung Kim. OptiDICE: Offline policy optimization via stationary distribution correction estimation. In International Conference on Machine Learning, pp. 6120–6130. PMLR, 2021.
> > > > > > >
> > > > > > > Modi, Aditya, Jinglin Chen, Akshay Krishnamurthy, Nan Jiang, and Alekh Agarwal. "Model-free representation learning and exploration in low-rank mdps." arXiv preprint arXiv:2102.07035 (2021).
> > > > > > >
> > > > > > > Zhang, Xuezhou, Yuda Song, Masatoshi Uehara, Mengdi Wang, Alekh Agarwal, and Wen Sun. "Efficient reinforcement learning in block mdps: A model-free representation learning approach." In International Conference on Machine Learning, pp. 26517-26547. PMLR, 2022.
> > > > > > >
> > > > > > > Mengjiao Yang, Ofir Nachum, Bo Dai, Lihong Li, and Dale Schuurmans. Off-policy evaluation via the regularized Lagrangian. Advances in Neural Information Processing Systems, 33:6551–6561, 2020.
> > > > > > >
> > > > > > > XuanLong Nguyen, Martin J Wainwright, and Michael I Jordan. Estimating divergence functionals and the likelihood ratio by convex risk minimization. IEEE Transactions on Information Theory, 56(11):5847–5861, 2010.

---

> > > > > > > > ### Comment · Reviewer_WYf4 · 2022-11-18
> > > > > > > > **Thanks for the detailed response and update**
> > > > > > > >
> > > > > > > > I have read all reviews and responses. Thanks a lot for addressing my questions on writing, robustness results, occupancy validity, novelty, and computational complexity. The discussions addressed my concerns, and I would like to bump up my score to 8.
> > > > > > > >
> > > > > > > > Regarding the computational complexity, maybe it is helpful to add the discussion above on how to get the final min-max form to the final version of the paper. As you mentioned, min-max oracles are used in offline RL algorithms as well as some online RL algorithms [1][2].

---

> > > > > > > > > ### Author Response · Authors · 2022-11-18
> > > > > > > > > **Thanks for reading our response and updating the score**
> > > > > > > > >
> > > > > > > > > We would like to express our gratitude to the reviewer for reading our response and updating their score. We are glad that we have addressed their concerns. Based on the reviewer's suggestion, we have submitted an updated draft, adding the discussion on computational complexity to Appendix E.1 on practical implementation details.

---

### Official Review · Reviewer_bCR3 · 2022-11-02

**Confidence:** 2
**Correctness:** 4
**Technical Novelty And Significance:** 3
**Empirical Novelty And Significance:** Not applicable
**Recommendation:** 8

**Clarity, Quality, Novelty And Reproducibility:**

Clarity: Despite the somewhat dense topic, the paper is quite clear. In particular, the organization of the narrative (MAB to contextual to MDP) helps to build understanding. There is some unusual notation involving squiggly less-than signs which, while defined up front, might be better to avoid unless alternatives really hurt clarity.

Novelty: The point of the paper is to show that a relatively simple trick (adding an augmented Lagrangian term to the existing Lagrangian to enforce feasibility) is "enough", as the title says.

Originality: The paper seems original enough, although I am not deep enough into this subfield of conservative offline RL to be able to tell conclusively.

Reproducibility: the work is wholly theoretical so reproducibility is not a concern.

**Strength And Weaknesses:**

Strengths: The paper gives a clear presentation of improved algorithms for a topic which, while somewhat niche, is convincingly relevant.

Weaknesses: there aren't any obvious weaknesses to the paper.

**Summary Of The Paper:**

The paper deals with offline RL problems, i.e. learning an optimal policy given some data sampled from a behavior policy.

Here, while the behavior policy itself is assumed known, its state-action occupancy distribution is available only through samples. The offline RL approach considered is "marginal importance sampling": the goal is to learn weights for each state-action pair that will transform the estimated behavior occupancy into a new occupancy for the optimal policy. Another key assumption is "single-policy concentrability": in particular there is some $C^*$ which upper bounds the largest importance-sampling ratio required (for any state-action pair) between the behavior policy and the optimal policy.

Offline RL with these assumptions is relatively well-studied previously. The previous approaches the authors compare to all have a general theme: they are primal-dual methods, maximizing expected reward, and using a Lagrangian to enforce a constraint that the occupancy distribution be a valid probability distribution. Such approaches also include a regularization term to keep the learned policy close to the behavior policy. The authors' improvement is to replace this regularization with an extra quadratic penalty on the weight feasibility (forgoing behavior regularization), which improves performance in MAB, contextual bandits, and full MDP settings.

**Summary Of The Review:**

The paper seems like a reasonable theoretical contribution to a topic relevant to the ICLR audience, so I recommend acceptance.

---

> ### Author Response · Authors · 2022-11-18
> **Response**
>
> We thank the reviewer for the time spent evaluating our paper, thorough review, and thoughtful comments. We appreciate their positive evaluation of our work and are glad that they didn't find any obvious weaknesses in the paper.

---

### Official Review · Reviewer_sKtr · 2022-11-03

**Confidence:** 3
**Correctness:** 4
**Technical Novelty And Significance:** 4
**Empirical Novelty And Significance:** 2
**Recommendation:** 8

**Clarity, Quality, Novelty And Reproducibility:**

This paper is polished and well written. The exposition is good and the discussion is easy to follow. Detailed discussion of the context of this work and its related work made appreciating the contributions easy. The result appear novel though I'm not familiar enough with the topic to assert this confidently.


**Strength And Weaknesses:**

The results are conclusive and insightful. Assist by their theoretical contributions, the authors make a strong case for the impact of occupancy validity on the performance in offline learning and provide an interesting explanation for the limitations of behavior regularization. Their proposed approach seems likely to inspire future work, both theoretical and empirical.

Minor nitpicks and questions
============

Not that it matters much, but the authors might want to consider a different acronym. I would avoid CORAL since it's already quite popular (in general) and already has a notable presence in some parts of ML where it is known as "correlation alignment", for instance [1] (one of several top search results).

Big nitpick, "THE optimal policy", should be "some" or maybe add a quick word to say we'll only consider one arbitrary optimal policy from this point on.

On a related topic, what impact does having multiple optimal policies have? I would expect that it would only make some definitions and theorem statements more tedious. If the authors are willing to indulge my curiosity, do you think considering multiple policies could offer anything useful, for instance, looser occupancy validity requirements, in some interesting special cases with many such policies?

p. 5, "the objective (15)", should this be a reference to (2) instead?

I've mostly seen augmented lagrangian methods with a coefficient scaling the penalty term which typically grows over time. Is it's omission at all relevant or is it simply an irrelevant implementation detail (in the context of this work)?

p. 6, I wasn't able to allocate enough time to figure out the steps going from (10) to (12). Could the authors provide some additional details or point me to a similar derivation in related work?

Somewhat tangential to this work, but the discussions related to the significance of the occupancy validity reminded me of recognizers [2], which was proposed as a way to describe target policies in the context of off-policy learning. The basic idea is that you can define the policy by probabilistically accepting or rejecting actions sampled by the behavior policy. This offers leads to better behaved off-policy learning than standard importance sampling. This is most likely the result of constraining target policies, but I wonder if this work's insight and analysis could reveal something interesting? Maybe new useful ways to constrain the target policies in cases where we don't seek optimality (e.g., in off-policy learning)?


[1] Sun, B., Feng, J. and Saenko, K., 2017. Correlation alignment for unsupervised domain adaptation. In Domain Adaptation in Computer Vision Applications (pp. 153-171). Springer, Cham.
[2] Precup, Doina, Cosmin Paduraru, Anna Koop, Richard S. Sutton, and Satinder Singh. "Off-policy learning with options and recognizers." Advances in Neural Information Processing Systems 18 (2005).

**Summary Of The Paper:**

The authors propose an augmented lagrangian approach to various offline decision making learning problems based on marginal importance sampling using function approximation. The authors show that their method exhibits improved qualities over behavior regularization, such as improved (sub)optimality. The present 4 variants of their approach, specialized for multi-arm bandits, contextual bandits and MDPs (model-based and model-free). The authors conclude with a small illustrative example comparing their approach to behavior regularization.

**Summary Of The Review:**

This paper is interesting and of high quality, and I support it's acceptance.

---

> ### Author Response · Authors · 2022-11-18
> **Response (Part 1/2)**
>
> We thank the reviewer for the time spent reading our paper, insightful feedback and suggestions, interesting questions, and a positive review of our work. We respond to the reviewer's comments below.
>
> ### Algorithm's name
>
> > Not that it matters much, but the authors might want to consider a different acronym. I would avoid CORAL since it's already quite popular (in general) and already has a notable presence in some parts of ML where it is known as "correlation alignment", for instance [1] (one of several top search results).
>
> We thank the reviewer for their suggestion. We have changed algorithm names to ALMIS (augmented Lagrangian method with marginalized importance sampling) for offline bandits and RL.
>
>
> ### Multiple optimal policies
>
> > Big nitpick, "THE optimal policy", should be "some" or maybe add a quick word to say we'll only consider one arbitrary optimal policy from this point on.
>
> We thank the reviewer for pointing this out. Our algorithm and its guarantees work for any (covered) optimal policy as stated e.g. in Theorems 3 and 4. We have revised the draft to reflect that we are competing with an optimal policy in other places.
>
> > On a related topic, what impact does having multiple optimal policies have? I would expect that it would only make some definitions and theorem statements more tedious. If the authors are willing to indulge my curiosity, do you think considering multiple policies could offer anything useful, for instance, looser occupancy validity requirements, in some interesting special cases with many such policies?
>
> Our analysis remains the same, even if we have multiple optimal policies, provided that at least one optimal policy is covered in the single-policy concentrability sense. However, having multiple optimal policies (especially when one allows for stochastic policies) can favorably impact the concentrability coefficient. For example, consider the case of multi-armed bandits with $K$ optimal arms and consider competing with an optimal policy that has a uniform distribution over these $K$ arms. Then the concentrability coefficient becomes $\frac{1/K}{\mu(a)} \leq C^\star$ for $a \in \{1, \dots, K\}$. Compared to the concentrability coefficient of an optimal policy that is deterministic on one of these arms $\frac{1}{\mu(a^\star)} \leq C^\star$, the former is smaller by a factor of $1/K$. Note that this also affects the value of $B_w$ which must satisfy $B_w \geq C^\star$ due to the realizability of $w^\star$.
>
> ### Objective reference
>
> > p. 5, "the objective (15)", should this be a reference to (2) instead?
>
> The reference to (15) ((16) in the revised paper) is correct since objective (2) is the population version of the optimization problem and (16) is the corresponding empirical version. In Proposition 1, we prove that the unregularized MIS algorithm, which solves the empirical version of the objective, fails in bandits. The intuition behind the failure of unregularized MIS is that when the initial distribution of the suboptimal arm is very small, there is a constant probability that the empirical reward of the suboptimal arm exceeds that of the optimal arm and the unregularized algorithm cannot recognize the optimal one.
>
> ### Scaling coefficient in the augmented Lagrangian method.
>
> > I've mostly seen augmented lagrangian methods with a coefficient scaling the penalty term which typically grows over time. Is it's omission at all relevant or is it simply an irrelevant implementation detail (in the context of this work)?
>
> In our proofs, we only require the scaling parameter for the augmented Lagrangian term to be a constant with respect to the number of samples $N$ and infinite norm-bound parameters such as $B_w$. For simplicity, here we set it to one, since, unlike behavior regularization, this scaling parameter does not require tuning to handle bias/variance tradeoff (as ALM does not introduce any bias).
>
> In practical implementation, particularly when solving the optimization problem in iterations, one can use the approach you mentioned of growing the scaling parameters over iterations.

---

> > ### Author Response · Authors · 2022-11-18
> > **Response (Part 2/2)**
> >
> > ### Steps going from (10) to (12)
> >
> > > p. 6, I wasn't able to allocate enough time to figure out the steps going from (10) to (12). Could the authors provide some additional details or point me to a similar derivation in related work?
> >
> > Although Nachum et al. (2019) provide derivation and details of a similar result, in the paper, we immediately conclude (12) (now (13)) from (10) (now (11)) by interpreting $u(s,a)$ as a Q-function. We provide a more detailed explanation of our argument below.
> >
> > First, notice that the term $J_{\tilde{\mathcal{M}}}(\pi_w) := E_{d^{\pi_w}}[f_*(x(s,a))]$ in (11) is the *expected return* (discounted cumulative reward) of policy $\pi_w$ in an MDP $\tilde{\mathcal{M}}$ with the original MDP dynamics (the MDP from which we collected offline data) but rewards $f_*(x(s,a))$. Let $u(s,a)$ denote the Q-function of policy $\pi_w$ in MDP $\tilde{\mathcal{M}}$. This Q-function satisfies the Bellman equation (Equation (12) in the paper):
> >
> > $\qquad u(s,a) = f_*(x(s,a)) + \gamma (\mathbb{P}^{\pi_w}) u(s,a) \iff x(s,a) = f_*^{-1} \left(u(s,a) - \gamma (\mathbb{P}^{\pi_w} u)(s,a) \right)$
> >
> > Here, we also wrote $x(s,a)$ in terms of $u(s,a)$ which comes in handy later. Now, we write the expected return (first term in (10)) in terms of an expectation over Q-function, taken with respect to initial state distribution $\rho$ and policy $\pi_w$:
> >
> > $\qquad J_{\tilde{\mathcal{M}}}(\pi_w) = E_{d^{\pi_w}}[f_*(x(s,a))] = (1-\gamma) E_{s \sim \rho, a \sim \pi_w} [u(s,a)]$
> >
> > For the second term in (11), we conduct a change of variable and write $x(s,a)$ in terms of $u(s,a)$ and use the fact that by definition $d_w(s,a)= w(s,a) \mu(s,a)$:
> >
> > $\qquad E_{d_w}[x(s,a)] = E_\mu [w(s,a) x(s,a)] = E_\mu [w(s,a) f_*^{-1} \left(u(s,a) - \gamma (\mathbb{P}^{\pi_w} u)(s,a) \right)].$
> >
> > Summing up the first and second terms gives Equation (13).
> >
> > ### Recognizers
> >
> > > Somewhat tangential to this work, but the discussions related to the significance of the occupancy validity reminded me of recognizers [2], which was proposed as a way to describe target policies in the context of off-policy learning. The basic idea is that you can define the policy by probabilistically accepting or rejecting actions sampled by the behavior policy. This offers leads to better behaved off-policy learning than standard importance sampling. This is most likely the result of constraining target policies, but I wonder if this work's insight and analysis could reveal something interesting? Maybe new useful ways to constrain the target policies in cases where we don't seek optimality (e.g., in off-policy learning)?
> >
> > We thank the reviewer for bringing up this interesting line of work to our attention. The motivation for using recognizers is twofold: (1) to specify natural target policies in the options framework; (2) to ensure that the target policy has a bounded IS ratio with reduced variance. The second motivation is of the same spirit as the single-policy concentrability framework studied in our paper: both aim to constrain the *target policy* in a way that the IS ratio between the target/behavior policies should be bounded, except that in our setting, this constraint is more often interpreted as a constraint for the behavior policy, because the target policy is usually taken to be the optimal policy. Moreover, new challenges arise when we move from the target policy to *estimated (learned) policy*: compared to the off-policy learning setting, our optimal offline RL setting requires novel ideas such as ALM to reduce the statistical error of learned policies. However, we agree with the reviewer that the work of recognizers reveals other interesting ways to constrain the target policy apart from the single-policy concentrability framework. In this context, we believe that our ALM-based algorithms and analysis can be adapted to deal with the more general case of competing with arbitrary target policies with bounded IS ratio, which in turn provides an approach to learning the recognizer-based target policies.
> >
> >
> > **References**
> >
> > Ofir Nachum, Yinlam Chow, Bo Dai, and Lihong Li. DualDICE: Behavior-agnostic estimation of discounted stationary distribution corrections. In Advances in Neural Information Processing Systems, pp. 2315–2325, 2019a.
> >
> > ---
> >
> > We hope that we have integrated the reviewer's feedback and answered their questions. We are happy to answer any questions and discuss further in case the reviewer believes there are any missing details.

---

### Author Response · Authors · 2022-11-18
**Summary of response and updates to the paper**

We thank all reviewers for the time spent reviewing our paper and their positive assessment of our work. We have updated the paper addressing the points raised by the reviewers, fixed typos, and improved the organization of the appendix. Here is a list of changes we made to the new draft:

*  Based on reviewer WYf4's question on robustness guarantees, we added Appendix F showing the theoretical guarantee of our algorithms in presence of model misspecification and optimization error.
* Based on reviewer WYf4's question on using Bellman flow constraints for occupancy validity, we added a new Appendix E.8 arguing insufficiency of an ALM term designed based on Bellman flow error in ensuring ratio-based occupancy validity.
* Based on reviewer WYf4's suggestion, we have included two additional tables in Appendix A. The first table summarizes our theoretical findings on MAB, CB, and MDP settings. The second table compares various aspects (such as assumptions and suboptimality guarantees) of our offline RL algorithm with several related work.
* Based on reviewer f5U7's suggestion, we have improved the intuitive explanation of why replacing behavior regularization with the augmented Lagrangian is key for the optimality of our algorithms (page 3).
* Based on reviewer sKtr's suggestion, we changed the name of our algorithm from CORAL to ALMIS (augmented Lagrangian method with marginalized importance sampling) for offline MAB, CB, and RL.
* We have addressed the typos in the paper, including those raised by reviewer f5U7 (such as defining $B_v$ in Theorem 2) and reviewer WYf4 (such as defining $\zeta$ on page 8). We have also improved the clarity of theorem statements.
* We found that our upper bound guarantee in Theorem 1 which was $O \left(\frac{\text{polylog}(N)}{\sqrt{N}} \right)$ can be improved to $O \left(\frac{1}{\sqrt{N}} \right)$. We modified the theorem statement and proof accordingly. We have also made the proof of Proposition 3 stronger.
* We have improved the organization of the proofs in the appendix.
* We have changed the title of the paper to Optimal Conservative Offline RL with General Function Approximation via Augmented Lagrangian.

We hope that we have integrated feedback from all reviewers in the revised paper and answered their questions in the individual responses below. We are happy to discuss more and answer any additional questions the reviewers might have.

---

### Decision · Program_Chairs · 2023-01-20

**Decision:**

Accept: notable-top-25%

**Justification For Why Not Higher Score:**

It seems that the offline RL community is gradually moving away from this classical formulation of learning purely based on offline data. After all, once you deploy that learned policy, you enter the online learning setting. An offline+online framework seems to be more appropriate, e.g. the Neurips 2022 offline RL workshop. For this very reason, I think the topic of this paper is slightly outdated for an inspiring oral presentation.

**Justification For Why Not Lower Score:**

The technical contribution seems novel to me and can be of independent interest to the RL community beyond the current setting being studied.

**Metareview: Summary, Strengths And Weaknesses:**

This paper studies offline RL with nonlinear function approximation and under single-policy concentrability. The paper presents a novel algorithm using the augmented Lagrangian method and leverages the marginalized importance sampling (MIS) in its analysis. The presentation is clear and the contribution is significant. There are no glaring weakness about this paper other than the minor typos and clarification issues that reviewers have raised. I vote for acceptance of this manuscript.

**Note From Pc:**

if the above contains the word "oral" or "spotlight" please see: "oral" presentation means -> notable-top-5% and "spotlight" means -> notable-top-25%. As stated in our emails, we are disassociating presentation type from AC recommendations